EMBO
Molecular Medicine

# TFE3 fusions drive oxidative metabolism and ferroptosis resistance in translocation renal cell carcinoma

Alexandra Helleux [1,2,3,4], Guillaume Davidson [1,2,3,4], Antonin Lallement[1,2,3,4], Fatima Al Hourani[1,2,3,4], Alexandre Haller[1,2,3,4], Isabelle Michel[1,2,3,4], Anas Fadloun[1,2,3,4], Christelle Thibault-Carpentier [1,2,3,4], Xiaoping Su[5], Véronique Lindner[6], Thibault Tricard[7], Hervé Lang[7], Nizar M Tannir[8], Irwin Davidson [1,2,3,4,10]✉ & Gabriel G Malouf [1,2,3,4,9,10]✉

## Abstract

The oncogenic mechanisms by which TFE3 fusion proteins drive translocation renal cell carcinoma (tRCC) are poorly characterized. Here, we integrated loss and gain of function experiments with multi-omics analyses in tRCC cell lines and patient tumors. High nuclear accumulation of NONO-TFE3 or PRCC-TFE3 fusion proteins promotes their broad binding across the genome at H3K27ac-marked active chromatin, engaging a core set of M/E-box-containing regulatory elements to activate specific gene expression programs as well as promiscuous binding to active promoters to stimulate mRNA synthesis. Within the core program, TFE3 fusions directly regulate genes involved in ferroptosis resistance and oxidative phosphorylation metabolism (OxPhos). Consequently, human tRCC tumors display high OxPhos scores that persist during their epithelial to mesenchymal transition (EMT). We further show that tRCC tumor aggressiveness is related to their EMT and their associated enrichment in myofibroblast cancer-associated fibroblasts (myCAFs) that are both hallmarks of poor prognostic outcomes. We define tRCC as a novel metabolic subtype of renal cancer and provide unique insights into how broad genomic binding of TFE3 fusion proteins regulates OxPhos and ferroptosis resistance.

**Keywords** TFE3; Metabolism; Ferroptosis; RNA Synthesis; Cancer Associated Fibroblasts
**Subject Categories** Cancer; Chromatin, Transcription & Genomics; Urogenital System

## Introduction

TFE3 and TFEB are members of the microphthalmia-associated transcription (MiT) family that regulate lysosomal function, nutrient/stress response and drug resistance (Raben and Puertollano, 2016). Translocation renal cell carcinomas (tRCC) are characterized by genetic translocation events involving one member of the (MiT) family, most frequently *TFE3* (90% of cases) (Bakouny et al, 2021; Caliò et al, 2019). TRCC is a rare subtype of kidney cancer that represents 1–4% of all adult RCC (Ellati et al, 2017; Moch et al, 2016), but is the most frequent RCC arising in children and adolescents (Raben and Puertollano, 2016; Geller et al, 2015; Sukov et al, 2012). The translocations result in expression of chimeric fusion proteins comprising TFE3 and another gene partner, under the control of constitutively active gene promoters (Kauffman et al, 2014). The complexity of tRCC is underscored by the identification of over 20 TFE3-partners, accentuating histological and clinical heterogeneity. Among these, *SFPQ*, *ASPSCR1*, *PRCC*, and *NONO* are the most frequent fusion partners. Aside these translocations, no other highly recurrent mutations have been identified, suggesting that the fusion proteins are the primary drivers of transformation. In agreement with this, all fusion proteins retain the TFE3 DNA binding and dimerization (basic-helix-loop-helix leucine-zipper; bHLH-LZ) domains allowing them to bind to cis-regulatory elements in the genome. Despite recent recognition of tRCC as a distinct RCC subtype, the precise oncogenic mechanisms fueling tRCC tumorigenesis remain elusive. Consequently, there is an urgent medical need to unravel the molecular function of TFE3-fusion proteins to comprehensively understand the molecular basis of tRCC and identify novel therapeutic targets.

To better understand how TFE3 fusion proteins drive transformation in tRCC, we performed gain and loss of function experiments complemented by multi-omics approaches to profile genome occupancy of TFE3 fusions, identify active cis-regulatory elements and characterize the gene expression programs under their control in model cell lines and human tumors. Our findings

[1]Institut de Génétique et de Biologie Moléculaire et Cellulaire, BP 163, 67404 Illkirch, France. [2]Centre National de la Recherche Scientifique, UMR7104, 67404 Illkirch, France. [3]Institut National de la Santé et de la Recherche Médicale, U1258, 67404 Illkirch, France. [4]Université de Strasbourg, 67404 Illkirch, France. [5]Department of Bioinformatics and Computational Biology, The University of Texas MD Anderson Cancer Center, Houston, TX, USA. [6]Department of Pathology, Strasbourg University Hospital, Strasbourg, France. [7]Department of Urology, Strasbourg University Hospital, Strasbourg, France. [8]Department of Genitourinary Medical Oncology, The University of Texas MD Anderson Cancer Center Houston, Houston, TX, USA. [9]Department of Medical Oncology, Institut de Cancérologie de Strasbourg-Europe, Strasbourg, France. [10]These authors contributed equally as senior authors: Irwin Davidson, Gabriel G Malouf. ✉E-mail: irwin@igbmc.fr; maloufg@igbmc.fr

revealed that NONO-TFE3 and PRCC-TFE3 fusions exhibit broad binding to thousands of sites across the genome, engaging distal and proximal M/E-box-containing regulatory elements, as well as promiscuously associating with active promoters. Amongst the core regulated genes are those driving oxidative phosphorylation (OxPhos) metabolism, culminating in elevated OxPhos levels in tRCC cells, and those promoting ferroptosis resistance. Analyses of RNA-seq of tRCC patient tumors demonstrated heightened OxPhos scores and ferroptosis resistance genes confirming the observations from the cell lines. Remarkably, elevated OxPhos score persisted in tumors that had undergone epithelial to mesenchymal transition (EMT), a known marker of tumor aggressiveness. Furthermore, using transcriptional signatures derived from clear cell renal carcinoma (ccRCC), we revealed that tRCC tumors with EMT signatures were enriched in myofibroblastic cancer-associated fibroblasts (myCAFs) whose presence correlated with poorer survival. Thus, EMT and presence of myCAFs emerge as common hallmarks, adversely impacting the prognosis for both ccRCC and tRCC.

## Results

### Abundant constitutive nuclear expression of TFE3 fusion proteins is essential for tRCC cell line viability

To address the role of TFE3 fusion proteins in tRCC, we used 4 cell lines expressing either NONO-TFE3 (lines UOK109, TF1) or PRCC-TFE3 (UOK120, UOK146). Each fusion protein presents a different breakpoint within TFE3, but conserve its bHLH-LZ domain maintaining their ability to specifically bind DNA. To assess expression of TFE3 fusion proteins compared to native TFE3, we performed immunoblots on extracts from the above lines as well as 2 non-transformed renal cell lines (RPTEC, HEK293T), 4 clear-cell (cc)RCC lines (RCC4, UOK121, A-498, 786-O) and 2 papillary (p)RCC lines (UOK112, ACHN). High expression of NONO-TFE3 and lower expression of PRCC-TFE3 fusions in the corresponding tRCC lines was observed, while expression of native TFE3 in the other cell lines was variable, but considerably lower than the fusion proteins (Appendix Fig. S1A). In contrast, no native TFE3 protein was visible in tRCC lines consistent with the *TFE3* gene location on the X chromosome and hence in cells of male origin, UOK109, TF1 and UOK120, the single copy of *TFE3* is subject to translocation (Grépin et al, 2014; Clark et al, 1997; Sidhar, 1996) whereas in the female UOK146 line expression of the non-translocated allele transcript is silenced (Clark et al, 1997). In addition, biochemical fractionation of the nuclear and cytoplasmic compartments showed strong enrichment of TFE3 fusion proteins in the nucleus similar to the control BRG1, the catalytic subunit of the SWI/SNF chromatin remodeling complex, but unlike GAPDH enriched in the cytoplasm (Appendix Fig. S1B). Thus, the tRCC cell lines abundantly express the expected nuclear TFE3-fusion proteins.

To characterize the gene expression programs of the tRCC cell lines, triplicate RNA-seq was performed for each line. More than 80% of the 3000 most highly expressed protein-coding genes were common to all 4 cell lines (Appendix Fig. S1C) and were enriched for pathways previously associated with function of native TFE3 such as lysosome function and endocytosis

(Appendix Fig. S1D). In addition, various metabolic pathways were enriched notably those linked to the electron transport chain, oxidative phosphorylation (OxPhos), reactive oxygen species (ROS), and carbon metabolism (Appendix Fig. S1D). Each line displayed hybrid EMT-type markers (Appendix Fig. S1E). For example, while TF1 and UOK146 showed highest expression of the epithelial marker *EPCAM*, they overexpress the mesenchymal *PRXX1* and *SNAI1/2* genes, respectively. Of note, the EMT regulators *SNAI1/2* were mostly expressed in the PRCC-TFE3 lines as compared to *TWIST1* or *TWIST2* most expressed in the NONO-TFE3 lines. Overall, UOK120 appeared to be the most mesenchymal line with the lowest expression of most epithelial markers, and higher expression of multiple mesenchymal markers and/or regulators (Appendix Fig. S1E).

To assess the role of TFE3 fusion proteins in the transformed phenotype, their expression was silenced using siRNA specifically directed against the TFE3 C-terminal region and compared with non-targeting control siRNA. RT-qPCR indicated efficient knock-down of TFE3 in the 2 PRCC-TFE3 lines (UOK120, UOK146; >80% reduction) as compared to control. Less efficient silencing was observed in the NONO-TFE3 lines (UOK109, TF1) perhaps due to the higher fusion expression (Fig. 1A). Similar effects were seen on immunoblots with strongly reduced levels of fusion protein in UOK120 and UOK146 lines, with less attenuated effects in UOK109 and TF1 lines (Fig. 1B). TFE3 fusion silencing dramatically reduced colony forming ability with a reduction reaching 90% for both NONO-TFE3 cell lines as compared to 60% and 80% for UOK120 and UOK146 lines, respectively (Fig. 1C). Thus, high levels of fusion protein were essential for tRCC cell viability irrespective of their epithelial/mesenchymal states.

### TFE3 fusions drive expression of genes involved in OxPhos

To identify genes regulated by TFE3 fusions, RNA-seq was performed on each cell line after siRNA-control (siCTR) or siRNA-TFE3 (siTFE3). Following silencing, the number of reads at the fusion junction and read coverage over TFE3 exons present in both NONO- and PRCC-TFE3 fusions were strongly decreased (Appendix Fig. S2A,B). Following TFE3-fusion protein silencing, we observed up- or down-regulation of hundreds of genes (range: ~200–500) in each tRCC cell line using a standard cut-off (log2 fold change > +/−1, and adjusted *p*-value < 0.05, Appendix Fig. S2B and Dataset EV1).

Gene Set Enrichment Analyses (GSEA) using the Hallmarks gene sets showed consistent down-regulation of the Oxidative Phosphorylation (OxPhos) pathway in all tRCC lines. In addition, we observed enrichment for down-regulation of oncogenic pathways in several cell lines such as MYC targets and pathways related to cell cycle and proliferation (E2F targets, G2M checkpoint; Fig. 1D; Appendix S2C). Limited pathways were enriched in the up-regulated genes (Fig. 1D). Among the 200 genes comprising the GSEA OxPhos signature (Dataset EV2), between 115 and 139 were down-regulated depending on the line, with 54 commonly down-regulated in all 4 lines (Fig. 1E). These genes encode multiple subunits of the mitochondrial respiration chain complexes I (*NDUF* (NADH: ubiquinone oxidoreductase), II (*SDHA/B*), III *UQC* (ubiquitol-cytochrome c oxidoreductase) and IV (*COX* cytochrome c-oxidase) (Fig. 1F). In addition, multiple subunits of

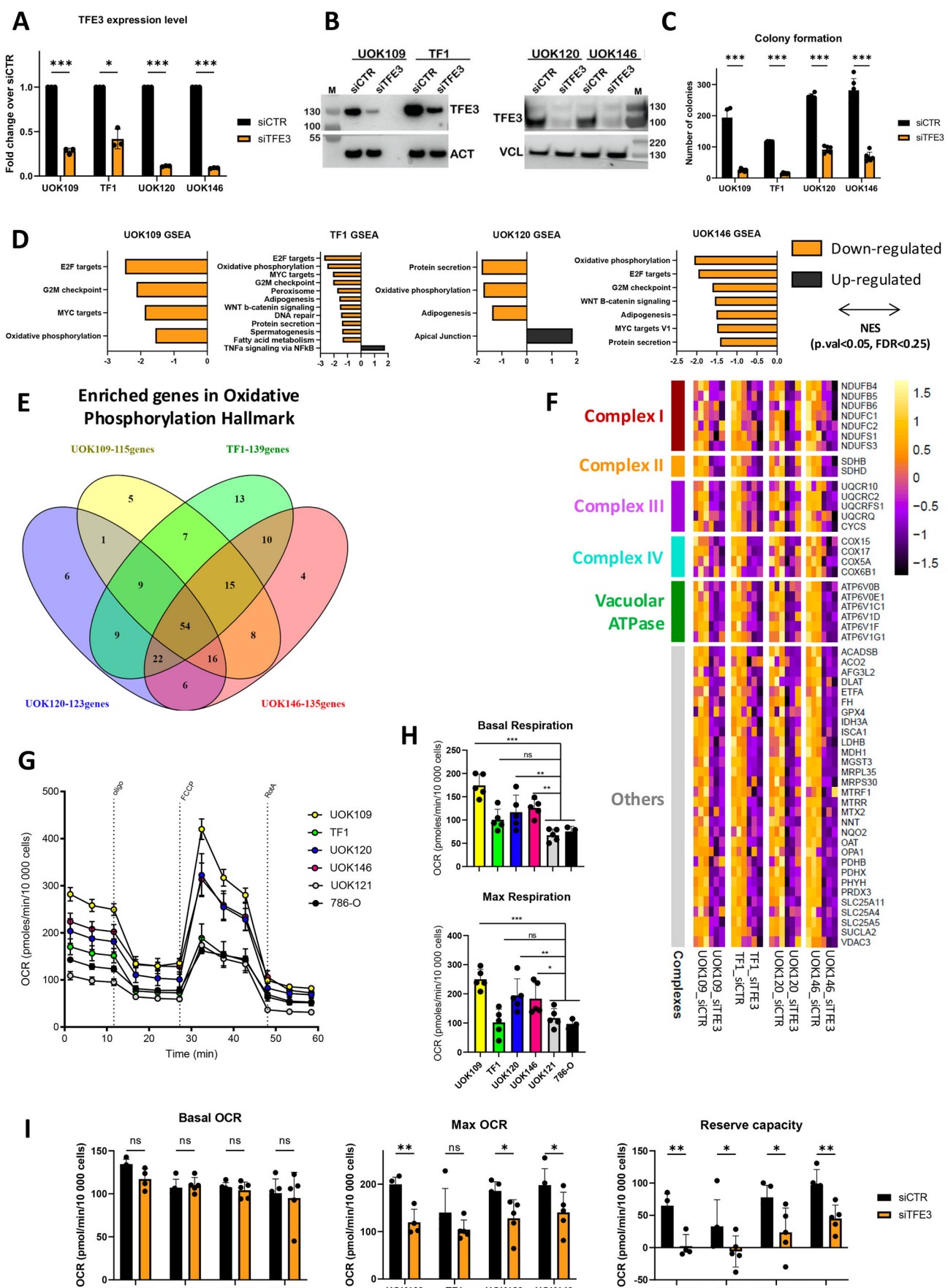

**Figure 1.   TFE3 fusion proteins regulate colony forming capacity and OxPhos gene expression.**

(A) TFE3-fusion protein expression was measured by RT-qPCR 72 h after siRNA transfection and expressed as fold change over siCTR normalized to housekeeping gene RPLP0 ($n = 3$). Conditions were compared by paired T-test (*: $p$.val < 0.05, ***: $p$.val < 0.001). Exact $p$-values UOK109 7.13E−04; TF1 1.18E−02; UOK120 1.79E−05; UOK146 8.88E−06. (B) Immunoblot of total protein extracts after siCTR or siTFE3 treatment. TFE3 blots for UOK109/TF1 and UOK120/UOK146 lines were exposed separately for 3 s and 10 s, respectively. However, to compensate the much higher expression of NONO-TFE3 compared to PRCC-TFE3, only 0.2 μg of extract was loaded. ACTB or VCL were used as a loading controls. (C) Cells were treated with siCTR or siTFE3 and 5000 cells seeded and cultured for 14 days, fixed and colored with crystal violet to assess colony forming ability ($n = 6$). Conditions were compared by paired T-test (***: $p$.val < 0.001). Exact $p$-values UOK109 6.62E−06; TF1 3.51E−08; UOK120 3.1E−06; UOK146 9.9E−06. (D) Gene Set Enrichment Analysis (GSEA) results comparing siCTR with siTFE3 using the Hallmark gene sets. Significantly deregulated pathways (weighted Kolmogorov–Smirnov test $p$.val<0.05, FDR < 0.25) are represented for each of the 4 lines. (E) Venn diagram showing overlap between significantly deregulated OxPhos genes in each cell line as defined by GSEA. (F) Heatmap showing expression of the 54-commonly down-regulated OxPhos genes as a Z-score. (G) Measure of OCR in tRCC (UOK109, TF1, UOK120, UOK146) and ccRCC (UOK121, 786-O) cell lines ($n = 5$). OCR values measured in tRCC and ccRCC lines were normalized by the number of cells in each well. (H) OCR values ($n = 5$) were used to measure the basal and the maximal respiration levels and compared these levels between each tRCC and the two ccRCC lines by one-way ANOVA (Dunnett's multiple comparison test, ns: non-significant – $p$.val > 0.05, *: $p$.val < 0.05, **: $p$.val < 0.01, ***: $p$.val < 0.001). Exact $p$-values basal respiration ccRCC vs UOK109 2.18E−07; TF1 1.8E−01; UOK120 6.7E−03; UOK146 1.27E−03. Exact $p$-values maximal respiration ccRCC vs UOK109 3.58E−05; TF1 9.9E−01; UOK120 8.64E−03; UOK146 2.4E−02. (I) Measurement of OCR in tRCC lines 72 h after treatment with siCTR or siTFE3 ($n = 5$). OCR values were used to measure the basal and the maximal respiration levels and reserve respiration capacity. siCTR and siTFE3 conditions were compared by paired T-test (ns: non-significant $p$.val > 0.05, *: $p$.val < 0.05, **: $p$.val < 0.01). Maximal respiration exact $p$-values UOK109 4.54E−03; TF1 7.02E−03; UOK120 1.32E−02; UOK146 2; 44E−02. Reserve capacity: exact $p$-values UOK109 1.42E−03; TF1 3.16E−02; UOK120 1.02E−02; U0K146 8.01E−03. All error bars indicate SD. Source data are available online for this figure.

the V-type proton ATPase, a complex responsible for acidification of intracellular organelles notably lysosomes, also comprised in the GSEA OxPhos pathway, were down-regulated by siTFE3. These observations suggested a direct role of TFE3 fusions in regulating expression of genes involved in OxPhos metabolism.

To evaluate whether these gene expression changes affected OxPhos, mitochondrial activity was measured by profiling the Oxygen Consumption Rate (OCR) using the Agilent SeaHorse (Fig. 1G). Comparison of OCR activity showed higher basal and maximum respiration capacities in 3 of the tRCC lines compared to the 786-O and UOK121 ccRCC lines, the exception being TF1 (Fig. 1H).

To determine the contribution of the TFE3 fusions to the elevated OxPhos capacity, SeaHorse experiments were performed in siTFE3 versus siCTR cells 72 h after transfection. Although basal OCR levels were unaffected, both maximum respiration and reserve capacity were significantly decreased after siTFE3 in UOK109, UOK146 and UOK120, with a tendency observed for TF1 line (Fig. 1I). These data showed that TFE3-fusions drive expression of genes involved in OxPhos resulting in a higher contribution of OxPhos to the metabolism of tRCC compared to ccRCC cell lines.

## Ectopic expression of TFE3 fusions leads to loss of renal identity and stimulates OxPhos

To determine whether TFE3 fusions can directly activate OxPhos, we performed gain of function experiments using HEK293T embryonic kidney cells. HEK293T cells were infected with lentiviral particles containing a Doxycycline (Dox)-inducible expression vector for either native TFE3 (T), NONO-TFE3 (NT), PRCC-TFE3 (PT), or GFP control. Twenty-four hours after Dox addition, a strong induction was seen at both the mRNA (Fig. 2A) and protein expression levels with NONO-TFE3 accumulating to lower levels than PRCC-TFE3 (Fig. 2B). Under these conditions, ectopic TFE3, NONO-TFE3, and PRCC-TFE3 proteins all accumulated in the nucleus (Fig. 2C).

To identify genes regulated by ectopic protein expression, we performed RNA-seq 24 h after Dox induction. Ectopic TFE3 expression deregulated over 1000 genes as compared to the GFP

control line using standard cut-off ($log2FC > +/−1$ and $adj.p$.val<0.05, Fig. 2D and Dataset EV3). Most genes deregulated by native TFE3 expression were also deregulated using fusion proteins (Figs. EV1A,B), whereas additional gene sets either commonly or specifically deregulated by both fusion proteins were observed. Overall, 558 genes were up-regulated in all TFE3-expressing lines and linked to functions previously ascribed to native TFE3 such as lysosome, mTORC1-associated pathway or transport of small molecules through ion channels (Fig. EV1A). Although fewer (71) common down-regulated genes were found, ontology showed they were linked to morphogenesis and development processes including kidney development suggesting that ectopic fusion protein expression may lead to loss of epithelial and kidney identity (Fig. EV1B).

While hallmark GSEA analysis showed few significant pathways for down-regulated genes, several pathways were enriched amongst up-regulated genes such mTORC1 signaling and immune signaling (TNFa signaling via NFkB, IFNg response; Fig. 2E). Importantly and in line with loss of function experiments, OxPhos was up-regulated in all conditions (Figs. 2E and EV1C). Among the 200 genes in the OxPhos pathway, between 84 to 119 genes were up-regulated with 59 genes common to all 3 lines including those encoding subunits of complexes of the electron transport chain and the lysosomal V-ATPase, but also other genes involved in mitochondrial homeostasis and function (Fig. 2F,G). In line with these observations, ectopic expression of TFE3, and to a greater extent TFE3 fusions enhanced basal and maximal OCR respiration as well as reserve capacity compared to the GFP control line (Fig. 2H,I). Thus, both loss and gain of function converged to show that TFE3 fusion proteins activated genes of the OxPhos pathway leading to enhanced OxPhos metabolism.

## TFE3 fusions bind promiscuously at active promoters and stimulate RNA synthesis

To better understand how the fusion proteins regulate gene expression, we performed Cut&Tag (C&T) in all 4 tRCC cell lines to profile TFE3 fusion protein binding along with H3K27ac ChIP-seq to identify active cis-regulatory elements. Read density analyses

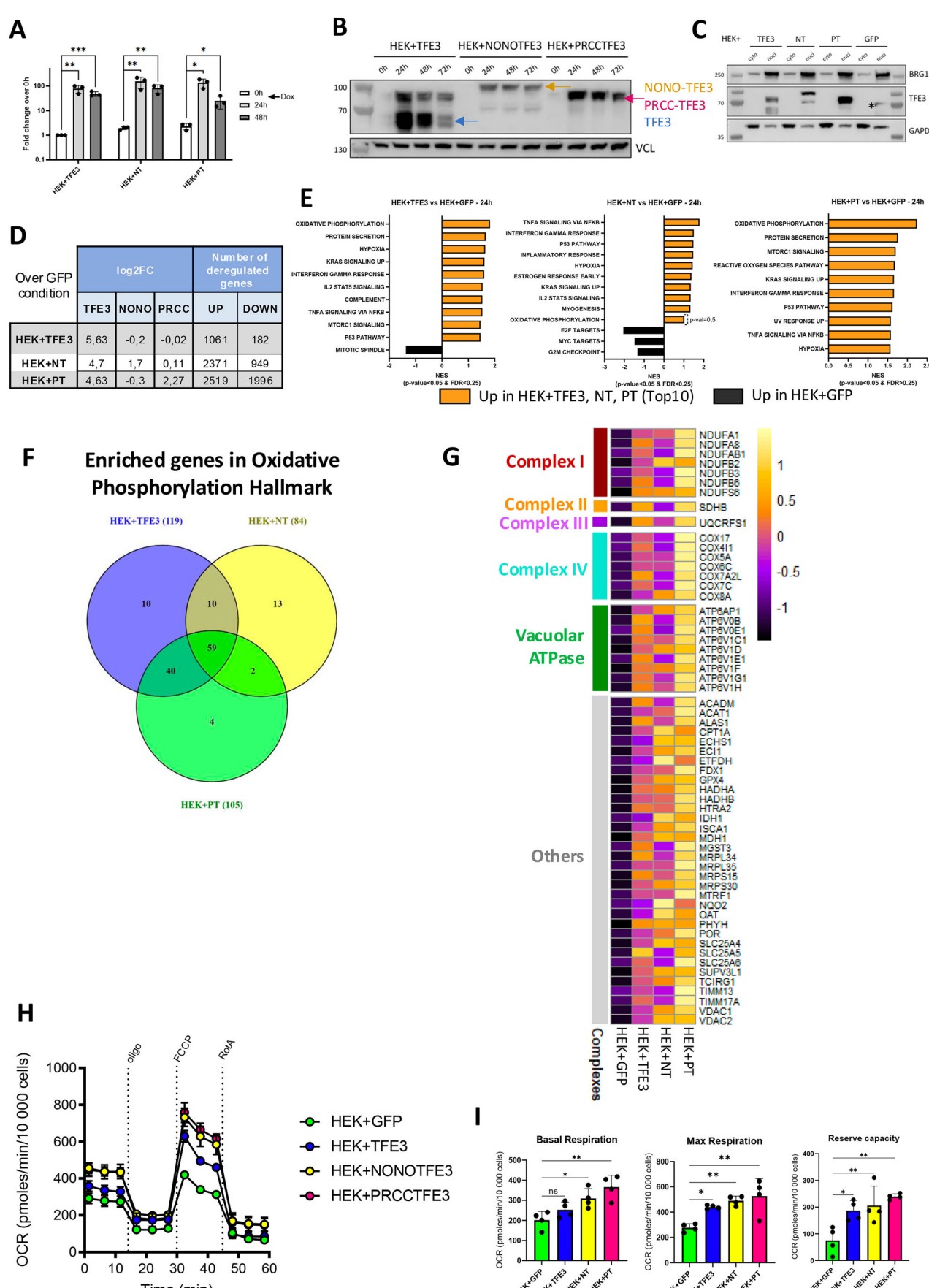

Figure 2. Enhanced OxPhos following ectopic expression of TFE3 fusions in HEK293T cells.

(A) Fold change of TFE3 expression compared to no-Dox treatment was measured by RT-qPCR at the indicated times after Dox addition ($n = 3$). Paired T-test, *: p.val < 0.05, **: p.val < 0.01, ***: p.val < 0.001. Exact p-values; 0 vs 24 h HEK-T 2.84E−03, HEK-NT 9.13E−03, HEK-PT 10.2E−02; 0 vs 48 h; HEK-T 9.48E−04; HEK-HT 6.67 E−03; HEK-PT 1.84E−02. (B) Immunoblots of total protein extracts after the indicated time of culture in presence of Dox. VCL was used as loading control. Blue arrow indicates native TFE3 protein, orange arrow NONO-TFE3 (NT), and pink arrow PRCC-TFE3 (PT). (C) Western blot analysis of cytoplasmic (cyto) and nuclear (nucl) protein extracts of indicated HEK293T cell lines. BRG1 was used as positive control for the nuclear extract, GAPDH for cytoplasmic extract. * Indicates a non-specific signal in the GFP control nuclear extract. (D) Log2FC values (over GFP) of TFE3, NONO, and PRCC in each line and number of genes deregulated 24 h after Dox addition (log2FC > ±1; adj p.val<0.05 over GFP) in each line ($n = 3$). (E) GSEA ontology analysis of differentially regulated genes in each cell line over GFP-line control. Significantly deregulated pathways (weighted Kolmogorov–Smirnov test p.val < 0.05, FDR < 0.25) are represented for each of the 3 lines, restricted to the top 10 pathways for HEK + PT line. (F) Venn diagram showing overlap between significantly up-regulated OxPhos genes according to GSEA. (G) Heatmap showing relative expression of each of the 59-commonly up-regulated OxPhos genes as a Z-score. (H, I) Measurement of the OCR values following ectopic TFE3, NT, or PT expression ($n = 4$). OCR values measured in each HEK line were normalized by the number of cells in each well and used to measure the basal and the maximal respiration levels and the reserve respiration capacity. Experimental conditions were compared over the GFP-condition by one-way ANOVA (Dunnett's multiple comparison test, ns: non-significant p.val > 0.05, *: p.val < 0.05, **: p.val < 0.01). Exact p-values basal respiration GFP vs HEK-T 3.4E−0; HEK-NT 2.06E−02; HEK-PT 1.09E−03. Exact p-values maximal respiration GFP vs HEK-T 2.2E−02; HEK-NT 3.88E−03; HEK-PT 1.3E−03. Exact p-values reserve capacity GFP vs HEK-T 1.72E−02; HEK-NT 6.36E−03; HEK-PT 1.15E−03. All error bars indicate SD. Source data are available online for this figure.

of the non-redundant TFE3 peaks from all lines revealed a large number of low-occupied sites that did not coincide with H3K27ac (Cluster 14 in Fig. EV2A). We therefore focussed our analyses on the 84,765 non-redundant sites strongly occupied in at least one of the 4 lines (Fig. 3A). Using this set of highly occupied sites, we found that the number of TFE3 peaks was around 2-fold higher (65,000–78,000) in both NONO-TFE3 expressing lines compared to PRCC-TFE3 (33,000–42,000) lines, whereas comparable numbers (44,000–64,000) of H3K27ac ChIP-seq peaks were observed in each cell line (Fig. EV2B).

MEME-ChIP-analyses of the top 1000 TFE3 peaks in the UOK120 and UOK146 lines revealed strong enrichment in the M/ E-box motif, that was the only enriched motif (Fig. EV2C). In contrast, analogous analyses of the UOK109 and TF1 TFE3 peaks revealed enrichment of GC-rich SP1/KLF-type motifs in addition to the M/E-box motif. Analyses of motif locations showed the M/E-box motif protected at the peak centre, whereas the other motifs were spread across the peaks. Similar results were found using RSAT software, where the M/E-box motif was highly represented in the UOK120 and UOK146 peaks, but was lower ranked in the other lines, where in addition the GC-rich SP1/KLF and NFY motifs were also detected (Fig. EV2D). Thus, the fusion proteins bind to their cognate M/E-box motif, but additional indirect binding was seen, in particular for NONO-TFE3.

The GC-rich SP1/KLF and NFY-binding CAAT-box motifs are prevalent in proximal promoters with the CATT-box located around 70 nucleotides upstream of the transcription start site (TSS) (Suske, 2017). Read density analyses showed that many of the 84,765 non-redundant were associated with H3K27ac either in a common or cell-specific manner (Fig. 3A) and consistent with the presence of SP1 and CATT-motifs, strong TFE3 fusion occupancy of ≈8700 active H3K27ac-marked promoters was seen (Fig. 3B). TFE3 fusion binding was also seen at 19,040 promoter distal regions (Fig. 3C) with a set of sites commonly occupied in all 4 lines (clusters 1–14), but showing differential association with H3K27ac (clusters 8–14). The remaining sites showed cell-specific occupancy such as cluster 15 preferentially occupied by NONO-TFE3 or clusters 16–19 corresponding to sites selectively occupied and/or marked by H3K27ac in a cell line-specific manner.

To consolidate these findings, we performed TFE3 C&T in the HEKT cells described above. As an additional control, these experiments were performed with an independent TFE3 antibody that was used in parallel in a replicate C&T in UOK109 cells. Comparison of the UOK109 data with the 2 TFE3 antibodies revealed high concordance of the strongly bound sites, with only the weak binding sites showing less signal with the second antibody (Appendix Fig. S3). In the HEK-GFP cells, binding of endogenous TFE3 was seen at only 1058 sites, with 615 at proximal promoters (+/−500 nt from TSS) associated with genes involved in lysosome and mTOR signaling as previously reported (Bakouny et al, 2022; Damayanti et al, 2018; Perera et al, 2019; Yin et al, 2019). Interestingly, OxPhos and autophagy (Settembre et al, 2011; Pastore et al, 2019) were also represented (Figs. EV3A,B). MEME-ChIP analyses of endogenous TFE3 sites revealed enrichment only of the M/E-box motif (Fig. EV3C). In contrast, 17–28-fold more sites were seen in HEKT cells expressing native TFE3 or PRCC-TFE3 fusions with NONO-TFE3 showing the highest number of peaks (Fig. EV3A). MEME-ChIP revealed enrichment of the M/E-box in the TFE3 and PRCC-TFE3 expressing lines together with the SP1/KLF motif, whereas in the NONO-TFE3 expressing cells, the SP1/KLF and NFY-binding CAAT-box motifs were the most enriched (Fig. EV3C). Again, the M/E-box was protected at the peak centre whereas the other motifs were spread across the peaks. While endogenous TFE3 was present at only 615 proximal promoters, the ectopically expressed proteins bound much more promiscuously at promoters, most notably NONO-TFE3 (Fig. EV3A). Thus, high nuclear accumulation of native TFE3, PRCC-TFE3 and, in particular of NONO-TFE3, in HEKT cells led to promiscuous promoter binding.

The broad genomic binding seemed inconsistent with the limited effects on gene expression seen by RNA-seq upon fusion protein silencing. It was previously reported that MYC can bind promiscuously to active promoters and stimulate mRNA synthesis (Lin et al, 2012; Nie et al, 2012). As such an effect cannot be readily detected by standard RNA-seq, we performed EU incorporation to determine if TFE3 fusion silencing reduced nascent mRNA synthesis. In each cell line, siTFE3 reduced EU incorporation compared to the siCTR and hence reduced levels of mRNA synthesis (Fig. 3D). The reduction was most prominent in the NONO-TFE3-expressing lines in particular an almost 45% reduction in UOK109 compared to the milder (15–20%) reduction in PRCC-TFE3 expressing lines. Together with the C&T results, these data indicated that TFE3 fusion proteins can both activate specific

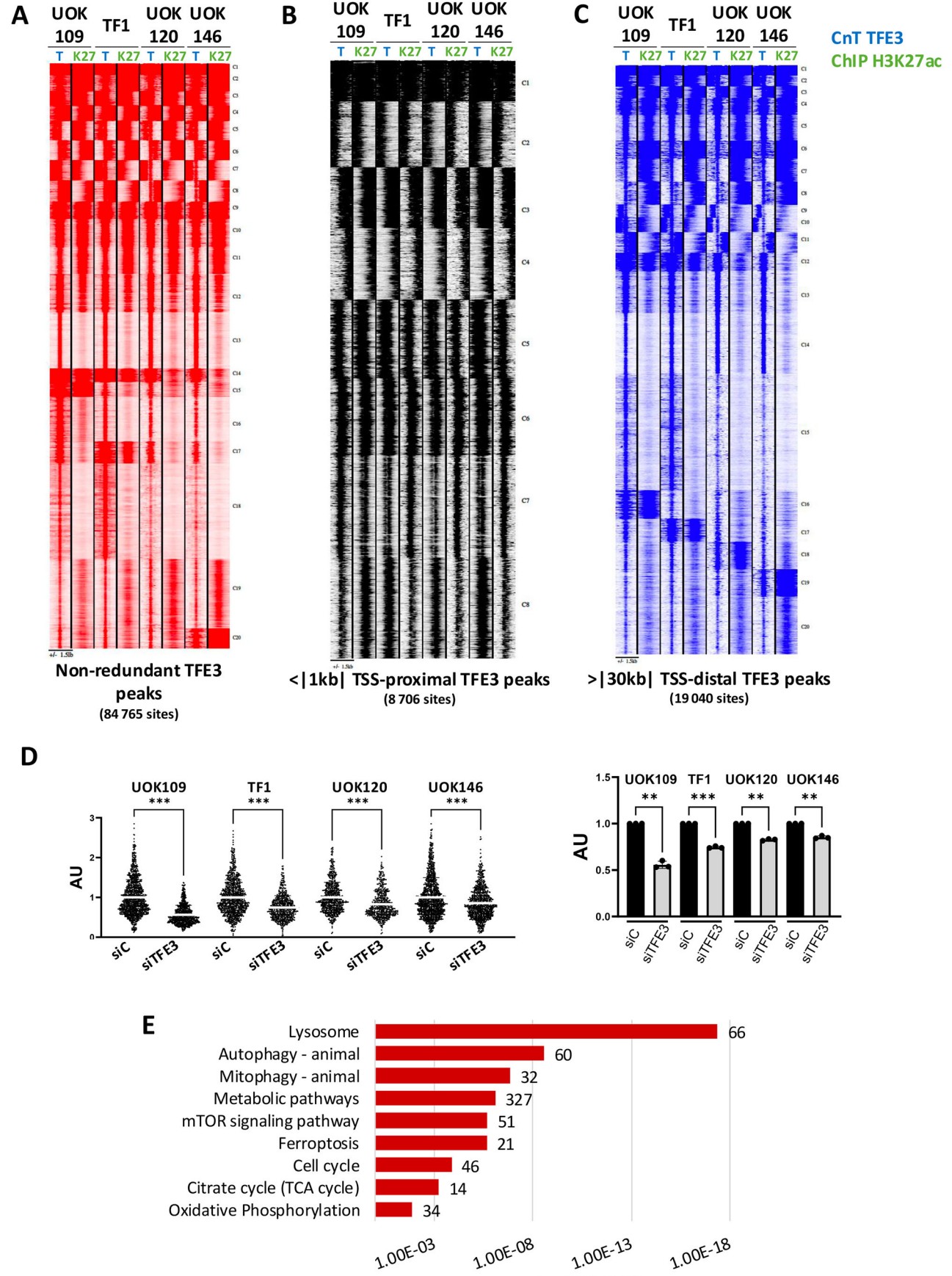

**A** Non-redundant TFE3 peaks (84 765 sites)

**B** <|1kb| TSS-proximal TFE3 peaks (8 706 sites)

**C** >|30kb| TSS-distal TFE3 peaks (19 040 sites)

CnT TFE3
ChIP H3K27ac

**E**

| Pathway | count |
|---|---|
| Lysosome | 66 |
| Autophagy - animal | 60 |
| Mitophagy - animal | 32 |
| Metabolic pathways | 327 |
| mTOR signaling pathway | 51 |
| Ferroptosis | 21 |
| Cell cycle | 46 |
| Citrate cycle (TCA cycle) | 14 |
| Oxidative Phosphorylation | 34 |

p-value (FDR<0.05)

◄ **Figure 3.  Profiling genome occupancy of TFE3 fusion proteins.**

(A) Read density maps for TFE3 fusion protein occupancy (T) and H3K27ac (K27) at the 84,765 non-redundant sites. (B, C) Read density maps for the non-redundant TFE3 fusion protein and H3K27ac sites at proximal promoters (<1 kb from the TSS, B) or putative distal enhancers (>30 kb from the TSS, C). (D) Level of EU-incorporation in each tRCC line 48 h after siCTR or siTFE3. The upper panel shows the combined quantification of the relative EU intensity signal in the nucleus for each cell line from the 3 independent experiments normalized by the mean intensity of the control condition. The left panel shows the pooled intensity signals in arbitrary units for the 3 independent biological replicates with a Fisher LSD ANOVA one-way test, all p-values < 0.001 (***): UOK109; UOK120, UOK146, p-value < 1.0E−15; TF1, 2.14E−13. The right panel shows the relative incorporation value for each line with the siControl value set as 1. Exact p-values UOK109 0.003 (**), TF1 < 0.001 (***) UOK120 0.001 (**) UOK146 0.004 (**). (E) KEGG ontology of genes associated with TFE3 fusion protein occupancy of M/E-box containing sites +/− 500 bp of the TSS. Each indicated KEGG pathway presented an FDR < 0.05 and an associated p-value calculated by hypergeometric distribution and Benjamini-Hochberg correction with the number of genes found in each pathway noted to the right of the bar. All error bars indicate SD. Source data are available online for this figure.

gene expression programs and, via binding to active promoters, more generally stimulate levels of mRNA synthesis.

## TFE3 fusions directly regulate OxPhos genes

To define the core program of TFE3-fusion driven genes, we identified ≈12,534 sites commonly bound in all 4 tRCC cell lines comprising an M/E-box of which 3900 were located in the proximal promoter (+/−500 nucleotides from TSS, Dataset EV4). KEGG ontology of the 3313 associated protein-coding genes showed enrichment in previously defined TFE3 functions (lysosome, mTOR pathway, autophagy and mitophagy) but also cell cycle and metabolic pathways including Oxphos and ferroptosis (Fig. 3E). Of the 235 OxPhos signature genes (comprising GSEA-OxPhos and KEGG-OxPhos genes Dataset EV2), 145 were associated with the presence of TFE3-bound sites comprising an M/E-box within a distance of +/−30 kB of the TSS marked by H3K27ac (Fig. EV3D). Among the 54 OxPhos genes down-regulated by siTFE3 in all 4 cell lines, 33 displayed a TFE3 bound site, while of the 59 genes up-regulated in the 3 modified HEK293T cells, 44 displayed TFE3 binding (Fig. EV3D).

Examples of TFE3 occupancy of M/E-box-containing sites at genes of mitochondrial electron transport chain complexes and the lysosomal V-ATPase in the tRCC lines are shown in Appendix Figs. S4A–C. Many of these genes were also bound by ectopic native TFE3 and the NONO-and PRCC-TFE3 fusions in HEKT cells along with endogenous TFE3 binding in the GFP line. For example, the COX5A and ATP6V0E1 loci displayed fusion protein binding to the same sites in the tRCC and HEKT cells (Appendix Fig. S4B,C). Similarly, *PPARGC1A* (PGC1A), a master regulator of mitochondrial biogenesis regulated by MITF in melanoma (Vazquez et al, 2013), also displayed multiple TFE3 fusion binding sites at the promoter and putative upstream and downstream regulatory elements in the tRCC lines (Appendix Fig. S4C). In contrast, no endogenous TFE3 was seen at the PPARGC1A locus, and only a subset of the sites seen in the tRCC lines were bound in HEKT cells expressing ectopic TFE3.

## Human tRCC tumors retain elevated OxPhos gene expression during EMT

To investigate whether the elevated TFE3-driven OxPhos in the 4 tRCC cell lines was also seen in human patient samples, we integrated clinical and RNA-seq data of 37 TFE3-tRCC primary samples and 2 metastasis samples (from the same patient, therefore 38 patients and 39 samples) from a cohort of patients from American and French institutes and TCGA whose clinical and

pathological features are summarized in Fig. EV4A,B and Dataset EV5. This cohort comprised 66% (n = 25) females and 34% (n = 13) males (ratio 1.92:1 female:male), with a median age of 34 years (range: 3–65), amongst which 8 were under 18. All stages were broadly represented with 53% cases presenting stage I–II and 47% stage III-IV, with at least 10 cases already spread in lymph nodes and 1 case was poly-metastatic with two different metastatic samples collected (Dataset EV5). *SFPQ* (n = 12), *PRCC* (n = 12), *ASPSCR1* (n = 5) were the most frequent fusion partners. The cohort also comprised the p34 subunit of general transcription factor TFIIH (GTF2H3) as a novel fusion partner of TFE3 (Fig. EV4C). At the end of follow-up, 14 (36%) patients died with median overall survival of 47 months (range: 8.3–231.1 months).

To evaluate general differences in the gene expression signatures between tRCC and NAT, we normalized the bulk RNA-seq of 18 normal adjacent tissue (NAT) from the TCGA KIRC and the 39 TFE3-tRCC samples. 2856 genes were up-regulated in tumor samples and 2146 in NAT (Fig. EV4D,E). Ontology analysis of genes up-regulated in tumors revealed enrichment in cell proliferation (regulation of cell activation, ribosome, mitotic cell cycle) and immune pathways (regulation of immune effector process, positive regulation of immune response and cytokine production). In contrast, genes up-regulated in NAT were enriched in kidney structure and function (inorganic ion transmembrane transport, kidney development) and several metabolic pathways (organic hydroxy compound metabolic process, monocarboxylic acid metabolic process). Hence, transformation was associated with loss of epithelial and kidney identity and function, but gain of proliferation and immune markers.

To refine these analyses, we performed unsupervised clustering of gene expression identifying 3 main clusters, each enriched in specific fusion partners (Fig. 4A). Cluster 1 was enriched in PRCC-TFE3 fusions (8/12 cases, p.val = 2.4E−3) with 3 of the remaining 4 in cluster 3. Cluster 2 was enriched in SFPQ- or NONO-TFE3 fusions (9/14, p.val = 3.7E−4) with 4 of the remaining 5 in cluster 3 (Fig. 4B). SFPQ and NONO cases were combined as they belong to the same RNA-binding protein family and are known to form a protein complex (Knott et al, 2016). Cluster 3 encompassed all 5 ASPSCR1-TFE3 fusions (enrichment p.val = 2.4E−3), but comprised a majority (10 of 16) of non-ASPSCR1 fusions. Hallmark GSEA analyses of genes differentially expressed between clusters revealed the mesenchymal characteristics and hot-immune profile of Cluster 3 (Fig. 4B). Conversely, clusters 1–2 were more epithelial with enrichment in OxPhos and other metabolic pathways.

The immune profiles were consolidated by MCP-counter analysis showing clusters 1 and 2 to be mostly composed of cold tumors (22/23), whereas cluster 3 displayed immune infiltration

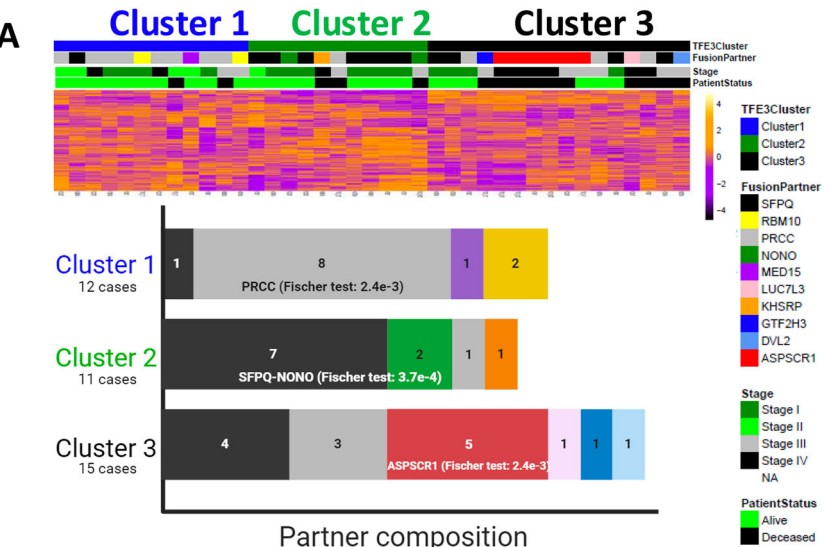

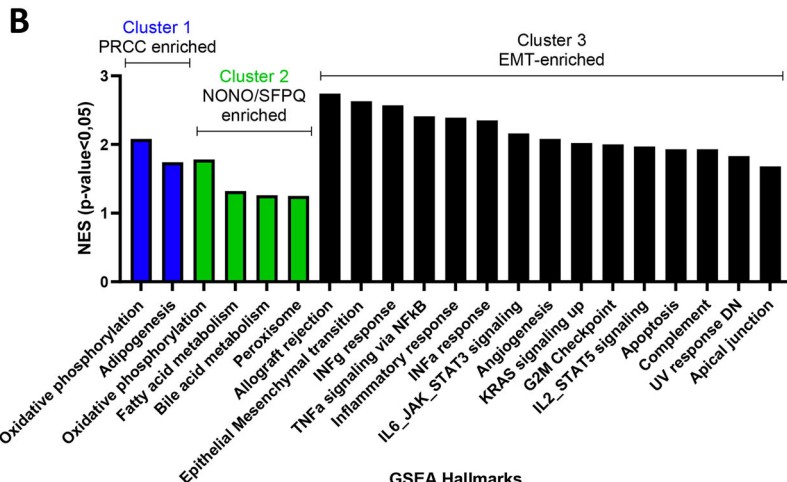

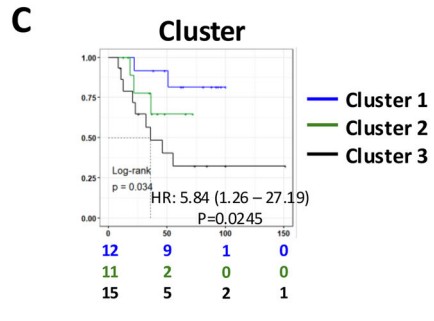

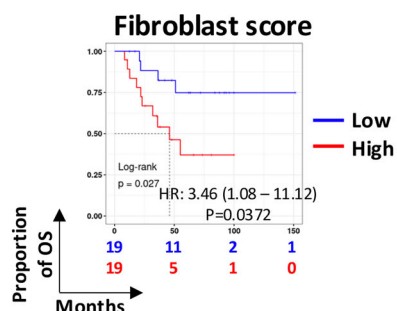

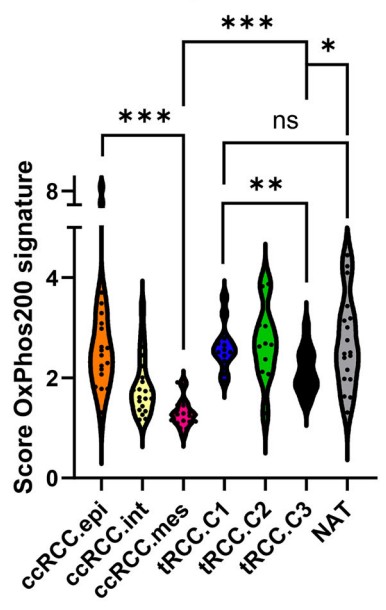

◄

**Figure 4. EMT and OxPhos in human tRCC.**

(A) Heatmap of the unsupervised clustering of gene expression with the indicated clinical and molecular parameters, showing the distribution of tRCC in 3 clusters. Distribution of tRCC samples according to fusion partner type within the 3 major clusters; p-values were indicated for most frequently represented partner in each cluster. (B) GSEA ontology analyses of differentially expressed genes between the different clusters indicating the enriched terms (p-value < 0.05, FDR < 0.25) for each cluster (with only the top 15 for Cluster 3). (C) Kaplan–Meier curves for overall survival in tRCC patients according to classification in the three clusters or fibroblast score, calculated using MCP counter, according to the optimal cut point method with the associated log-rank p-value. Hazard ratio (HR) was calculated using univariate Cox proportional-hazards model. Cluster scores HR 5.84 2.45E−02, fibroblast score HR 3.46 p-value (from Wald test) 7.72E−02. (D) Expression score of the 200-signature genes of GSEA-OxPhos Hallmark in NAT, each tRCC cluster and the indicated ccRCC tumor subtypes from the TCGA KIRC collection. Overall scores were compared by Wilcoxon test and are summarized in Dataset EV6 (**: p.val < 0.01, ***: p.val < 0.001). Exact p-values; ccRCC.epi vs ccRCC.mes 2.71E−09; ccRCC.mes vs tRCC.C3 3.72E −08; tRCC.C1 vs tRCC.C3 4.17E−03; tRCC.C3 vs NAT 2.31E−02; tRCC.C1 vs NAT 8.81E−01. CcRCC.epi, ccRCC.int, ccRCC.mes. $N = 20$: tRCC.C1 $n = 12$; tRCC.C2 $n = 11$; tRCC.C3 $n = 15$; NAT $n = 18$. Source data are available online for this figure.

(Appendix Fig. S5A). Similarly, use of a signature gene set confirmed the EMT gradient between clusters 1–3 (Appendix Fig. S5B). Combining these analyses further revealed the higher fibroblast signature score of mesenchymal tumors. Correlation with patient survival showed that cluster 3 (HR = 5.84, Log-rank p.val = 0.034) and fibroblast score (HR = 3.46, Log-rank p.val = 0.027) were strongly associated with poor survival (Fig. 4C).

We next examined the OxPhos signature score using the 200 genes of the GSEA-OxPhos Hallmark pathway (Dataset EV2). Clusters 1 and 2 displayed OxPhos scores comparable to the NAT, but were significantly higher than Cluster 3 (Fig. 4D and Dataset EV6). We also assessed the OxPhos scores of tumors carrying the PRCC-TFE3 or NONO-TFE3 fusions compared to the other fusions in each cluster finding no significant differences (Dataset EV6). In cluster 3 for example, ASPSCR1-TFE3 fusions have similar OxPhos scores to the others fusions of this cluster. However, OxPhos scores were reduced in the tumors having undergone EMT. These data support the idea that the TFE3 fusions had similar abilities to activate the OxPhos program and were similarly affected by EMT.

We additionally compared the tRCC OxPhos scores to those of ccRCC. So as not to compare stratified tRCC patients with non-stratified ccRCC patients that represent a heterogenous collection of tumors with different EMT and metabolic states, we stratified ccRCC tumors based on our previously reported single-cell RNA-seq signatures that defined an EMT gradient associated with poor survival and an OxPhos to glycolysis switch (Davidson et al, 2023). Based on deconvolution of the TCGA-KIRC collection with the ccRCC tumor cell signatures, we selected 20 tumors of the ccRCC.epi (epithelial), ccRCC.int (intermediate) and ccRCC.mes (mesenchymal) states and scored their OxPhos signatures (Fig. 4D). Comparison with the tRCC clusters showed that although ccRCC.epi had a comparable OxPhos score to tRCC clusters 1 and 2 and to NAT, all tRCC clusters showed significantly higher OxPhos score than ccRCC.int and ccRCC.mes (Fig. 4D and Dataset EV6). Cluster 3 in particular showed a higher OxPhos score than ccRCC.mes. Thus, while OxPhos score strongly declined upon EMT in ccRCC, it remained elevated during EMT in tRCC consistent with the ability of TFE3 fusions to drive OxPhos gene expression.

## EMT in tRCC tumors associates with presence of myCAFs and poor outcome

We investigated whether we could use the ccRCC tumor cell signatures to identify cell populations in tRCC. CibersortX

deconvolution of the tRCC cohort showed that the intermediate ccRCC.int signature was present in almost all tumors (Fig. 5A,B). In contrast, cluster 3 tumors showed enrichment in ccRCC.mes, whereas ccRCC.epi was associated with clusters 1 and 2. Deconvolution with the ccRCC myofibroblast (my)CAF and antigen-presenting (ap)CAF signatures revealed enrichment of myCAFs in cluster 3 and hence strong correlation with ccRCC.mes, whereas apCAFs were more broadly associated with tumors of each cluster (Fig. 5A–C). Both ccRCC.mes (HR 5.06 Log-rank p = 0.005) and myCAFs (HR 4.25, Log-rank p = 0.005) were strongly associated with poor tRCC outcome (Fig. 5D).

We confirmed these findings in an independent tRCC cohort (Sun et al, 2021) whose composition and clinical characteristics were described (Fig. EV5A). Unsupervised clustering defined 3 main clusters, each enriched in specific fusion partners (Fig. EV5B) with cluster 3 comprising the 9 ASPSCR1-TFE3 fusions and displaying analogous features to cluster 3 of the above-described cohort (Fig. EV5C). Cluster 3 showed strongest association with poor survival as did the fibroblast score (Fig. EV5D). Deconvolution with the ccRCC tumor cell signatures again revealed the strong myCAF-mes association and both cell populations strongly associated with poor survival (Figs. EV5E,F). Furthermore, scoring their OxPhos signatures again revealed that the tRCC EMT cluster 3 retained a higher OxPhos score compared to ccRCC.mes tumors (Fig. EV5G).

The data from these 2 cohorts showed that tRCC tumors displayed a high OxPhos score that was maintained during EMT. Moreover, tRCC EMT was associated with enrichment in mesenchymal cancer cells and myCAFs that both strongly associated with poor survival.

Together, our data suggest that TFE3 tRCC represent a spectrum of epithelial to mesenchymal tumors, with high expression of oxidative phosphorylation driven directly by a TFE3 core program.

## TFE3 fusions directly regulate ferroptosis genes

We noted that ferroptosis was represented in the core program of TFE3-fusion bound genes (Fig. 3E). In contrast, the ferroptosis pathway did not stand out in the ontology analyses of the patient transcriptomic data (Fig. EV4D,E). To address this, we defined two ferroptosis signatures encompassing anti-ferroptosis genes (signature of 57 genes) and comprising those involved in glutathione metabolism, GCLC; GCLM, the $X_c$ system, or genes (signature of 38 genes) that promote ferroptosis, such as ASCL1-6 that catalyze beta-oxidation of lipids or TFRC the transporter that imports iron into the cell to promote ferroptosis (Fig. 6A; Appendix Fig. S6A). Assessing their expression in NAT compared to tRCC clusters

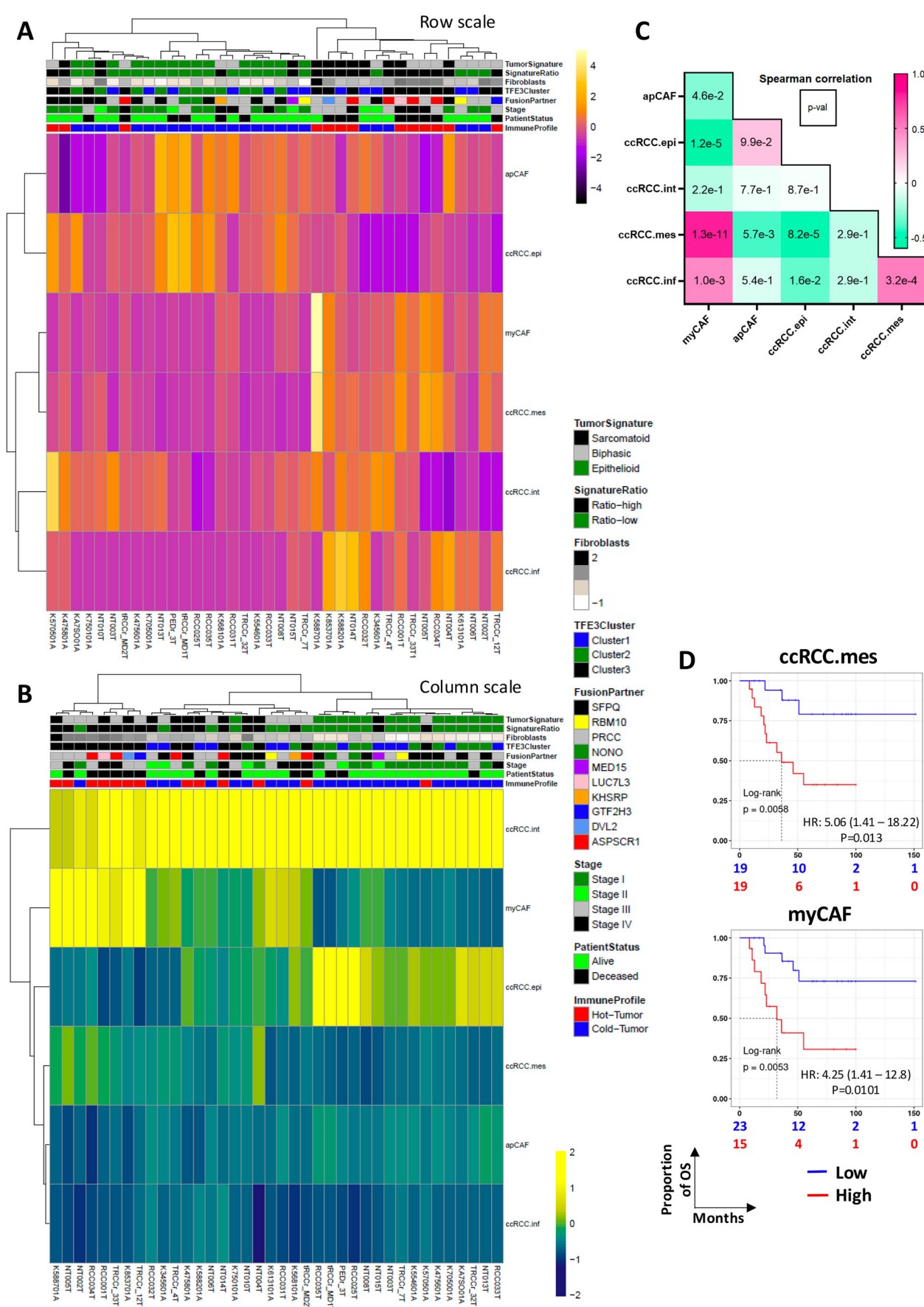

◀ **Figure 5. Deconvolution of tRCC tumors using ccRCC tumor cell and CAF signatures.**

(A, B) Heatmaps showing deconvolution using the ccRCC tumor cell and CAF signatures inferred by CIBERSORTx and displayed as row-scaled (**A**) or column-scaled (**B**) absolute scores on bulk RNA-seq data from the tRCC tumor samples. (**C**) Spearman correlation coefficient (colored box) and associated *p*-value (number in box) between the indicated populations. (**D**) Kaplan–Meier curves for overall survival according to the ccRCC.mes and myCAF proportions using optimal cut-off value method with the associated log-rank *p*-value and the HR from the univariate Cox proportional-hazards model. MES score HR 5.06 *p*-value (from Wald test) 1.3E−02, CAF score HR 4.25 *p*-value 1.01E−02. Source data are available online for this figure.

revealed that many key genes involved in ferroptosis resistance were up-regulated in tRCC compared to NAT (GPX4, GCLM, GCLC, SLC73A2, SLC7A11, TXNRD1), whereas those involved in promoting ferroptosis (TFRC, ASCL3, 4 and 6) were repressed (Fig. 6A; Appendix Fig. S6A).

In line with the above, all 4 tRCC lines showed low IC50 values for the GPX4 inhibitor RSL3 between 2.6 nM for UOK109, and 58 nM for TF1 (Fig. 6B). These values were similar to that of A375 melanoma cells known to be ferroptosis sensitive and much lower than for 501Mel melanoma cells (IC50 > 2 mM, Appendix Fig. S6B) reported to be ferroptosis resistant (Dias et al, 2024). RSL3-induced cell death was efficiently rescued in the presence of the ferroptosis inhibitor ferrostatin-1. Hence, tRCC tumorigenesis was accompanied by a switch from expression of mediators of ferroptosis to increased expression of ferroptosis resistance genes, with tRCC cells being highly sensitive to ferroptosis upon inhibition of GPX4.

These observations were partially reproduced in Dox-treated HEKT cells, in particular with PRCC-TFE3 that is best expressed in this model system, where ferroptosis resistance genes GPX4, GCLC, SLC73A2, SLC7A11, TXNRD1 were activated, whereas ASCL4 and TFRC were repressed (Fig. 6C; Appendix Fig S6C). The loci of GPX4 and GCLC showed prominent binding of fusion proteins in the tRCC lines and Dox-treated HEKT cells (Appendix Fig. S7A,B and Dataset EV3). Moreover, the GPX4 locus displayed a broad H3K27ac signal reminiscent of a super-enhancer in all 4 tRCC lines indicating that it may be a critical gene in these cells.

To investigate TFE3-fusion regulation of glutathione metabolism in HEKT cells, we measured the levels of reduced (GSH) and oxidized glutathione (GSSG) following Dox induction. While DMSO and Dox-treated control GFP-HEKT cells have low intrinsic GSSG levels and hence a low ratio of GSSG/GSH, Dox-induced expression of native TFE3 or the fusion proteins significantly increased GSSG levels in a RLS3-repressible manner showing activation of GPX4 activity (Fig. 6D; Appendix Fig. S8A). In contrast, siTFE3 in the tRCC cells led to reduced levels of GSSG (Fig. 6E). The reduction was particularly marked in the NONO-TFE3 expressing lines that had overall lower intrinsic GSSG levels of compared to the PRCC-TFE3 lines with higher levels (Appendix Fig. S8B,C). These observations strongly support the idea that TFE3 fusions bind to and directly activate key genes involved in ferroptosis resistance and glutathione metabolism.

## Discussion

### TFE3-fusion driven gene expression

Here, we profiled genomic binding of the NONO-TFE3 and PRCC-TFE3 fusions revealing their broad binding over the genome with more than 12,000 M/E-box-containing sites occupied in all 4 tRCC

lines, of which 3900 were located in proximal promoters of genes associated with functions previously found to be regulated by native TFE3, such as lysosome and mTOR pathway, but also novel pathways pertinent to oncogenic transformation such as cell cycle, metabolism and ferroptosis. Previous studies reported only small numbers of TFE3 fusion binding sites [around 3032 sites in UOK146 and SFPQ-TFE3 xenograft lines (Damayanti et al, 2018), 4129 in UOK109 and 1584 in UOK120 (Yin et al, 2019)] and they provided no information on association with active H3K27ac-marked regulatory elements. Nevertheless, while this study was under review Li et al (2025) reported profiling of NONO-TFE3 and ASPSCR1-TFE3 together with analyses of H3K27ac-marked regulatory elements. In line with our observations, this study also revealed direct regulation of the OxPhos program by the TFE3 fusions, but did not observe the broader binding of these fusions. In contrast, Sicinska et al (Sicinska et al, 2024) performed HA-ChIP-seq in alveolar soft part sarcoma cells ectopically expressing HA-tagged ASPSCR1-TFE3 finding that it binds broadly to active chromatin sites and is enriched at proximal promoters. While it could have been argued that this was a consequence of ectopic overexpression or a specific property of the ASPSCR1-TFE3 fusion associated with more aggressive tumors (Prakasam et al, 2024; Sun et al, 2021), they are in agreement with our observations on endogenous NONO-TFE3 and PRCC-TFE3 proteins in tRCC cells. Hence promiscuous promoter binding is a common property of TFE3 fusion proteins shared in both tRCC and alveolar soft part sarcoma.

Several mechanisms may account for the ability of TFE3 fusions to associate with active promoters. The NONO-SFPQ complex was previously shown to interact with the C-terminal domain (CTD) of the largest subunit of RNA polymerase II (Pol II) and bind to proximal promoters (Lim et al, 2020; Emili et al, 2002; Kameoka et al, 2004). NONO-TFE3 may therefore be recruited in an analogous manner via the Pol II CTD. TFE3 fusion partners are almost invariably proteins involved in RNA splicing or metabolism, many of which assemble on the CTD to coordinate transcription and splicing (Saldi et al, 2016; Custódio and Carmo-Fonseca, 2016; Shenasa and Bentley, 2023). For example, the TFE3 partner ZC3H4 involved in restriction of non-coding transcription (Hughes et al, 2023; Estell et al, 2023; Rouvière et al, 2023) was shown to be recruited to active promoters. Moreover, TFE3 can be fused with Mediator subunit MED15 or the p34 subunit (GTF2H3) of TFIIH, general transcription factors integral to the Pol II preinitiation complex.

Recruitment of TFE3 fusions at proximal promoters may impact both transcription and splicing. We found that TFE3 fusions stimulated RNA synthesis with the strongest effects were seen in UOK109 and TF1 consistent with the propensity of NONO-TFE3 to bind active promoters. General stimulation of RNA synthesis has been reported as an oncogenic mechanism for overexpressed MYC

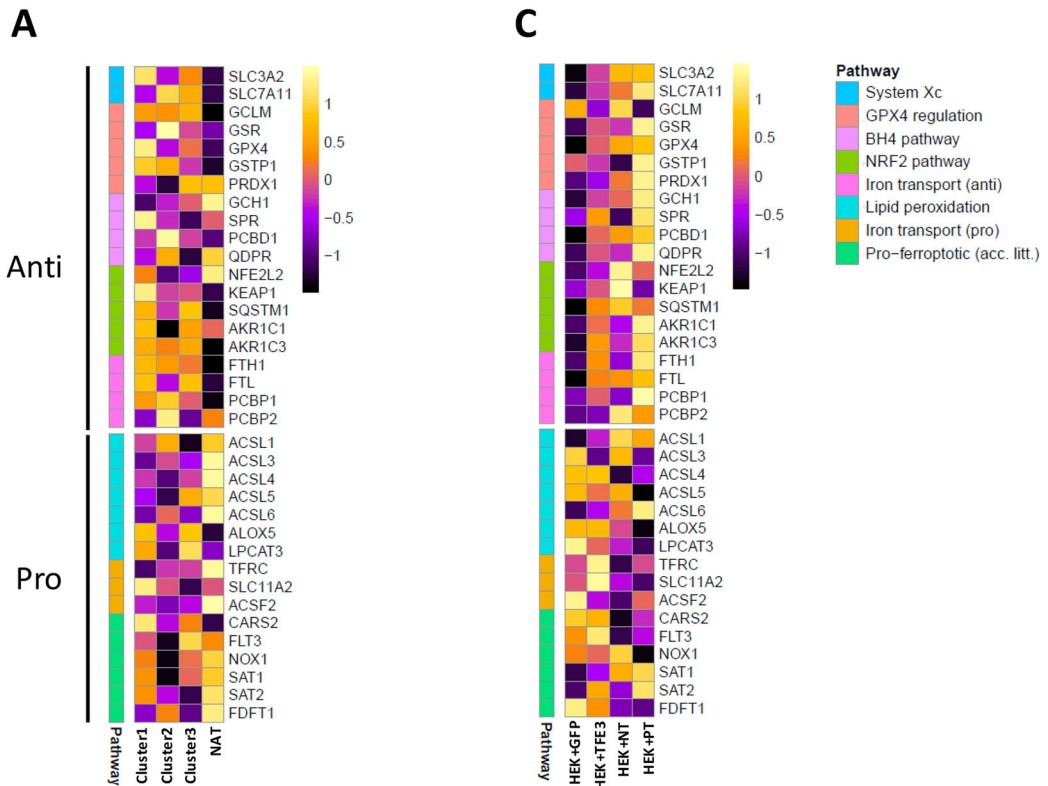

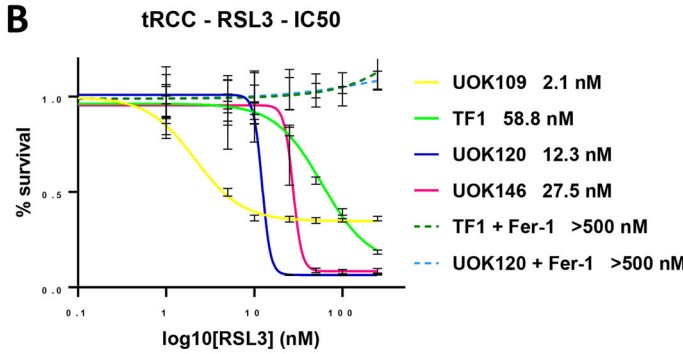

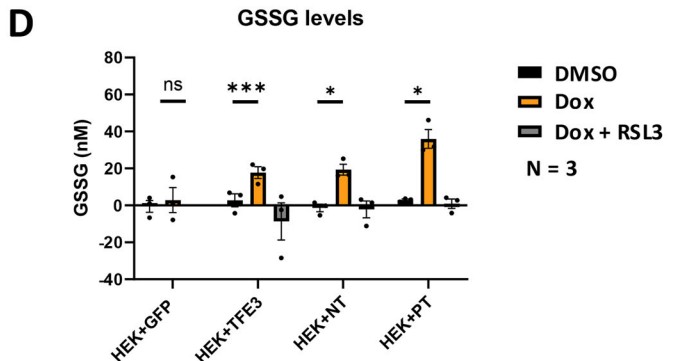

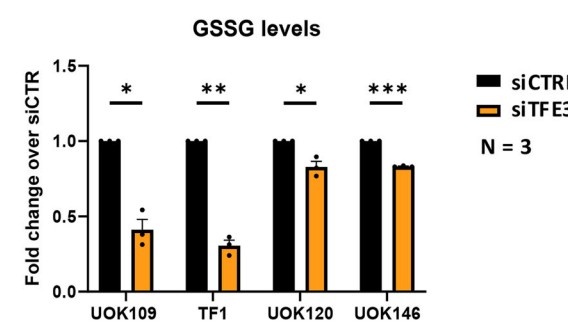

◄ **Figure 6.** **TFE3-fusion protein-driven ferroptosis resistance.**

(A) Heatmap of expression of the selected key ferroptosis genes in the NAT compared to patient tRCC clusters 1–3. (B) RSL3 IC50 values of tRCC lines in presence or absence of 1 μM ferrostatin-1 (Fer-1) as indicated. (C) Heatmap of expression of the selected key ferroptosis genes in the HEKT cells 24 h after Dox treatment. (D) GSSG levels in the indicated HEK cells 24 h following DMSO or Dox treatment. Paired T-test; exact $p$-values HEK GFP 6.38 E−01; HEK-T 9.66 E−04, HEK-NT 2.23E−02; HEK-PT 2.76 E−02. (E) GSSG levels in the indicated tRCC cells 48 h following siCTR or siTFE3. Values in the siC cells were normalized as 1 and the siTFE3 shown as ratio compared to control. Paired T-test; *: $p.val < 0.05$, **: $p.val < 0.01$, ***: $p.val < 0.001$, NS non-significant. Exact $p$-values UOK109 1.32E−02; TF1 2.63E−03; UOK120 4.33 E−02; UOK146 2.58E−04. All error bars indicate SEM. Source data are available online for this figure.

(Lin et al, 2012; Nie et al, 2012) and oncogenic RAS (Kotsantis et al, 2016; Johnson et al, 2003; Johnson et al, 2000). Hence, TFE3 fusions may act in a bipartite manner, activating specific genes via binding to M/E-box containing regulatory elements and more generally stimulating mRNA synthesis via general promoter binding. Furthermore, Damayanti (Damayanti et al, 2024) reported that TFE3 fusion with NONO or PRCC promoted their nuclear trafficking in paraspeckle domains and altered the global splicing landscape. Hence, promiscuous binding at proximal promoters via M/E-boxes, indirect tethering, or interaction with the Pol II CTD allows TFE3 fusion proteins to impact global RNA synthesis and/or splicing.

In HEKT cells, ectopically expressed native TFE3 and the NONO- and PRCC-TFE3 fusion proteins accumulated in the nucleus. The NONO-TFE3 fusion displayed broadest binding in both tRCC and HEK293T cells with enrichment for the E/M-box, but also promoter-enriched SP1/KLF and NFY motifs, unlike PRCC-TFE3 and native TFE3 that bound principally, but not exclusively, via M/E-box motifs. While, this may be explained by the much higher nuclear accumulation of NONO-TFE3 in the tRCC lines, this is not the case in the HEKT cells, rather suggesting that the higher propensity for indirect promoter binding is an intrinsic property of the NONO-TFE3 fusion. Deep learning-based in silico modeling (Damayanti et al, 2024) suggested new protein structures and changes in conformation of TFE3 fusion proteins with increased potential for binding DNA, RNA, and protein. Moreover, the conformation of the TFE3 bHLH-LZ domain may be altered in fusion proteins, perhaps in part explaining TFE3-fusion binding to M/E-box motifs and indirect tethering via other transcription factors at the proximal promoter. Intriguingly, Damayanti et al, proposed that the bHLH-LZ domain in PRCC-TFE3 adopted a conformation close to that of native TFE3. This is concordant with the predominant PRCC-TFE3 binding via M/E-box motifs, similar to overexpressed native TFE3, whereas NONO-TFE3 showed enhanced indirect binding via tethering.

Nevertheless, ectopically expressed native TFE3 accumulated in the nucleus and displayed broad promoter binding. The fusion partner is therefore not absolutely necessary for promoter binding, but as mentioned above may increase the number of bound sites and impact splicing and/or selectivity of gene activation. For example, while ectopic native TFE3 activates the OxPhos program, it is less efficient than the fusion proteins that in addition activate a larger contingent of genes perhaps via specific interactions of the fusion partners with additional cofactors. Fusion-specific interactions with cofactors and selective gene activation, may also explain the propensity of ASPSCR1 tumors to more readily undergo EMT.

The ability of constitutively nuclear native TFE3 to bind broadly to promoters may, however, be pertinent to tRCC where TFEB is either amplified or fused with the non-coding RNA MALAT1

(Kauffman et al, 2014; Napolitano and Ballabio, 2016; Argani et al, 2016). In both situations, native TFEB accumulates in the nucleus suggesting that activation of OxPhos and/or promiscuous promoter binding may also be facets of its oncogenic function.

## TFE3 fusion-driven OxPhos and ferroptosis resistance as mechanisms driving tRCC tumorigenesis

Loss and gain of function experiments revealed that TFE3 fusions directly drive a large set of OxPhos genes. This was clearly shown upon ectopic fusion protein expression in HEK293T cells that activated numerous OxPhos genes leading to increased functional OxPhos levels. These observations were consolidated by the globally elevated OxPhos scores seen in two independent patient tumor cohorts compared to ccRCC (Qu et al, 2022; Bakouny et al, 2022; TCGA, 2013). This function of TFE3 in tumors is reminiscent of its role in metabolism and mitochondrial function in normal tissues (Pastore et al, 2017). Furthermore, by stratifying ccRCC tumors of different EMT status, we showed that OxPhos score was strongly reduced during EMT in ccRCC, but maintained at higher levels in tRCC. Moreover, in a recent preprint, Li et al (2024) reported that numerous genes involved in mitochondrial function and OxPhos, in particular components of the electron transport chain that we defined here as direct TFE3 targets were selective vulnerabilities of tRCC. Together these data strongly consolidate the idea the TFE3-fusions directly activate genes promoting OxPhos that are essential for tRCC tumorigenesis. High OxPhos has been associated with an increased propensity for oncogenic transformation and resistance to apoptosis (Bonnay et al, 2020; Spurlock et al, 2021). Similarly, maintaining elevated OxPhos may promote tumor growth in the stressful cellular environments encountered in primary tumors and during metastatic dissemination (Bergers and Fendt, 2021). The TFE3-related transcription factor MITF was shown to drive OxPhos in melanoma through regulation of PPARGC1A, but also the non-coding RNAs SAMMSON and LENOX (Vazquez et al, 2013; Leucci et al, 2016; Gambi et al, 2022). High OxPhos may hence represent a common feature of MiT family-dependent tumors.

Another mechanism that may contribute to tRCC tumorigenesis is ferroptosis resistance with pro-ferroptotic genes down-regulated during tumorigenesis and those promoting resistance up-regulated. Accordingly, tRCC lines showed marked sensitivity to GPX4 inhibition with IC50 values in the nanomolar range similar to renal medullary carcinoma lines (Vokshi et al, 2023). Ferroptosis-related genes were found in the core program of TFE3-fusion proteins, with TFE3-fusions bound to M/E-box motifs in the *GPX4*, *HMOX1,* and *GCLC* proximal promoters and the up-regulation of these genes by TFE3 fusion protein expression in HEKT cells. Notably, *GCLC* and *HMOX1* have been described as NFE2L2

(NRF2) targets (Yan et al, 2023; Yang et al, 2023b), a pathway reported to be active in tRCC (Bakouny et al, 2022). Although NFE2L2 can regulate antioxidant and ferroptosis response in different cellular contexts, in tRCC TFE3 fusions may either bypass or cooperate with NFE2L2 to directly regulate key ferroptosis resistance genes and protect from oxidative stress.

Direct TFE3 fusion regulation of genes involved in OxPhos and ferroptosis therefore constitute two complementary mechanisms that may contribute to tRCC tumorigenesis. Moreover, the observation of ferroptosis resistance in tRCC underlines this process as a common mechanism for RCC development (Yang et al, 2023a).

## EMT and myCAFs, markers of poor outcome common to tRCC and ccRCC

Analyses of the tRCC cohorts revealed that EMT and presence of myCAFs both associated with poor outcome, characteristics shared with ccRCC (Davidson et al, 2023). In each tRCC cohort, we stratified tumors that differed in EMT status, hot-immune microenvironment and presence of CAFs. As previously observed in human tumors (Sun et al, 2021) and tRCC murine models (Prakasam et al, 2024), ASPSCR1-TFE3 tumors displayed a strong propensity for EMT, but tumors with other fusion partners also underwent EMT. Interestingly, tumors with PRCC or SFPQ fusions displayed either epithelial or mesenchymal signatures highlighting the variability among their phenotypes. EMT of tRCC cancer cells was further consolidated by deconvolution with signatures for epithelial and mesenchymal ccRCC cancer cells. Tumors with EMT signatures were enriched in mesenchymal cancer cells showing that the bulk EMT signature was not only due to presence of fibroblasts or other components of the TME. Similarly, deconvolution revealed that EMT tumors were enriched in myCAFs and that enrichment in both mesenchymal tumor cells and myCAFs were strongly associated with poor survival. Thus, despite the distinct oncogenic mechanisms, both ccRCC and tRCC tumors display EMT and infiltration by myCAFs that constitute hallmarks of poor disease outcome. Future studies will clarify the similarities and differences in the myCAF across RCC subtypes and whether this could be used to personalize treatments of patients using immune checkpoint inhibitors (Alhalabi et al, 2023; Boilève et al, 2018).

# Methods

### Reagents and tools table

| Reagent/Resource | Reference or Source | Identifier or Catalog Number |
|---|---|---|
| **Experimental models** | | |
| UOK109 cells (*H. sapiens*) | Dr M Linehan | N/A |
| UOK120 cells (*H. sapiens*) | Dr M Linehan | N/A |
| UOK146 cells (*H. sapiens*) | Dr M Linehan | N/A |
| TF1 cells (*H. sapiens*) | Dr G Pages | N/A |
| UOK121 cells (*H. sapiens*) | Dr M Linehan | N/A |
| UOK112 cells (*H. sapiens*) | Dr M Linehan | N/A |
| RCC4 cells (*H. sapiens*) | Cellosaurus | CVCL_0498 |

| Reagent/Resource | Reference or Source | Identifier or Catalog Number |
|---|---|---|
| A-498 cells (*H. sapiens*) | ATCC | HTB-44 |
| ACHN cells (*H. sapiens*) | ATCC | CRL-1611 |
| HEK293T cells (*H. sapiens*) | ATCC | CRL-3216 |
| 786-O | ATCC | CRL-1932 |
| **Recombinant DNA** | | |
| pUC57+ | GenScript | N/A |
| pCW57-GFP-P2A-MCS | Addgene | 71783 |
| pCMV8.91 | Addgene | 202687 |
| phCMV-VSV | Addgene | 207318 |
| **Antibodies** | | |
| Rabbit anti-TFE | Sigma-Aldrich | HPA-023881 |
| Rabbit anti-TFE | Abcam | Ab93808 |
| Mouse anti-VCL | Sigma-Aldrich | V4505 |
| Rabbit anti-BRG1 | Abcam | ab110641 |
| Rabbit anti-GAPDH | In house | N/A |
| Rabbit anti-Histone H3 | Abcam | ab1791 |
| Rabbit anti-H3K27ac | ActiveMotif | 39133 |
| Rabbit anti-IgG | Abcam | ab171870 |
| **Oligonucleotides and other sequence-based reagents** | | |
| ON-TARGETplus Non-targeting Control pool | Horizon Discovery | D-001810-10 |
| ON-TARGETplus Human TFE3 siRNA - n°6 | Horizon Discovery | J-009363-06 |
| ON-TARGETplus Human TFE3 siRNA - n°8 | Horizon Discovery | J-009363-08 |
| ON-TARGETplus Human TFE3 siRNA - n°9 | Horizon Discovery | J-009363-09 |
| **Chemicals, Enzymes and other reagents** | | |
| OPTI-MEM | GibcoTM | 31985070 |
| lipofectamine RNAiMAX | Invitrogen | 13778150 |
| polybrene | Sigma-Aldrich | TR-1003 |
| puromycin | InvivoGen | ant-pr-5 |
| doxycycline | Thermo Scientific Chemicals | J60422.06 |
| TRI Reagent | MRC | TR 118 |
| TurboDNase kit | ThermoFisher | AM1907 |
| SuperScript IV | ThermoFisher | 18090050 |
| SYBR Green I | Roche | 4707516001 |
| Protease Inhibitor cocktail | ThermoFisher | 1861280 |
| Bradford protein quantification assay | BioRad | 5000006 |
| Pierce ECL Western Blotting substrates | Thermo Scientific TM | 32106 |
| ECL Detection system | GE Healthcare | N/A |
| Crystal Violet | Sigma-Aldrich | C0775 |
| poly-D-lysine | Fisher Scientific | 16021412 |
| XF Cell MitoStress Test kit | Agilent | 103015-100 |

| Reagent/Resource | Reference or Source | Identifier or Catalog Number |
|---|---|---|
| SeaHorse XF base medium | Agilent | 103335-100 |
| SeaHorse XF 1.0 glucose solution | Agilent | 103577-100 |
| SeaHorse XF 100 mM pyruvate solution | Agilent | 103578-100 |
| SeaHorse XF 200 mM glutamine solution | Agilent | 103579-100 |
| SeaHorse XF Calibrant Solution | Agilent | 103059-000 |
| SeaHorse XF Pro M FluxPak | Agilent | 103775-100 |
| DAPI | Invitrogen Fisher Scientific | 11580246 |
| RNAseA | Thermo Scientific | HY-129046 |
| proteinase K | Thermo Scientific | EO0491 |
| phenol-chloroform | Invitrogen | AM9722 |
| DynaGreen TM Protein G Magnetic beads | Invitrogen | 17852854 |
| CellStripper | Corning | 25-056-CI |
| Cut&TAG-IT TM Assay kit | ActiveMotif | 53160 |
| NexteraTM Compatible Multiplex primers | ActiveMotif | 53155 |
| Click-IT TM RNA Alexa Fluor 594 Imaging kit | Invitrogen | C10330 |
| 8-well cell culture slide | Mattek | CCS8 |
| Actinomycin D | Thermo Fisher | A7592 |
| DAPI-ProLongGold | Thermo Fisher | P36935 |
| RSL3 | SelleckChem | S7243 |
| PrestoBlue TM | Invitrogen | A13261 |
| GSH/GSSG-Glo Assay kit | Promega | V6612 |
| **Software** | | |
| R v4.3.3 | https://www.r-project.org/ | |
| Rstudio Build 764 | https://posit.co/download/rstudio-desktop/ | |
| ComBat-seq | https://github.com/zhangyuqing/ComBat-seq | |
| CIBERSORTx | https://cibersortx.stanford.edu/ | |
| Survival v3.2 | https://cran.r-project.org/web/packages/survival/index.html | |
| Survminer v0.4.9 | https://cran.r-project.org/web/packages/survminer/index.html | |
| MCPcounter v1.2.0 | https://github.com/ebecht/MCPcounter | |
| ConsensusClusterPlus v3.17 | https://bioconductor.org/packages/release/bioc/html/ConsensusClusterPlus.html | |
| Pheatmap v1.0.12 | https://www.rdocumentation.org/packages/pheatmap/versions/1.0.12/topics/pheatmap | |
| DESeq2 v1.16 | https://bioconductor.org/packages/release/bioc/html/DESeq2.html | |
| STAR v2.5.3a | https://github.com/alexdobin/STAR | |
| HTSeq v0.6.1p1 | https://htseq.readthedocs.io/en/latest/ | |
| Bowtie v2.2.8 | https://bowtie-bio.sourceforge.net/index.shtml | |
| MACS v2.2 | https://pypi.org/project/MACS2/ | |
| DiffBind v3.8.4 | https://bioconductor.org/packages/release/bioc/html/DiffBind.html | |
| HOMER v4.11 | http://homer.ucsd.edu/homer/motif/ | |
| bedtools v2.26 | https://bedtools.readthedocs.io/en/latest/ | |
| seqMINER | https://www.france-genomique.org/seqminer/ | |
| MEME | https://meme-suite.org/meme/ | |
| RSAT | http://rsat.sb-roscoff.fr/peak-motifs_form.cgi | |
| GSEA software v3.0 | https://www.gsea-msigdb.org | |
| GSVA v1.48.3 | https://www.bioconductor.org/packages/release/bioc/html/GSVA.html | |
| clusterProfiler v4.8.2 | https://bioconductor.org/packages/release/bioc/html/clusterProfiler.html | |
| Database for Annotation, Vizualisation and Integrated Discovery (DAVID) | http://david.abcc.ncifcrf.gov/ | |
| Venny | https://bioinfogp.cnb.csic.es/tools/venny/ | |
| UCSC genome browser | https://genome.ucsc.edu/ | |
| Metascape | https://metascape.org | |
| GraphPad Prism v10 | https://www.graphpad.com | |
| ImageJ | https://www.graphpad.com | |
| **Other** | | |

## Cell culture

UOK109, UOK120, UOK121, UOK112, and UOK146 were generous gift from Dr Martson Linehan (NIH, Bethesda, USA) while TF1 line was a generous gift from Dr Gilles Pagès (Nice, France). The tRCC cells were authenticated by RNA-seq and immunoblot to express the expected TFE3 fusion proteins whereas cell lines obtained from databanks were not reauthenticated. These cell lines were grown in DMEM medium supplemented with

glucose (4.5 g/L), 10% heat-inactivated Fetal Calf Serum (FCS), Non-essential amino acid (AANE) and 1% penicillin/streptomycin. The remaining cell lines were purchased from ATCC or Cellosaurus. The embryonic kidney HEK293T cell line and the derived lines expressing ectopic TFE3 fusions as well as UOK121, UOK112, and RCC4 cells were grown in DMEM medium supplemented with glucose (1 g/L), 10% FCS and 1% penicillin/ streptomycin. The primary kidney cell line RPTEC was grown in DMEM/F12 medium supplemented with 2.5 mM L-Glutamine, 15 mM HEPES, hTERT Immortalized RPTEC growth kit-supplement A&B (ATCC ACS-4007), gentamycin (40 μg/mL), and G418 (100 μg/mL). 786-O ccRCC cells were grown in RPMI 1640 medium supplemented with glucose (2.5 g/L), HEPES (10 mM), 10% FCS, sodium pyruvate (1 mM) and gentamycin (40 μg/mL). A-498 ccRCC cells were grown in MEM medium (LifeTechnologies 31095-029) supplemented with 10% heat-inactivated FCS and gentamycin (40 μg/mL). The papillary carcinoma renal cell line ACHN was grown in MEM medium (ATC 30-2003) supplemented with 10% FCS and gentamycin (40 μg/mL). All cells were cultured at 37 °C, 5% $CO_2$. All cell lines were regularly tested using the Venor GeM Mycoplasma Detection Kit, and used at less than 10 passages.

## Cell line transfection with siRNA

For siRNA knockdown experiments, 400,000 cells were plated in 6-well plates 18 h before transfection. Transfection was performed in OPTI-MEM medium using lipofectamine RNAiMAX (13778150 – Invitrogen) with 25 nM siRNA (mix of 3 individual siRNA for knockdown of TFE3 – J-009363 -06, -08, -09, or with a pool of 4 non-targeting siRNA as control – D-0018-10 – Dharmacon). After 7 h of incubation, OPTI-MEM was replaced by complete medium. siRNAs sequences are listed in Dataset EV7.

## Plasmid production, lentiviral particle production, cell line transduction and expression

TFE3, NONO-TFE3 and PRCC-TFE3 cDNAs were synthesized by GenScript and subsequently cloned into the all-in-one doxycycline inducible lentiviral vector pCW57-GFP-P2A-MCS vector (Addgene plasmid #71783) in place of GFP. Lentiviral particles were produced to express independently each of the 3 genes of interest or GFP control by transfection of HEK293T cells with packaging plasmids. Lentiviral particles were purified by filtration and ultracentrifugation and resuspended in PBS. After titration, HEK293T cells were infected at multiplicity of infection of 1 using polybrene and selected by puromycin addition to the media (1 μg/mL). After establishment of the stable cell lines, expression of the proteins of interest was induced by adding of Doxycycline (Dox) in the medium (10 μg/mL).

## RNA extraction and quantitative PCR

Total mRNA isolation was performed using Trizol (TR 118) and isopropanol precipitation. RNA was then treated with DNAseI following the TurboDNAse kit instructions (AM1907 – ThermoFisher) and 1 μg of RNA was reversed transcribed using SuperScript IV (18090050 – ThermoFisher) following manufacturer instructions. qPCR (program: 95 °C, 15 min; (95 °C, 10 s; 55 °C, 15 s; 72 °C, 10 s) x60; (95 °C, 5 s; 70 °C, 1 min 05; 95 °C, 5 min) x1; cool down) was performed using SYBR Green

I (Roche) and monitored by a LightCycler 480 (Roche). Target gene expression was normalized using the geometric means of *TBP*, *HMBS*, *GAPDH*, *RPLP0* as reference genes. Primers for RT-qPCR were designed using the OligoAnalyzer™Tool from IDT and are listed in Dataset EV7.

## Total protein extraction

Whole cell extracts were prepared by 4 freeze-thaw cycles in LSDB 0.5 M buffer (0.5 M KCl, 25 mM Tris pH 7.9, 10% glycerol v/v, 0.05% NP-40 v/v, 16 mM DTT and protease inhibitor cocktail, 1861280 – Thermo Fisher). After 20 min on ice, extracts were centrifuged for 10 min at $13,000 \times g$, 4 °C and supernatant was retrieved.

## Differential extraction of nuclear and cytoplasmic proteins

Protein extracts from nuclear and cytoplasmic compartments were prepared using manual lysis and sucrose buffer. Briefly, cells were resuspended in hypotonic buffer (1 mL buffer/g of cells – 10 mM Tris pH 7.65, 1.5 mM $MgCl_2$, 10 mM KCl), incubated on ice for 5 min and transferred to a loose Dounce. After 15 strokes, sucrose buffer (0.33 mL/g cells – 20 mM Tris pH 7.65, 15 mM KCl, 60 mM NaCl, 0.34 M sucrose) and PMSF (0.2 mM) were added to the lysed solution and centrifuged for 10 min at $4500 \times g$, 4 °C. The supernatant was recovered (cytosolic extract) and the pellet resuspended in sucrose buffer (1 mL/g cells). Hypertonic buffer (20 mM Tris pH 7.65, 25% glycerol, 1.5 mM $MgCl_2$, 0.2 mM EDTA, 900 mM NaCl) was added dropwise under slight agitation (0.35 mL/g cells), tubes were incubated for 30 min at 4 °C on a wheel and centrifuged for 10 min at $4700 \times g$, 4 °C. The supernatant formed the soluble nuclear extract.

## Immuno-blots

Proteins were quantified using the Bradford protein quantification assay (BioRad) and 20 μg of extract were used to perform SDS-polyacrylamide gel electrophoresis. Proteins were transferred to a nitrocellulose membrane. Membranes were incubated overnight at 4 °C with primary antibody diluted in PBS + 5% dry-fat milk + 0,01% Tween-20. After washing steps, membranes were incubated with HRP-conjugated secondary antibodies for 1 h at room temperature and revealed using the ECL detection system (GE Healthcare). Antibodies used and their dilutions are listed in Dataset EV7.

## Colony forming analyses by crystal violet staining

Colony formation was assessed by plating 5000 cells in 6-well plates, incubating them for 14 days and finally fixing in 4% paraformaldehyde and staining with 0.05% Crystal Violet solution (Sigma-Aldrich). Modified HEK293T lines used for Dox-inducible gain of function experiments were plated on poly-D-lysine pre-coated dishes and Dox was renewed every 48 h.

## RNA-sequencing and analysis

Cells were harvested by scraping 72 h after siRNA transfection and total mRNA was extracted with Trizol as described above. For HEKT cells, RNA was prepared 24 h after Doxycycline (Dox)

addition. Following QC and library preparation, RNA-seq was performed on an Illumina Hiseq4000 as 2*100 bp paired-end sequencing following Illumina's instructions. After sequencing raw reads were pre-processed in order to remove adapter and low-quality sequences (Phred quality score below 20) using cutadapt version 1.10 and reads shorter than 40 bases were discarded. Reads were mapped to rRNA sequences using bowtie version 2.2.8 and matching reads removed. Remaining reads were then mapped onto the hg19 assembly of Homo sapiens genome using STAR version 2.5.3a. Gene expression quantification was performed from uniquely aligned reads using htseq-count version 0.6.1p1, with annotations from Ensembl version 75 and "union" mode. Only non-ambiguously assigned reads were retained for further analyses. Read counts were normalized across samples with the median-of-ratios method. Comparisons of interest were performed using the Wald test for differential expression and implemented in the Bioconductor package DESeq2 version 1.16.1. Genes with high Cook's distance were filtered out and independent filtering based on the mean of normalized counts was performed. $P$-values were adjusted for multiple testing using the Benjamini and Hochberg method. Deregulated genes were defined as genes with log2(fold-change) $>1$ or $<-1$ and adjusted $p$-value $< 0.05$. Gene set enrichment analyses were done with the GSEA software v3.0 using the hallmark gene sets of Molecular Signature Database v6.2. Gene Ontology analysis was performed using the Database for Annotation, Visualization and Integrated Discovery (http://david.abcc.ncifcrf.gov/). Gene list intersections were computed and Venn diagrams represented using the web tool Venny (https://bioinfogp.cnb.csic.es/tools/venny/). Gene set variation analysis (GSEA) was computed using the Bioconductor R package GSVA v1.48.3.

## Measure of oxygen consumption rate (OCR) in living cells

OCR in living cells was measured using the MitoStress kit and the XF96 extracellular analyser (Agilent) following manufacturer's instructions. Cells were seeded 48 h prior to measurement in SeaHorse microplates (Agilent) in order to have confluent cells on the day of measurement: 30,000 cells per well for translocation cell lines, 40,000 cells for Dox-inducible HEK293T lines, 10,000 cells for 786-O and 20,000 cells for UOK121. Dox-inducible HEK293T cells were treated with Dox 24 h prior to measurement. For siRNA-mediated knockdown, 30,000 UOK109 cells were seeded per well, after 7 h of incubation with siRNA and 65 h prior to measurement, whereas 20,000 cells were seeded for TF1, UOK120, UOK146. For each line, 6 wells were considered for technical replicates per experiment and up to 5 biological replicates were performed. The day before measurement, the cartridge was hydrated. One hour before measurement, medium on cells was changed for XF base medium supplemented with 1 mM pyruvate, 2 mM glutamine and 10 mM glucose and cells were incubated at 37 °C with no $CO_2$. After calibration of the cartridge, cells were sequentially exposed to 1 µM oligomycin, 1 µM carbonyl cyanide-4-(trifluorome-thoxy)-phenylhydrazone (FCCP), and 0.5 µM rotenone/antimycin A, with 13 min between each drug. After measurement, cells were fixed with 4% PFA, permeabilized with 0.2% Triton and stained with DAPI (1 µg/µL). Number of cells per well was determined by microscopy at the IGBMC High Throughput Screening Facility and used to normalize obtained measurements per well. Calculation of basal, maximal and reserve capacity were performed as defined in the Agilent User Guide.

## ChiP-seq and Cut&TAG

For H3K27ac chromatin immunoprecipitation, cells were grown in 15 cm petri dishes and fixed with 0.4% PFA for 10 min at room temperature with agitation. Crosslinking was quenched by adding glycine (2 M, pH 8) for 10 min with agitation. Fixed cells were washed once with PBS, harvested by scraping, pelleted, and resuspended in lysis buffer (2-times pellet volume – 50 mM TrisHCl pH 8, 1% SDS, 10 mM EDTA, 0.1 M PMSF, protease inhibitor cocktail). Lysates were transferred to Covaris tubes and sonicated in a Covaris sonicator to obtain DNA fragments (median size 200–500 bp). After sonication, lysates were centrifuged at $13,000 \times g$, 4 °C for 10 min. Chromatin sonication was verified on agarose gels after decross-linking (overnight incubation at 65 °C with 0.3 M NaCl, RNAseA and 1 h incubation with proteinase K at 42 °C) and phenol-chloroform extraction. 100 µg of chromatin were diluted with ChIP dilution buffer (16.7 mM Tris HCl pH 8, 167 nM NaCl, 1.2 mM EDTA, 1.1% Triton X-100, 0.01% SDS) and precleared with Protein G DNA magnetic beads (30 µL/mL lysate) for 1 h, 4 °C. After bead removal, chromatin extracts were incubated with anti-H3K27ac (1 µg/20 µg of chromatin) at 4 °C, overnight. To recover antibody-chromatin complexes, Protein G DNA magnetic beads were added to each sample and incubated at 4 °C for 1 h. Beads were then washed with 4 different buffers, two times each (Low Salt Buffer: 20 mM Tris HCL pH 8, 150 mM NaCl, 2 mM EDTA, 1% Triton X-100, 0.1% SDS; High Salt Buffer: 20 mM Tris HCL pH 8, 500 mM NaCl, 2 mM EDTA, 1% Triton X-100, 0.1% SDS; LiCl Buffer: 10 mM Tris HCl pH 8, 1 mM EDTA, 1% sodium deoxycholate, 1% NP40, 0.25 mM LiCl; TE Buffer: 10 mM Tris HCl pH 8, 1 mM EDTA). Finally, beads were resuspended twice in DNA elution buffer (0.1 M $NaHCO_3$, 1%SDS) and incubated 15 min at room temperature. Decross-linking and phenol-chloroform extraction were then performed and extracted DNA was resuspended in water. After QC and library preparation, DNA was sequenced on Illumina HiSeq 4000 sequencer as 1*50 bp single-read following Illumina's instructions.

For Cut&TAG, cells were plated in 10 cm petri-dish for 48 h and harvested using CellStripper solution and washed 2 times in PBS 1X. 500,000 cells were used for Cut&TAG experiments following the manufacturer's instructions (Cut&TAG-IT™ Assay kit – n°53160 – ActiveMotif). Libraries were prepared using the Nextera™-Compatible Multiplex Primers (96-plex – n°53155 – ActiveMotif). Libraries were verified on a BioAnalyzer before 2 rounds of purification using SPRI beads as described in the Cut&TAG kit.

## Analysis of ChIP-seq and Cut&TAG

Following sequencing, reads were mapped to the Homo sapiens genome assembly hg19 using Bowtie with the following arguments: -m 1 --strata --best -y -S -l 40 -p 2. Peak detection was performed using the MACS software v1.4.3 for transcription factors and v2.1.1 for chromatin marks. Peaks were annotated with the 'annotate-Peaks' command from HOMER v4.11 using the GTF annotation file from ENSEMBL v75 (http://homer.ucsd.edu/homer/ngs/

annotation.html). Global clustering analyses with quantitative comparisons and representations of read density heatmaps were performed using seqMINER. Gene ontology analyses from peak annotations were performed with the Bioconductor R package clusterProfiler v4.8.2 and the metascape web tool (https://metascape.org). Motif discovery from ChIP-seq peak sequences was performed using the MEME CHIP and RSAT peak-motifs algorithms (http://rsat.sb-roscoff.fr/peak-motifs_form.cgi). Peak intersections were computed using bedtools v2.26.0 and for multiple samples, the R package DiffBind v3.8.4. Visualization of ChIP-seq signal at specific gene loci of interest was achieved using the UCSC genome browser (https://genome.ucsc.edu/).

## Patient samples

The tRCC patient cohort comprised 39 samples from 38 patients with one patient presenting a primary tumor and metastasis resulting in two samples. The in-house cohort comprises 26 samples and an additional 13 samples from TCGA KIRC and KIRP cohorts. A full description is provided in Dataset EV5 and Fig. EV4A,B. Identification of fusion transcripts for 17 of those samples have been previously reported (Classe et al, 2017; Malouf et al, 2014) and nine samples were novel. All patients had previously provided written informed consent for tumor collection and analysis and the experiments conformed to the principles set out in the WMA Declaration of Helsinki and the Department of Health and Human Services Belmont Report. The study was approved by the ethical committee of the Pitie-Salpetriere Hospital (IDF-6, Ile de France).

## RNA-sequencing and analysis of patient tumor samples

Library preparation, sequencing and fusion transcript detection for those novel samples was performed as previously described (Malouf et al, 2014). After sequencing, raw read counts were adjusted for batch-effect correction using ComBat-seq by providing the sequencing batch as the 'batch' argument. For analyses requiring integration with TCGA-KIRC samples, an additional covariate specifying the type of sample (tRCC, normal adjacent tissue or ccRCC) was provided as the 'group' argument. The adjusted raw-count matrices were normalized by sequencing depth using DESeq2 size-factors and then gene-counts were divided by median transcript lengths.

Consensus clustering of TFE3-fusion samples was performed using the R package ConsensusClusterPlus v3.17 following standard procedure. In short, matrices are filtered in order to keep only coding genes based on their biotype annotation. The 5000 most variable genes were retained with the mad() function and the matrices were median centered with the sweep(), apply() and median() functions. Consensus clustering was done with the ConsensusClusterPlus() function using base parameters. The appropriate numbers of clusters were chosen based on the curve of cumulative distribution function. Differential gene expression analyses between the different groups were done with DESeq2 with functional analyses and visualization performed as described in the previous section for cellular models.

Translocation tumor samples were subjected to deconvolution analysis using the R-package MCP-counter v1.2.0. Then, hierarchical clustering was performed based on the MCP-counter

population scores using the hclust() function with "ward.D2" linkage in order to define "hot" and "cold" tumor groups.

Translocation tumor samples were scored according to their mesenchymal/epithelial signature ratios in the following way: epithelial and mesenchymal genes derived from ccRCC samples described in Liang et al (2018) were subjected to unsupervised clustering using tRCC samples then these genes were assigned into mesenchymal and epithelial categories based on their expression patterns in tRCC in order to adapt the original signatures. Some genes such as VEGFA and ANGPTL4 showed different expression patterns in tRCC compared to what was described in ccRCC and were re-assigned to their proper categories while some genes such as CA9 had no clear expression patterns in tRCC and were removed. Lastly, mesenchymal genes and proximal tubule markers identified from single-cell analyses were added to complete the new mesenchymal and epithelial signatures in order to have the 22 representative genes in each category. The 'SignatureRatio' value was computed as the geometric mean of the mesenchymal signature divided by the geometric mean of the epithelial signature.

Deconvolution of tRCC samples using the ccRCC single-cell RNA-seq signatures we previously described was performed using CIBERSORTx algorithm (Davidson et al, 2023). To stratify epithelial, intermediate and mesenchymal ccRCC samples, tumor samples from the TCGA-KIRC were ordered by the ccRCC.epi/ccRCC.mes ratio of their CIBERSORTx score, then the top 20 were chosen as epithelial, the bottom 20 as mesenchymal and the 20 around the median as intermediate.

Scoring of a considered list of genes to determine patient survival was defined as the geometrical mean of representative genes of the transcriptional signature.

Patient survival was computed in R using packages survival v3.2 and survminer v0.4.9. Patients were stratified using the surv_cut-point() function and Kaplan–Meier curves were represented with the survfit() and ggsurvplot() functions. Hazard ratios were determined by univariate Cox proportional-hazards models with the "coxph()" function.

## Labeling nascent RNA with 5-ethynyluridine (EU)

EU incorporation experiments were conducted using the Click-iT™ RNA Alexa Fluor™ 594 Imaging kit from Invitrogen (C10330), following the supplier's recommendations. Cells were transfected with siCTR or siTFE3 in 6-well plates before replating 25,000 cells in 8-well cell culture slides (CCS8 – Mattek) and growing for 24 h. Fresh media containing 1 mM of the cell-permeable uridine analog, 5-ethynyluridine (EU) and a final concentration of 50 ng/mL Actinomycin D were added for one hour prior to fixation. Fixation was performed with a 3.7% Formaldehyde solution in PBS for 15 min, followed by permeabilization with a 0.5% Triton-X PBS solution for an additional 15 min before continuing with the EU labeling in accordance with the protocol's instructions. After one wash step with PBS 1x, cells were fixed with 3.7% formaldehyde for 15 min at RT, washed once with PBS and permeabilized with 0.5% Triton X-100 in PBS 1x for 15 min and washed again with PBS. Cells were then incubated with Click-iT® reaction cocktail as described in the kit for 30 min at RT protected from light. The cell nuclei were counterstained with Hoechst 33342. After two washes in PBS 1x, microscope slides were mounted using DAPI-

**The paper explained**

**Problem**

Translocation renal cell carcinoma (tRCC) is a rare subtype of kidney cancer characterized by genetic translocation events frequently involving transcription factor TFE3 or more rarely TFEB. While the resulting fusion proteins are considered as the oncogenic drivers, their mechanism of action remains poorly understood.

**Results**

By integrative multi-omics analyses in tRCC cell lines and patient tumors together with loss and gain of function experiments, we found broad binding of TFE3-fusion proteins at active promoters and identified a core set of target genes involved in multiple pathways, including oxidative metabolism (OxPhos) and ferroptosis. Consequently, tRCC cell lines displayed higher functional OxPhos levels and patient tumors displayed elevated OxPhos scores and ferroptosis resistance gene expression. Analyses of tRCC patient transcriptome data further revealed that mesenchymal tRCC tumors are enriched in myofibroblastic cancer-associated fibroblasts that are hallmarks of poor prognostic outcome.

**Impact**

This study advances understanding of the molecular mechanisms underlying oncogenic transformation by TFE3 fusion proteins by defining a core program of their target genes and key features of tumors and their microenvironment that negatively impact patient outcome. Our integrative multi-omics and functional analyses reveal how extensive genomic binding of TFE3 fusion proteins drives high levels of oxidative metabolism, ferroptosis resistance, and general RNA synthesis.

ProLongGold (P36935 – ThermoFisher). Images were capture with a SP8-X confocal microscope and the EU fluorescence signal from the nucleus was quantified using the Hoechst signal as a mask for determining the mean fluorescence of the EU signal in each nucleus by analyses using a FlowJo macro program.

### Treatment with RSL3 and determination of IC50

10,000 cells were plated in 96-well plate and cultured overnight. Cells were treated with 1, 5, 10, 25, 50, 100, or 250 nM of RSL3 with or without 1 µM ferrostatin-1 (SelleckChem, #S7243) and cultured for 48 h. Each condition was performed in triplicate. Complete medium was replaced by a solution of 1:1 complete medium/PrestoBlue™ (A13261 – Invitrogen) and incubated for 1 h at 37 °C, 5% $CO_2$. Viability was then evaluated by measuring the absorbance at 590 nm. Dose response curves and the half maximal inhibitory concentration (IC50) were calculated using GraphPad Prism software.

### Measurement of SGGS and GSH levels

Glutathione (GSH) and oxidized glutathione (GSSG) levels were measured in cells using the GSH/GSSG-Glo Assay kit (Promega, V6612) following manufacturer's instructions. Cells were plated on 96-well plate (60,000 cells/well) and incubated at 37 °C, 5% $CO_2$. Dox-inducible HEK293T cells were treated with Dox 24 h prior to measurement. For siRNA-mediated knockdown, tRCC cells were transfected for 7 h with siRNA prior to seeding in 96-well plates and then harvested and measured after a further 41 h. GSH and GSSG concentrations were measured in each well. For each cell line,

3 wells were considered for technical replicates per experiment and 3 biological replicates were performed. GSH and GSSG standard curves were prepared for each experiment. After removing the media, cells were lysed with either total (GSH) or oxidized (GSSG) glutathione reagent for 5 min at room temperature with shaking. Luciferin generation reagent was then added for each well and incubated for 30 min, followed by the addition of Luciferin detection reagent for 15 min. Prior to measurement, samples were transferred to a white, opaque, flat-bottom 96-well plate and luminescence was measured using BERTHOLD Centro XC³ LB 960 Luminometer.

### Statistics

The statistical tests used in each analysis, mainly paired t-test, one-way ANOVA (with Dunett's correction), or Wilcoxon test, are described in the corresponding figure legends with the p-values. No blinding nor randomization were performed and sample size was at least $N = 3$ biological replicates (Haq et al, 2013).

## Data availability

The sequencing data regarding mechanistic studies reported in this manuscript are available as SuperSeries GSE268093 in the GEO data base. The underlying raw sequencing data regarding transcriptomic analysis of patients tumor samples are not publicly available for this manuscript due to patient privacy requirements and lack of authorization for distribution.

The source data of this paper are collected in the following database record: biostudies:S-SCDT-10_1038-S44321-025-00221-7.

## Peer review information

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

## Acknowledgements

We thank the staff of all the IGBMC common facilities. This work was supported by grants from the Institut National du Cancer (PRTK 2020-046), ARC foundation, MSD AVENIR and the Ligue Nationale contre le Cancer. ID is an 'équipe labellisée' of the Ligue Nationale contre le Cancer. GM is an 'équipe labellisée' of the ARC. AH was supported by the Ministère de la Recherche and the Ligue Nationale contre le Cancer. Sequencing was performed by the GenomEast platform, a member of the 'France Génomique' consortium (ANR-10-INBS-0009).

## Author contributions

**Alexandra Helleux**: Conceptualization; Data curation; Formal analysis; Investigation; Visualization; Methodology; Writing—original draft; Writing—review and editing. **Guillaume Davidson**: Data curation; Software; Formal analysis; Writing—original draft; Writing—review and editing. **Antonin Lallement**: Investigation. **Fatima Al Hourani**: Investigation. **Alexandre Haller**: Investigation. **Isabelle Michel**: Investigation. **Anas Fadloun**: Investigation. **Christelle Thibault-Carpentier**: Resources; Software; Investigation. **Xiaoping Su**: Resources; Software. **Véronique Lindner**: Resources; Investigation. **Thibault Tricard**: Resources. **Hervé Lang**: Resources. **Nizar M Tannir**: Resources. **Irwin Davidson**: Conceptualization; Data curation; Formal analysis; Supervision; Funding acquisition; Investigation; Methodology; Writing—original draft; Project administration; Writing—review and editing. **Gabriel G Malouf**: Conceptualization; Resources; Funding acquisition; Visualization; Methodology; Project administration; Writing—review and editing.

Source data underlying figure panels in this paper may have individual authorship assigned. Where available, figure panel/source data authorship is listed in the following database record: biostudies:S-SCDT-10_1038-S44321-025-00221-7.

## Disclosure and competing interests statement

GGM reported consulting fees from BMS, MSD, IPSEN, and Eisai as well as research grants from MSD AVENIR.

# Expanded View Figures

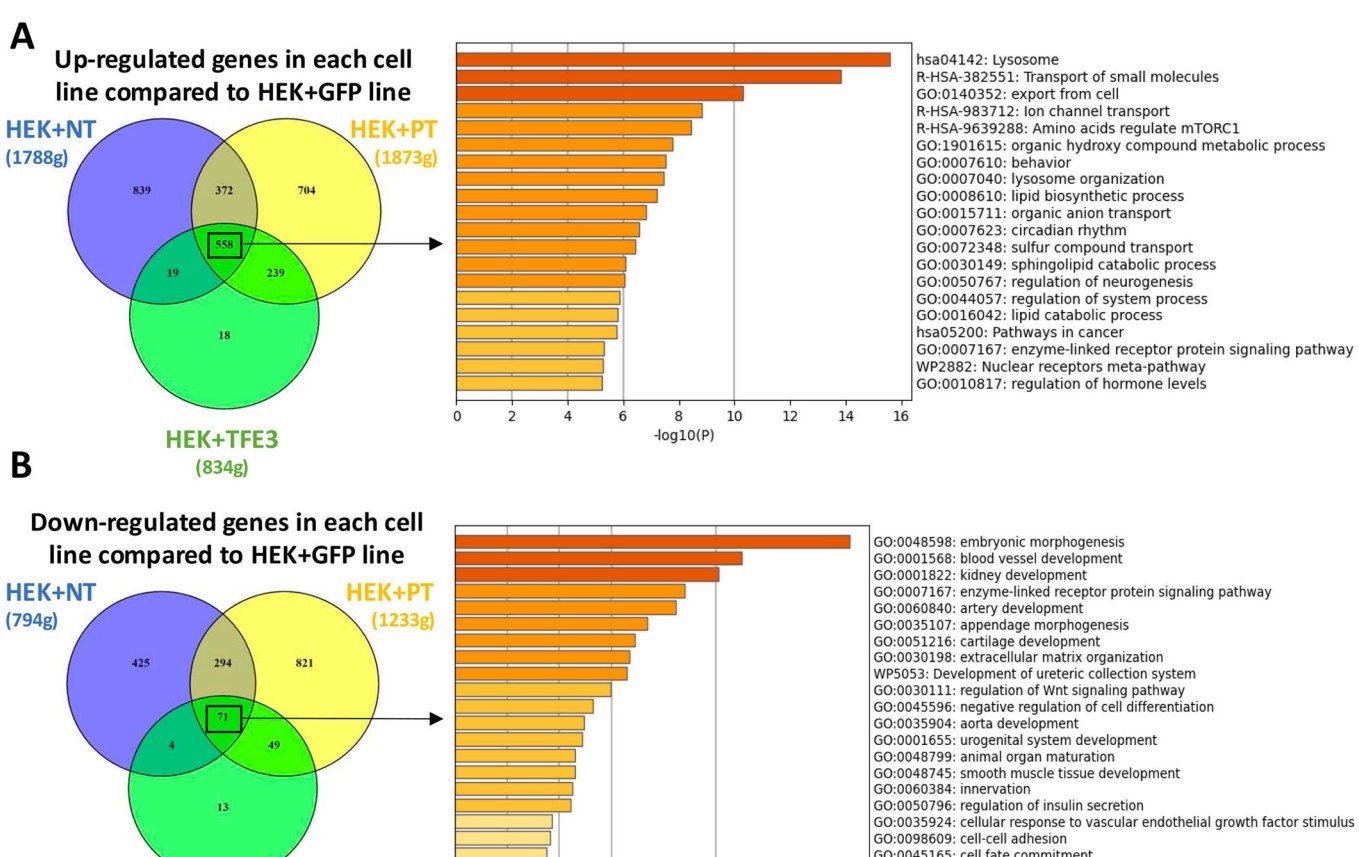

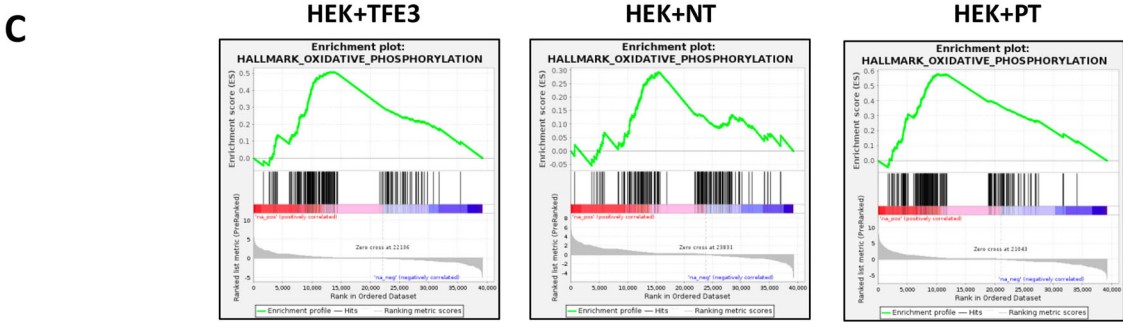

**Figure EV1. Genes regulated by ectopically expressed TFE3 fusion proteins.**

(A, B) Venn diagrams showing the number of deregulated genes in each line and the associated ontology analysis of the commonly regulated genes using MetaScape software with *p*-values calculated by an accumulative hypergeometric test. (C) Enrichment plots of the Oxidative Phosphorylation (OxPhos) pathway according to GSEA analysis with weighted Kolmogorov–Smirnov test after ectopic expression of native TFE3 or NT or PT fusion proteins in HEKT cells over GFP condition. Source data are available online for this figure.

**A**

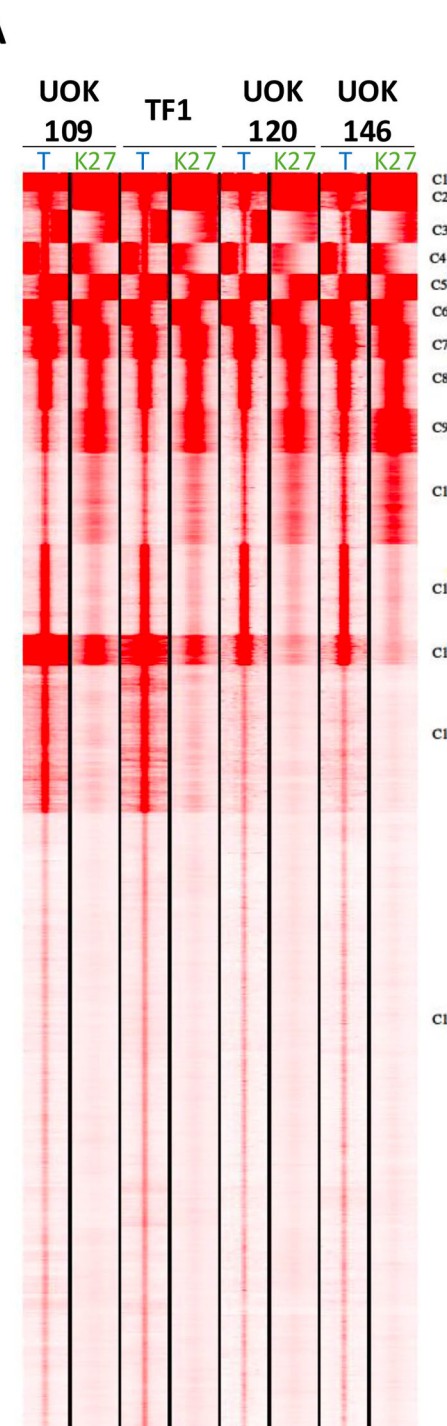

**Non-redundant TFE3 peaks**

**(169 896 sites)**

**B**

| Cell line | Cut&Tag TFE3 | ChIPseq H3K27ac | TFE3 with H3K27ac and E-box |
|---|---|---|---|
| UOK109 | 65901 | 44600 | 18466 (28%) |
| TF1 | 78598 | 56957 | 19919 (25%) |
| UOK120 | 33819 | 64545 | 11106 (33%) |
| UOK146 | 42760 | 51008 | 12714 (30%) |

**C**

| Cell line | Motif | e-value | Distribution |
|---|---|---|---|
| UOK109 | SP2 (MA0516.3) KLF5 (MA0599.1) SP4 -MA0685.2) | 3.4e-126 | |
| | FOSL2 (MA0478.1) FOSB::JUNB (MA1135.1) FOS::JUN (MA0099.3) | 2.0e-120 | |
| | MITF (MA0620.3) TFEB (MA0692.1) TFEB_full | 1.7e-53 | |
| TF1 | FOSL2 (MA0478.1) FOSL2::JUNB (MA1138.1) | 4.0e-164 | |
| | KLF12 (MA0742.2) KLF10 (MA1511.2) SP2 (MA0516.3) | 8.1e-84 | |
| | MITF (MA0620.3) USF1_DBD TFEB (MA0692.1) | 3.2e-46 | |
| UOK120 | TFEB (MA0692.1) TFEB_full TFE3_DBD | 3.0e-618 | |
| UOK146 | TFEB (MA0692.1) TFEB_full TFE3_DBD | 8.6e-587 | |

**D**

| Top1000 Peaks – RSAT analysis | UOK109 | | TF1 | | UOK120 | | UOK146 | |
|---|---|---|---|---|---|---|---|---|
| Motif | Rank | e-value | Rank | e-value | Rank | e-value | Rank | e-value |
| MITF, USF2 motif | 2 | 2.9e-61 | 6 | 3.4e-28 | 1 | 1.6e-66 | 2 | 6.9e-56 |
| SP1, KLF15 motif | 1 | 1.5e-79 | 1 | 5.6e-73 | 2 | 3.0e-54 | 1 | 3.6e-58 |
| NFYA, NFYC motif | 3 | 5.8e-29 | 3 | 7.6e-30 | / | / | / | / |

**Figure EV2. Profiling of TFE3 fusion protein genomic occupancy.**

(A) Read density maps for TFE3 fusion protein occupancy (T) and H3K27ac (K27) at all non-redundant sites. (B) Total numbers of peaks with the indicated characteristics in each cell line after removal of low-occupied sites (C14 of the density map in A). (C) Results of MEME-ChIP analyses of the DNA motifs at the top 1000 TFE3 bound sites in each cell line, with associated e-value and motif distribution over the peak. (D) Results of RAST analyses of transcription factor binding motifs at the top 1000 TFE3 bound sites in each cell line showing associated rank and e-value.

**A**

| HEK293T Cell line | Cut&Tag TFE3 peaks | TFE3 peaks at TSS (500nt) |
|---|---|---|
| + GFP | 1058 | 615 (58%) |
| + TFE3 | 17250 | 9716 (56%) |
| + NONO-TFE3 | 28815 | 14690 (51%) |
| + PRCC-TFE3 | 20255 | 9707 (48%) |

**B**

### KEGG pathways

| Pathway | count |
|---|---|
| Lysosome | 19 |
| Oxidative phosphorylation | 19 |
| Autophagy - animal | 20 |
| Collecting duct acid secretion | 7 |
| Synaptic vesicle cycle | 11 |
| Human T-cell leukemia virus 1 infection | 20 |
| Chemical carcinogenesis - reactive oxygen species | 20 |
| mTOR signaling pathway | 16 |
| FoxO signaling pathway | 14 |
| Circadian rhythm | 7 |

p-value (FDR<0.05)

**C**

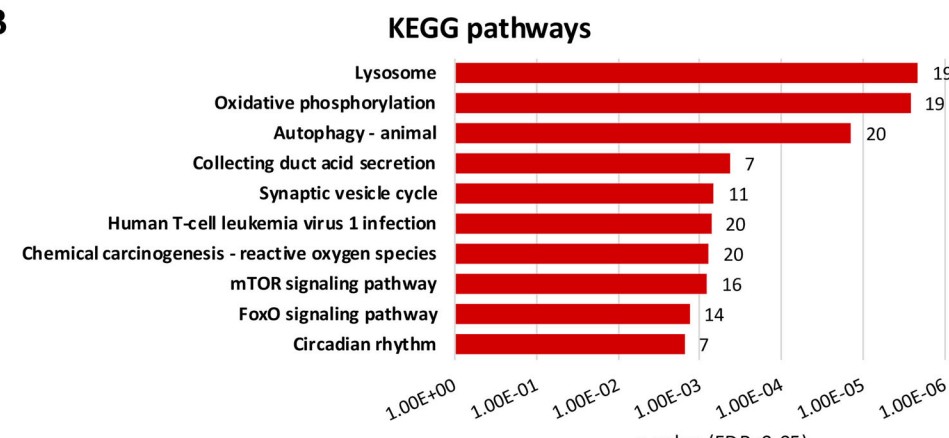

| Cell line HEK293T + | Motif | | e-value | Distribution |
|---|---|---|---|---|
| GFP | TFE3_DBD TFEB (MA0692.1) TFEB_full | | 1.0e-184 | |
| TFE3 | KLF15 (MA1513.1) KLF12 (MA0742.2) KLF10 (MA1511.2) | | 2.5e-203 | Not centrally enriched |
| | TFEC (MA0871.2) USF2 (MA0526.4) Bhlhb2_primary (UP00050_1) | | 1.2e-54 | |
| NONO-TFE3 | KLF15 (MA1513.1) SP2 (MA0516.3) KLF12 (MA0742.2) | | 3.6e-428 | |
| | NFYA (MA0060.3) NFYC (MA1644.1) NFYB (MA0502.2) | | 3.2e-330 | |
| PRCC-TFE3 | SP2 (MA0516.3) KLF12 (MA0742.2) KLF10 (MA1511.2) | | 1.6e-196 | |
| | TFE3_DBD TFEB (MA0692.1) TFEB_full | | 1.1e-91 | |

**D**

| Number of OxPhos genes common to the 4 tRCC lines With open-TFE3+H3K27ac-Ebox associated peak | |
|---|---|
| Among the 235 OxPhos-signature genes | 145 |
| Among the downregulated genes in siTFE3 (54) | 33 |
| Among the upregulated genes in HEK+ (59) | 44 |
| Among both downregulated genes in siTFE3 + upregulated genes in HEK+ (23) | 19 |

◄ **Figure EV3.  TFE3 binding in HEKT cells.**

(**A**) Total numbers of peaks and those at the proximal promoters (500 nt from the TSS) in the indicated HEKT cell line. (**B**) KEGG ontology analyses of genes with binding of endogenous TFE in the proximal promoter in the HEK-GFP line. Each indicated KEGG pathway presented an FDR < 0.05 and an associated *p*-value calculated by hypergeometric distribution and Benjamini-Hochberg correction with the number of genes found in each pathway was noted to the right of the bar. (**C**) Results of MEME-ChIP analyses of the DNA motifs at the top 1000 TFE3 bound sites in each cell line, with associated e-value and motif distribution over the peak. (**D**) Number of OxPhos genes in each category associated with TFE3 occupied M/E box containing a H3K27ac-marked regulatory elements common to each cell line. Source data are available online for this figure.

**A**

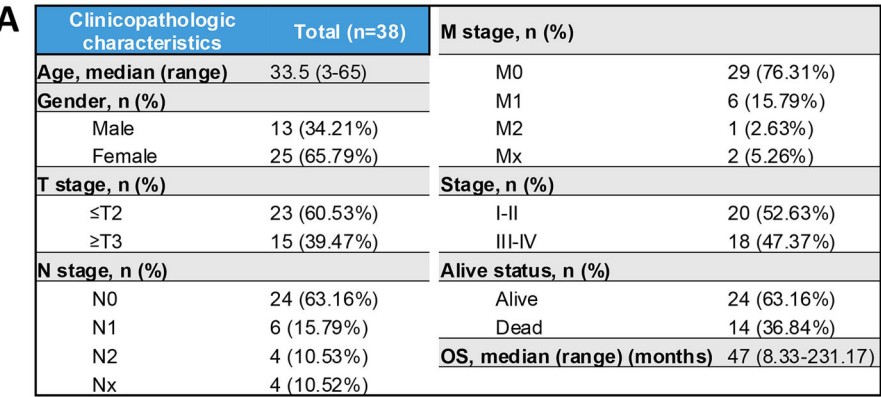

| Clinicopathologic characteristics | Total (n=38) | M stage, n (%) | |
|---|---|---|---|
| Age, median (range) | 33.5 (3-65) | M0 | 29 (76.31%) |
| Gender, n (%) | | M1 | 6 (15.79%) |
| Male | 13 (34.21%) | M2 | 1 (2.63%) |
| Female | 25 (65.79%) | Mx | 2 (5.26%) |
| T stage, n (%) | | Stage, n (%) | |
| ≤T2 | 23 (60.53%) | I-II | 20 (52.63%) |
| ≥T3 | 15 (39.47%) | III-IV | 18 (47.37%) |
| N stage, n (%) | | Alive status, n (%) | |
| N0 | 24 (63.16%) | Alive | 24 (63.16%) |
| N1 | 6 (15.79%) | Dead | 14 (36.84%) |
| N2 | 4 (10.53%) | OS, median (range) (months) | 47 (8.33-231.17) |
| Nx | 4 (10.52%) | | |

**B**

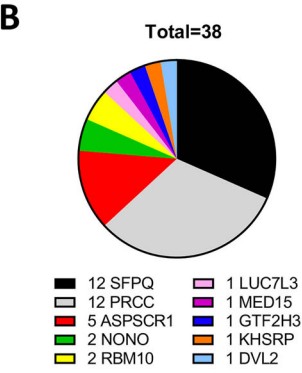

Total=38

- 12 SFPQ
- 12 PRCC
- 5 ASPSCR1
- 2 NONO
- 2 RBM10
- 1 LUC7L3
- 1 MED15
- 1 GTF2H3
- 1 KHSRP
- 1 DVL2

**C**

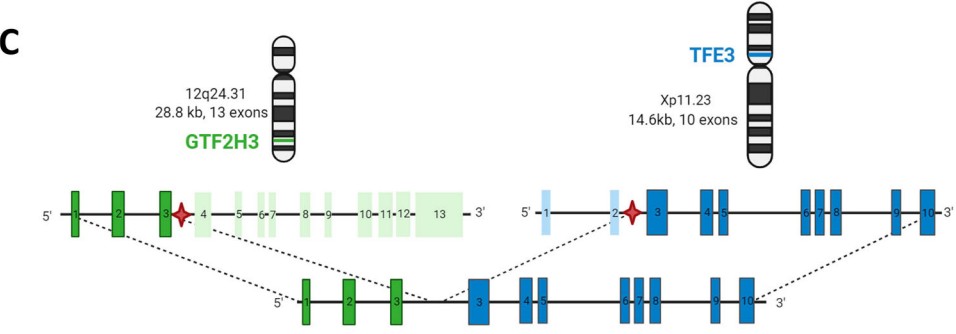

Fusion transcript GTF2H3(e1-3)-TFE3(e3-10)

**D**

### Gene ontology on protein coding genes upregulated in Tumor samples vs NAT (2856g)

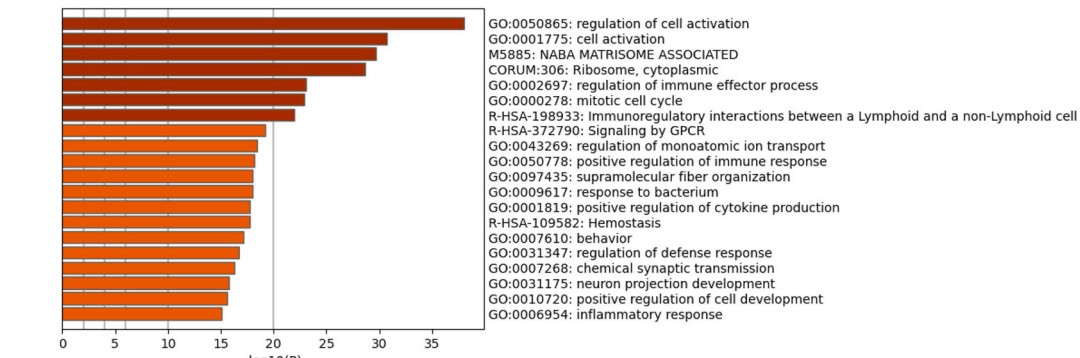

GO:0050865: regulation of cell activation
GO:0001775: cell activation
M5885: NABA MATRISOME ASSOCIATED
CORUM:306: Ribosome, cytoplasmic
GO:0002697: regulation of immune effector process
GO:0000278: mitotic cell cycle
R-HSA-198933: Immunoregulatory interactions between a Lymphoid and a non-Lymphoid cell
R-HSA-372790: Signaling by GPCR
GO:0043269: regulation of monoatomic ion transport
GO:0050778: positive regulation of immune response
GO:0097435: supramolecular fiber organization
GO:0009617: response to bacterium
GO:0001819: positive regulation of cytokine production
R-HSA-109582: Hemostasis
GO:0007610: behavior
GO:0031347: regulation of defense response
GO:0007268: chemical synaptic transmission
GO:0031175: neuron projection development
GO:0010720: positive regulation of cell development
GO:0006954: inflammatory response

-log10(P)

**E**

### Gene ontology on protein coding genes upregulated in NAT vs Tumor samples (2146g)

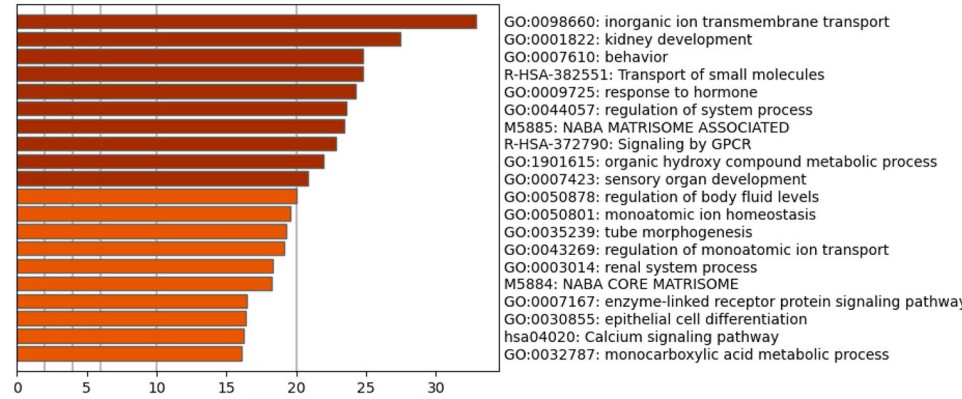

GO:0098660: inorganic ion transmembrane transport
GO:0001822: kidney development
GO:0007610: behavior
R-HSA-382551: Transport of small molecules
GO:0009725: response to hormone
GO:0044057: regulation of system process
M5885: NABA MATRISOME ASSOCIATED
R-HSA-372790: Signaling by GPCR
GO:1901615: organic hydroxy compound metabolic process
GO:0007423: sensory organ development
GO:0050878: regulation of body fluid levels
GO:0050801: monoatomic ion homeostasis
GO:0035239: tube morphogenesis
GO:0043269: regulation of monoatomic ion transport
GO:0003014: renal system process
M5884: NABA CORE MATRISOME
GO:0007167: enzyme-linked receptor protein signaling pathway
GO:0030855: epithelial cell differentiation
hsa04020: Calcium signaling pathway
GO:0032787: monocarboxylic acid metabolic process

-log10(P)

**Figure EV4. Clinical characteristics of the tRCC cohort.**

(A) Clinical and pathology characteristics of the tRCC patient tumor cohort. A full summary is presented in Dataset EV5. (B) Pie chart showing the contribution of the indicated fusion partners to the cohort. (C) Schematic representation of a novel *TFE3* fusion with *GTF2H3* encoding the p34 subunit of general transcription factor TFIIH. (D, E) Ontology of genes up- (D) or down- (E) regulated in tRCC tumors compared with NAT with *p*-values calculated by an accumulative hypergeometric test. Source data are available online for this figure.

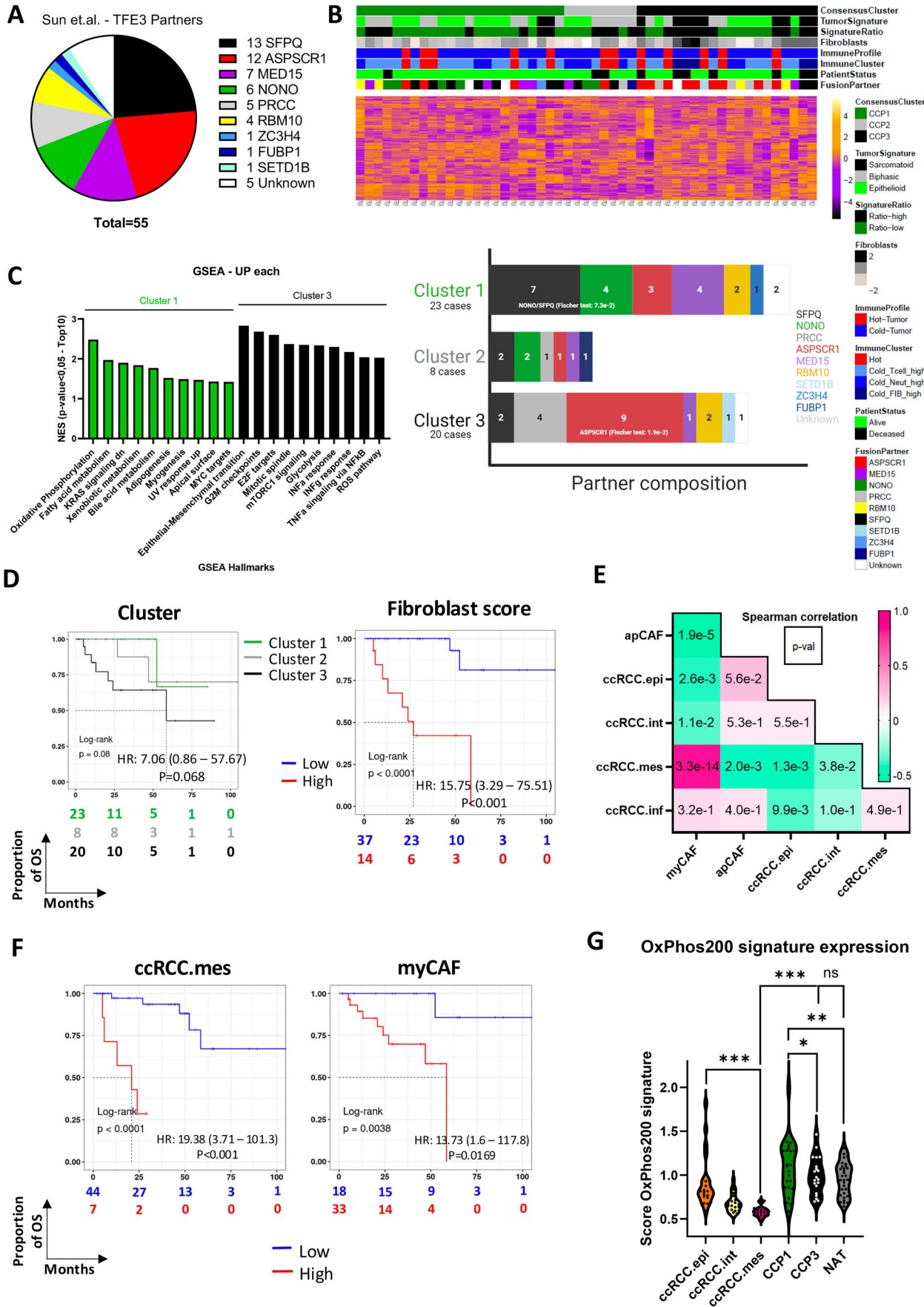

**Figure EV5. Clinical and molecular characteristics of an independent tRCC cohort.**

(A) Pie chart showing the contribution of the indicated fusion partners to the cohort. (B) Heatmap of the unsupervised clustering with the indicated clinical and molecular parameters showing the division of samples in 3 clusters (upper panel). Distribution of samples by fusion partner within the 3 major clusters (lower panel). (C) GSEA analyses of genes differentially expressed between clusters 1 and 3 indicating the Top 10 enriched terms (weighted Kolmogorov–Smirnov test $p$-value < 0.05, FDR < 0.05). (D) Kaplan–Meier curves for overall survival in patients according to the cluster or the fibroblast score, calculated from MCP-counter, using the optimal cut point method with the associated log-rank $p$-value and Hazard ratio (HR) from the univariate Cox proportional-hazard model. (E) Spearman correlation coefficient (colored box) and associated $p$-value (number in box) between the indicated populations after deconvolution using the ccRCC tumor cell and CAF signatures inferred by CIBERSORTx on bulk RNA-seq data from the tRCC tumor samples. (F) Kaplan–Meier curves for overall survival in patients according to ccRCC.mes or myCAF scores using the optimal cut point method with the associated log-rank $p$-value from Wald test and Hazard ratio (HR) from the univariate Cox proportional-hazard model. CcRCC.mes $p$-value 9.99E−07; MyCAF $p$-value 5.26E−06. (G) Expression score of the OxPhos genes in NAT, each tRCC cluster and the indicated ccRCC tumor types from the TCGA.KIRC collection. Overall scores were compared by Wilcoxon test (*: $p$.val < 0.05, **: $p$.val < 0.01, ***: $p$.val < 0.001) and are summarized in Dataset EV6. Exact $p$-values; ccRCC.epi vs ccRCC.mes 2.71E−09; ccRCC.mes vs CCP3 1.45E−11; CPP1 vs NAT 8.21E−03; CPP1 vs CPP3 3.28E−03; CPP3 vs NAT 7.28E−01. CcRCC.epi, ccRCC.int, ccRCC.mes $n = 20$; CCP1 $n = 23$; CPP3 $n = 20$; NAT $n = 18$. Source data are available online for this figure.

