## [Peer Review File · EMBO Molecular Medicine]

TFE3 fusions drive oxidative metabolism and ferroptosis resistance in translocation renal carcinomas

Alexandra Helleux, Guillaume Davidson, Antonin Lallement, Fatima Al Hourani, Alexandre Haller, Isabelle Michel, Anas Fadloun, Christelle Thibault-Carpentier, Xiaoping Su, Veronique Lindner, Thibault Tricard, Hervé Lang, Nizar Tannir, Irwin Davidson, and Gabriel Malouf

Corresponding authors: Gabriel Malouf (gabriel.malouf@igbmc.fr) , Irwin Davidson (irwin@igbmc.fr)

Review Timeline:

Submission Date:	2nd Aug 24
Editorial Decision:	12th Sep 24
Appeal:	26th Sep 24
Editorial Decision:	1st Oct 24
Revision Received:	10th Jan 25
Editorial Decision:	17th Feb 25
Revision Received:	27th Feb 25
Accepted:	4th Mar 25

Editor: Lise Roth

Transaction Report:

12th Sep 2024

Decision on your manuscript EMM-2024-20385

Dear Prof. Malouf,

Thank you for the submission of your manuscript to EMBO Molecular Medicine, and please accept my apologies for the delay in getting back to you as one referee needed more time to complete his/her review. We have now received the feedback from the three referees who reviewed your manuscript.

As you will see from the reports below, while they acknowledge the potential interest of the study and the amount of work performed, they also raise several concerns, including (but not limited to) the compromised novelty, the lack of validation of identified therapeutic targets, and the lack of direct evidence that elevated Ox-Phos in tRCC is driving oncogenesis or promoting progression. As clear and conclusive novel translational insight is key for publication in EMBO Molecular Medicine, and given that at EMBO Press we encourage a single round of revisions in a limited time frame, I am afraid I see little choice but to return the manuscript to you at this point with the decision that we cannot offer to publish it.

I am very sorry to disappoint you in this occasion, and hope that the referees' comments are helpful in your continued work in this area.

Sincerely,

Lise Roth

Lise Roth
Senior Editor
EMBO Molecular Medicine

***** Reviewer's comments *****

Referee #1 (Remarks for Author):

The manuscript by Helleux et al is a comprehensive transcriptional evaluation of tRCC. The abundant amount of work and data analysis to answer key questions is clear in this study. Although exciting and much needed in the field of tRCC, there is a lot of data presented which makes the final manuscript less clear and focused. This manuscript appears to read as two stand-alone papers, and the authors should consider the necessity of publishing this as a combined, massive manuscript. The following comments need to be addressed:

1. Supplemental Figure 1B should include fractionation controls.
2. When you overexpress TFE3 (Fig 2B), what is the additional band that is observed almost at the same molecular weight as the PRCC-TFE3 size? Is overexpression of TFE3 inducing a fusion event in HEK cells?
3. What were the ox-Phos related genes exclusively bound by either NONO-TFE3 or PRCC-TFE3 i.e. not common but those that were differentially expressed genes.
4. Is the impact of GPX4 inhibitor greater (i.e., is there enhanced sensitivity) in Nono-TFE3 cell lines versus PRCC-TFE3?
5. In Fig 4D, the authors are comparing Ox Phos score of tRCC to ccRCC but the more relevant comparison should be with NAT, is the signature higher in tRCC in this scenario?
6. The study has conducted a well-executed, in -depth analysis of transcriptional signatures of TFE3 fusion protein containing cell lines and samples from patients. However, the Ox-Phos related gene cluster upregulated in response to the TFE3-NONO and PRCC fusions do not appear to be consistent with the patient data, where the Ox-Phos data is prominent for the non-NONO and non-PRCC cluster 3. So, although the story came together very well with Ox-Phos and ferroptosis related gene signatures being upregulated in cell lines, the patient data does not directly support the cell line data.
7. The comparisons to ccRCC need to be rationalized and explicitly explained.
8. There is no direct evidence suggesting that elevated Ox-Phos in tRCC is driving oncogenesis or promoting progression. The same is true for ferroptosis, in fact there is no ferroptosis signature in patient samples. Additional experimentation will be required to make these claims, and the discussion should accordingly be modified to reflect this.
9. This manuscript is comprehensive and feels like two manuscripts. The data at this juncture will benefit from simply addressing what happens when you have the NONO and PRCC fusion proteins and when you have non-NONO and PRCC fusion proteins.
10. The motif/element that the fusion proteins bind to does not appear to make a difference since non-NONO and PRCC patient samples showed maximal OX-PHOS gene signatures.
11. The discussion is too lengthy and needs to be significantly shortened.

Referee #2 (Comments on Novelty/Model System for Author):

2. The novelty of several findings reported in this study seem to be compromised by some recent publications. For instance, the involvement of oxidative phosphorylation metabolism and ferroptosis (glutathione metabolism) in the tumorigenesis of tRCC were previously described in Reference 45 (Qu et al. 2022). The involvement of EMT in tRCC tumorigenesis was already discussed in Reference 25 (Sun et al. 2021) and Prakasam et al. 2024 (doi.org/10.1172/JCI1170559).

3. Several potential therapeutic targets for tRCC are speculated in the Discussion of this study. At least, some in vitro and, preferentially, pre-clinical evidence with models of tRCC would be desirable to be conducted to show the medical feasibility of this study.

4. No pre-clinical model system is described in this study. Only cell lines in vitro and several patient samples.

Referee #2 (Remarks for Author):

The manuscript by Helleux et al. reports that 'TFE3 fusions drive oxidative metabolism and ferroptosis resistance in translocation renal carcinomas'. The authors conducted well-designed in vitro experiments to prove the necessity and sufficiency of translocation TFE3 to better understand the tumorigenesis of translocation renal cell carcinomas (tRCCs). Firstly, the oxidative phosphorylation pathway was found to be enriched both by silencing TFE3 in TFE3-translocated cell lines and by expressing translocated TFE3 in normal kidney HEK293T cells. These results were validated by assessing the Oxygen Consumption Rate (OCR). The authors then performed state-of-the-art Cleavage Under Targets and Tagmentation (CUT&Tag) in the tRCC cell lines along with H3K27ac ChIP-Seq to identify active cis-regulatory elements of translocation TFE3. As expected, the M/E-box motif was highly overrepresented, followed by SP1/KLF and NFY motifs, in particular in NONO-TFE3 fusion cells. Since the ferroptosis pathway was overrepresented in the promoter occupancy at M/E-box sites of TFE3 fusion proteins, the tRCC cell lines were treated with the GPX4 inhibitor RSL3 and showed a sensitivity that was blocked by the ferroptosis inhibitor ferristatin-1. Furthermore, analyses of tRCC patient samples confirmed the overrepresentation of the oxidative phosphorylation pathway, besides the identification of epithelial to mesenchymal transition (EMT) and myofibroblastic cancer-associated fibroblasts (myCAFs), which were associated to poor patient survival. Finally, a 10-gene signature of TFE3-fusion regulated genes was proposed, which was correlated with poor patient overall survival. Overall, the manuscript is well written (despite some typos) and most experiments were carefully controlled. Conclusions are mostly supported by the data presented. However, there are several concerns that need to be addressed before the manuscript could be accepted for publication, as follows:

- The novelty of several findings reported in this study seem to be compromised by some recent publications. For instance, the involvement of oxidative phosphorylation metabolism and ferroptosis (glutathione metabolism) in the tumorigenesis of tRCC were previously described in Reference 45 (Qu et al. 2022). The involvement of EMT in tRCC tumorigenesis was already discussed in Reference 25 (Sun et al. 2021) and Prakasam et al. 2024 (doi.org/10.1172/JCI1170559). Therefore, the findings described in this manuscript should be placed in context of previous discoveries already reported in the literature in the Introduction and/or Discussion sections. In addition, a preprint posted shortly after the submission of this manuscript (doi.org/10.1101/2024.08.09.607311) might also be discussed.

- This study performed sophisticated CUT&Tag experiments not present in similar publications in the literature to identify active cis-regulatory elements of TFE3 translocation. It is a pity that, besides the well-known M/E-box motif, the molecular characterization, validation and functional significance of novel motifs is very limited.

- Ferroptosis is highlighted in the title, abstract and throughout the manuscript, but the evidence presented to support the involvement of this pathway in tRCC tumorigenesis is limited to the promoter occupancy at M/E-box sites of TFE3 fusion proteins and the in vitro response to ferroptosis drugs. The RNA-Seq of gain and loss-of-function experiments for the ferroptosis genes should be compared. The treatments with RSL3 and ferristatin-1 should also be performed on the tRCC cell lines with TFE3 silencing and HEK cells with TFE3 overexpression to rule out the involvement of TFE3 translocation. In addition, similarly to OCR assays to validate oxidative phosphorylation, it would be necessary to show some ferroptosis markers, such as deregulated glutathione metabolism and/or lipid peroxidation, in the gain and loss-of-function in vitro experiments to claim involvement of ferroptosis in tRCC tumorigenesis.

- Several potential therapeutic targets for tRCC are speculated in the Discussion of this study. At least, some in vitro and, preferentially, pre-clinical evidence with models of tRCC would be desirable to be conducted to show the medical feasibility of this study.

- The OCR assays should also be performed on the tRCC cell lines with TFE3 silencing and ccRCC cell lines with TFE3 silencing to further evaluate the involvement of TFE3 in the basal and maximal respiration levels.

- Statistics need to be revised. The Dunnett's multiple comparison test compares the means from several experimental groups against only one control group, whereas the Wilcoxon test compares the means of only two groups. Unfortunately, the text to the Fig. 4D in the Results section describes comparisons not tested for some groups. Therefore, the Tukey post hoc test should be more appropriate for comparing means among all the experimental groups after ANOVA. In addition, the authors claim that TF1 showed "higher basal, but not maximal OCR levels (Fig. 1H)", but differences in basal respiration of TF1 show no significance.
- Information regarding the ethics approvals and the informed patient consents for the in-house tRCC patient samples must be provided. In addition, a file with the individual clinicopathological characteristics for each patient from the in-house cohort and TCGA cohort (summarized in Fig. S9A) should be provide as supplementary dataset.
- All the multi-omics data reported in this study, such as RNA-Seq, ChIP-Seq and CUT&Tag, should be deposited in a public repository, such as Gene Expression Omnibus (GEO) or ArrayExpress.
- Some typos should be double checked. For instance, UOK129 in Fig. 1B should be UOK120.

Referee #3 (Comments on Novelty/Model System for Author):

This is an elegant study that further dissects the oncogenic role of TFE3 fusion proteins in tRCC. There are some recent papers that should be discussed. Additional experiments would strengthen the author's conclusions.

Referee #3 (Remarks for Author):

In this manuscript Helleux A et al investigated the oncogenic mechanisms by which TFE3 fusion proteins drive tRCC. The authors report that high nuclear accumulation of NONO-TFE3 or PRCC-TFE3 fusion proteins promotes binding across the genome at H3K27ac-marked active chromatin and engage M/E-box containing regulatory elements to activate gene expression and stimulate mRNA synthesis. Furthermore, TFE3 fusions directly regulate genes involved in ferroptosis resistance and oxidative phosphorylation metabolism. The authors also report that tRCC tumor aggressiveness is associated with EMT and the presence of mesenchymal tRCC cancer cells associates with myofibroblast cancer-associated fibroblasts. Overall, the authors define tRCC as a novel metabolic subtype of renal cancer suggesting that genomic binding of TFE3 fusion proteins promotes tRCC tumorigenesis by regulating oxidative phosphorylation and ferroptosis resistance.

1. The authors should discuss recent papers on the oncogenic properties of different TFE3 fusion proteins and the OxPhos transcriptional program in tRCC (Li J et al TFE3 fusions direct an oncogenic transcriptional program that drives OXPHOS and unveils vulnerabilities in translocation renal cell carcinoma BioRxiv August 10, 2024; Damayanti N et al TFE3-Splicing Factor Fusions Represent Functional Drivers and Druggable Targets in Translocation Renal Cell Carcinoma. Cancer Res April 15, 2024). Thus, the authors should highlight the novelty of their findings.
2. The authors should comment on why in In Fig. 1, despite a very modest KD of TF3 protein expression in UK109 cells as compared with the other cell lines, there is still a significant inhibition of cell proliferation. In addition, the Western blots show differential TFE3 protein expression across the different models but the Seahorse analysis for baseline OCR does not seem to correlate with that (TF1 and UK146 cells appear to have similar baseline OCR despite significant difference in TFE3 protein expression). The authors should elaborate more on whether additional factors might be involved in regulating OxPhos in these models.
3. What was the rationale for introducing four ccRCC lines (RCC4, UOK121, A-498, and 786-0) but only progressing with two (UOK121 and 786-0-Figure 1G)? Why were these two lines chosen over RCC4 and A-498?
4. The overlap in upregulated oxidative phosphorylation consistent across HEK+TFE3, HEK+NT, and HEK+PT challenges the claim that "TFE3 fusion proteins activated genes of the OxPhos pathway leading to enhanced OxPhos metabolism." The results instead suggest the increase in oxidative phosphorylation is driven by TFE3, independent of the fusion. A greater discussion on the potential specific role of the fusion protein (i.e fusion partner) should be provided.
5. The authors should comment on potential mechanisms responsible for a differential sensitivity to ferroptosis induction between TF1 and UK109 cells (Fig. 3I) which carry the same fusion protein.
6. The experimental platform would benefit from including an ASPSCR1-TFE3 fusion model since the authors show that PRCC and NONO somewhat cluster together in patients while ASPSCR1 seems to have a different phenotype.
7. Regarding the patient data cohorts, the authors identified Cluster 1 and 2 as being more oxidative while Cluster 3 was primarily associated with EMT and poorer prognosis. ASPSCR1 fusion is already known to be more aggressive and has been reported to induce EMT (i.e Prakasam G et al Comparative genomics incorporating translocation renal cell carcinoma mouse model reveals molecular mechanisms of tumorigenesis JCI February 22, 2024). The authors should also elaborate more on the potential mechanisms underlying differential expression of EMT across the different fusion proteins despite a similar OxPhos induction.

Minor:

1. In the abstract the following sentence seems to be incomplete: We further show that tRCC tumour aggressiveness is related to EMT which although enriched in tumours harbouring ASPSCR1-TFE3 fusions.

2. The statement that tRCC is "highly resistant to immune checkpoint inhibitors" (page 24) is not correct as different studies have shown activity in this patient population (Lee CH et al ,. Phase II Trial of Cabozantinib Plus Nivolumab in Patients With Non-Clear-Cell Renal Cell Carcinoma and Genomic Correlates. J Clin Oncol. 2022; Albiges L et al , Pembrolizumab plus lenvatinib as first-line therapy for advanced non-clear-cell renal cell carcinoma (KEYNOTE-B61): a single-arm, multicentre, phase 2 trial. Lancet Oncol. 2023).

As a service to authors, EMBO provides authors with the possibility to transfer a manuscript that one journal cannot offer to publish to another EMBO publication. The full manuscript and if applicable, reviewers reports are automatically sent to the receiving journal to allow for fast handling and a prompt decision on your manuscript. For more details of this service, and to transfer your manuscript to another EMBO title please click on Link Not Available

Dear Dr Roth,

Thank you for handling our paper and for expediting the review process. We have carefully considered the referees comments and are somewhat surprised by your decision to reject the paper and not give us the opportunity to respond to the comments. We felt that the three reviewers have positive comments and we can answer their concerns quickly.

The purpose of this letter is not to provide a detail point by point response to the all of their comments, but to highlight a number of key issues we think we can address, in particular the lack of novelty for which we fully disagree.

Referee 1 comments

“The abundant amount of work and data analysis to answer key questions is clear in this study. Although exciting and much needed in the field of tRCC, there is a lot of data presented which makes the final manuscript less clear and focused. This manuscript appears to read as two stand-alone papers, and the authors should consider the necessity of publishing this as a combined, massive manuscript”.

Reviewer 1 is very positive about the manuscript and the wealth of data it encompassed.

Referee 2 comments

‘ - The novelty of several findings reported in this study seem to be compromised by some recent publications. For instance, the involvement of oxidative phosphorylation metabolism and ferroptosis (glutathione metabolism) in the tumorigenesis of tRCC were previously described in Reference 45 (Qu et al. 2022).’

The observation that tRCC tumours display a high OxPhos signature is not novel and indeed we never claimed that our study was the first to show this. The referee is correct is saying that this is already shown in the Qu et al paper (as well as others) that we cited in our manuscript. Nevertheless, the molecular mechanism underlying this high OxPhos signature has never been fully elucidated and this is the major novelty of our manuscript. The manuscript by Qu et al. reported on the phosphoproteomic profiling of 74 tRCC but did not provide experimental analysis showing evidence that TFE3 mechanistically activates oxidative phosphorylation through binding to its targets.

-We report for the first time a fully integrated genomic profiling for 2 TFE3 fusion proteins in 4 tRCC cell lines. This has never been previously reported.

-We define a core set of TFE3-fusion target genes with E/M-box motifs in their proximal promoters, common to 4 different tRCC cell lines and 2 distinct fusions, including numerous genes involved in OxPhos. These are completely novel observations.

-We show that TFE3 fusions bind promiscuously to active promoters and generally stimulate mRNA synthesis, a phenomenon previously related to oncogenic transformation in other contexts. While this was documented for MYC it was never previously reported for TFE3 fusion proteins.

We perform genomic profiling for 2 TFE3 fusion proteins as well as overexpressed native TFE3 in a heterologous cell line reproducing many of the observations (promiscuous promoter binding, activation of OxPhos genes). No such experiments have been previously reported.

All of the above observations are novel and provide the most comprehensive framework to date for understanding TFE3 fusion functions and in particular how they activate OxPhos. We are surprised and disappointed that referees 1 and 2 make no consideration of these original findings.

The involvement of EMT in tRCC tumorigenesis was already discussed in Reference 25 (Sun et al. 2021) and Prakasam et al. 2024 (doi.org/10.1172/JCI170559).

We fully agree with this comment, however the Sun et al study was limited to analyses of bulk RNA-seq and concluded that ASPRC1-TFE3 fusions had propensity to undergo EMT. In our study, we highlight that EMT is also commonly observed with other TFE3 fusions, not just limited to ASPRC1 fusions. More importantly, we additionally show that EMT is associated with presence of myCAFs whose enrichment strongly correlates with poor patient survival. Again, these are novel observations and are of important clinical relevance.

Therefore, the findings described in this manuscript should be placed in context of previous discoveries already reported in the literature in the Introduction and/or Discussion sections.

We raised these issues in the discussion section of the paper (see for example page 23) and we can easily add the additional more recent publication from Prakasam, that deals with EMT, but does not mention OxPhos.

In addition, a preprint posted shortly after the submission of this manuscript (doi.org/10.1101/2024.08.09.607311) might also be discussed.

Here the referee refers to the manuscript by our colleagues from the Viswanathan lab. We mentioned the existence of this study to you in our cover letter. We do not know if it is EMBO Mol Med policy to cite preprints, but we would be happy to do so. As we previously noted in our cover letter, this manuscript, that arrives at very similar conclusions to ours concerning how TFE3 fusions activate OxPhos, is in favorable revision in Nature Metabolism. Obviously, the mechanistic insight into how TFE3 fusions active OxPhos was in this case considered original.

'Only cell lines in vitro and several patient samples.'

The referee implies that we used several patient samples. In fact, we report a novel cohort of 38 patient samples and confirmed the observations in a second public cohort of 55 patient, this cannot be qualified as 'several'.

We also take issue with some of the comments of Referee 1

First of all, there are a number of factual errors in the comments of referee 1. For example, even the first comment '*Supplemental Figure 1B should include fractionation controls*' is immediately factually incorrect as BRG1 is shown as a control for the nuclear fraction and GAPDH for the cytoplasmic fraction. However, more importantly, are the comments in 5 and 6.

'In Fig 4D, the authors are comparing Ox Phos score of tRCC to ccRCC but the more relevant comparison should be with NAT, is the signature higher in tRCC in this scenario?'

This data was indicated in the figure and it will be easy to add the statistical comparison, but it can already be clearly seen that the tRCC patients have OxPhos signatures comparable to the NAT, which is atypical of tumours that almost always show reduced OxPhos signature due to the increased glycolysis associated with transformation. The behavior of tRCC is rather unique in this respect. To consolidate these observations, we compared tRCC and ccRCC, but not comparing bulk signatures, but by stratifying tumours in both diseases based on their EMT status. This analyses in Fig.4D illustrates very spectacularly that the strong reduction in OxPhos seen upon EMT in ccRCC is not seen in tRCC. These novel and comprehensive analyses surpass all previous reported analyses.

The study has conducted a well-executed, in -depth analysis of transcriptional signatures of TFE3 fusion protein containing cell lines and samples from patients. However, the Ox-Phos related gene cluster upregulated in response to the TFE3-'NONO and PRCC fusions do not appear to be consistent with the patient data, where the Ox-Phos data is prominent for the non-NONO and non-PRCC cluster 3'

The referee is factually incorrect by saying that OxPhos is highest in the cluster 3. First, OxPhos is higher in clusters 1 and 2 that have lower EMT than in cluster 3. Second, cluster 3 also comprises PRCC-TFE3 fusions. What Figure 4D shows is that high OxPhos score is common to all TFE3-fusions, but varies upon EMT status (see also paragraph above). This is in no way inconsistent with our analyses. We never suggested that activation of OxPhos genes was a specific property of only NONO- or PRCC-fusions. Quite the contrary, we argued our observations revealed a mechanism that is common to all fusions and even to overexpressed native TFE3.

'So, although the story came together very well with Ox-Phos and ferroptosis related gene signatures being upregulated in cell lines, the patient data does not directly support the cell line data.'

We totally disagree with this comment. The patient data in Fig.4 shows that the OxPhos signature is high in tRCC tumours and is maintained during EMT. We can easily add an additional heatmap that further shows the increased expression of the ferroptosis signature genes in the patient data. Therefore, in contrast to the claims of the referee, the cell line data showing that TFE3-fusions activate OxPhos and ferroptosis is fully supported by the patient data.

Referee 1 was also concerned by the fact that the manuscript was rather long, and referee 2 that we had no data on targeting tRCC cells with the novel potential targets that we identified. We agree with them that the derivation of the 10 gene signature and its potential drugability is the most speculative aspect of the study. We think that this can be removed without compromising the novelty of our study that is focused on the mechanism by which TFE3 and its fusion proteins active OxPhos.

In summary, we can rapidly respond to a large majority of the referee's comments and perform the additional analyses and experimentation that they suggest. Both referees 1 (*Although exciting and much needed in the field of tRCC.. The study has conducted a well-executed, in -depth analysis of transcriptional signatures*) and 3 (*This is an elegant study that further dissects the oncogenic role of TFE3 fusion proteins in tRCC*) recognized quality of our work, even referee 2 concedes that (*Overall, the manuscript is well written (despite some typos) and most*

experiments were carefully controlled. The authors conducted well-designed in vitro experiments...).

Given the above considerations, we would kindly ask you if you would reconsider the decision to reject and allow us to respond with a revised version of our study that can be readily completed within a couple of months. We believe that we would be able provide a point by point response to three reviewers and provide the experimental data they propose.

With gratitude

Regards

Professor Gabriel Malouf and Dr Irwin Davidson

1st Oct 2024

Dear Prof. Malouf,

Thank you for your e-mail asking us to reconsider our decision on your manuscript and for providing a detailed appeal letter. I have carefully read it and discussed it with my colleagues here.

Initially, the referees agreed on the potential interest of the findings but nevertheless raised a number of serious concerns and stated that extensive revisions would be needed. In line with EMBO policy to encourage a single round of revisions in a limited timeframe, we decided to reject the manuscript.

In your appeal letter, you mentioned that you could "rapidly respond to a large majority of the referee's comments and perform the additional analyses and experimentation they suggest".

After further discussion within the team, we agreed that if you were to address all referees' concerns in a reasonable time frame, you may wish to submit a revised version of your manuscript.

Please note that the manuscript will be re-reviewed and that we cannot guarantee at this stage that the eventual outcome will be favorable. We would thus encourage you to adequately address ALL concerns raised by the referees. Acceptance or rejection of the manuscript will depend on the completeness of your responses included in the next, final version of the manuscript. For this reason, and to save you from any frustrations in the end, I would strongly advise against returning an incomplete revision. Please attach a covering letter giving details of the way in which you have handled each of the points raised by the referees.

We are expecting your revised manuscript within three months, if you anticipate any delay, please contact us.

We require:

4) A .docx formatted letter INCLUDING the reviewers' reports and your detailed point-by-point responses to their comments. As part of the EMBO Press transparent editorial process, the point-by-point response is part of the Review Process File (RPF), which will be published alongside your paper.

5) A complete author checklist, which you can download from our author guidelines (<https://www.embopress.org/page/journal/17574684/authorguide#submissionofrevisions>). Please insert information in the checklist that is also reflected in the manuscript. The completed author checklist will also be part of the RPF.

6) All Materials and Methods need to be described in the main text using our 'Structured Methods' format. According to this format, the Methods section includes a Reagents and Tools Table (listing key reagents, experimental models, software and relevant equipment and including their sources and relevant identifiers) followed by a Methods and Protocols section describing the methods, ideally using a step-by-step protocol format. The aim is to facilitate adoption of the methodologies across labs. Please download and fill our Reagents and Tools Table template (.docx), which you can find in our author guidelines: <https://www.embopress.org/page/journal/14693178/authorguide#structuredmethods>.

<https://www.embopress.org/doi/10.15252/msb.20178071>

7) It is mandatory to include a 'Data Availability' section after the Materials and Methods. Before submitting your revision, primary datasets produced in this study need to be deposited in an appropriate public database, and the accession numbers and database listed under 'Data Availability'. Please remember to provide a reviewer password if the datasets are not yet public (see <https://www.embopress.org/page/journal/17574684/authorguide#dataavailability>).

8) For data quantification: please specify the name of the statistical test used to generate error bars and P values, the number (n) of independent experiments (specify technical or biological replicates) underlying each data point and the test used to calculate p-values in each figure legend. The figure legends should contain a basic description of n, P and the test applied. Graphs must include a description of the bars and the error bars (s.d., s.e.m.). Please provide exact p values.

12) Author contributions: CRedit has replaced the traditional author contributions section because it offers a systematic machine readable author contributions format that allows for more effective research assessment. Please remove the Authors Contributions from the manuscript and use the free text boxes beneath each contributing author's name in our system to add specific details on the author's contribution. More information is available in our guide to authors.

13) Disclosure statement and competing interests: We updated our journal's competing interests policy in January 2022 and request authors to consider both actual and perceived competing interests. Please review the policy <https://www.embopress.org/competing-interests> and update your competing interests if necessary.

14) Every published paper now includes a 'Synopsis' to further enhance discoverability. Synopses are displayed on the journal webpage and are freely accessible to all readers. They include a short stand first (maximum of 300 characters, including space) as well as 2-5 one-sentences bullet points that summarizes the paper. Please write the bullet points to summarize the key NEW findings. They should be designed to be complementary to the abstract - i.e. not repeat the same text. We encourage inclusion of key acronyms and quantitative information (maximum of 30 words / bullet point). Please use the passive voice. Please attach these in a separate file or send them by email, we will incorporate them accordingly.

Please also suggest a visual abstract to illustrate your article as a PNG file 550 px wide x 300-600 px high. A cropped portion of this image will serve as thumbnail for the table of content on our webpage.

15) As part of the EMBO Publications transparent editorial process initiative (see our Editorial at <http://embomolmed.embopress.org/content/2/9/329>), EMBO Molecular Medicine will publish online a Review Process File (RPF) to accompany accepted manuscripts.

In the event of acceptance, this file will be published in conjunction with your paper and will include the anonymous referee reports, your point-by-point response and all pertinent correspondence relating to the manuscript. Let us know whether you agree with the publication of the RPF and as here, if you want to remove or not any figures from it prior to publication. Please note that the Authors checklist will be published at the end of the RPF.

I look forward to receiving your revised manuscript.

Yours sincerely,

Lise Roth

We thank the referees for their constructive comments and we provide here a detailed response to the issues raised.

Referee #1 (Remarks for Author):

The manuscript by Helleux et al is a comprehensive transcriptional evaluation of tRCC. The abundant amount of work and data analysis to answer key questions is clear in this study. Although exciting and much needed in the field of tRCC, there is a lot of data presented which makes the final manuscript less clear and focused. This manuscript appears to read as two stand-alone papers, and the authors should consider the necessity of publishing this as a combined, massive manuscript.

We would like to thank reviewer 1 for his/her positive comment on our manuscript regarding the amount of data generated and analysis done. We felt that it would be important to connect the mechanistic part of the study with validation using patients' samples. We have done lots of efforts in connecting both parts via the focus on the implication of ferroptosis pathway and shortening the text. We are confident that the current version has been improved and we are grateful for reviewer 1 for his suggestions.

The following comments need to be addressed:

1. Supplemental Figure 1B should include fractionation controls.

The fractionation controls were already present on this blot as explained in the results section of the text on page 5 of the original version. BRG1 and GAPDH were used as nuclear and cytoplasmic markers respectively. BRG1 is a subunit of the SWI/SNF complex involved in chromatin remodeling and GAPDH is an essential enzyme in the glycolysis process which takes place in the cytoplasm. We have modified this text to better clarify the controls used.

2. When you overexpress TFE3 (Fig 2B), what is the additional band that is observed almost at the same molecular weight as the PRCC-TFE3 size? Is overexpression of TFE3 inducing a fusion event in HEK cells?

Indeed, we noted that in cells expressing native TFE3 with a molecular mass of around 60 kDa, an additional slower migrating species were observed around 80-90 kDa. These additional species may be generated by post-translational modification of TFE3, including SUMOylation. This is indicated in the data sheet of the antibody and was previously reported (Liu et.al., Clin. Transl. Med., 2022, doi: [10.1002/ctm2.797](https://doi.org/10.1002/ctm2.797) ; Miller et.al., J. Biol. Chem., 2005, doi: [10.1074/jbc.M411757200](https://doi.org/10.1074/jbc.M411757200)). This slower migrating sumoylated species is also weakly seen in Fig. S1A in the ccRCC lines, but is more readily observed after ectopic native TFE3 expression in HEK293T cells where the signal for TFE3 is much stronger (Fig. 2B). We further note that analyses of the RNA-seq from these cells data did not reveal any evidence of a novel translocation event giving rise to a TFE3 fusion protein in these cells.

3. What were the ox-Phos related genes exclusively bound by either NONO-TFE3 or PRCC-TFE3 i.e. not common but those that were differentially expressed genes.

The great majority of OxPhos genes comprise a TFE3 binding site within 30kB around their TSS, often in the proximal promoter. Analysis of peaks exclusively found in either NONO-TFE3 expressing lines (UOK109, TF1) or PRCC-TFE3 expressing lines (UOK120, UOK146) reveals only 7 OxPhos genes (ACO2, COX6C, COX8A, FXN, MTRR, NDUFA8 et UQCRH) specific to NONO-TFE3 and only one gene (COX17) specific to PRCC-TFE3. Hence, the vast majority (145) of OxPhos genes are commonly bound in all tRCC 4 lines. However perhaps the referee is referring to the genes selectively activated in the HEKT cells following ectopic expression. For example, the 839 genes selectively up-regulated by NONO-TFE3 are enriched in BP_FAT terms associated with neurogenesis and calcium signaling where the 704 genes selectively up-regulated by PRCC-TFE3 are enriched in different catabolic processes. As we did not investigate the functional significance of these observations, they are not included in this manuscript.

4. Is the impact of GPX4 inhibitor greater (i.e., is there enhanced sensitivity) in Nono-TFE3 cell lines versus PRCC-TFE3?

As shown in Fig. 3F, UOK109, TF1, both NONO-TFE3 cell lines, UOK120 and UOK146, both PRCC-TFE3 cell lines, present similar sensitivity to RSL3, a GPX4 inhibitor with IC50 of 2.1, 58.8, 12.3, and 27.5 nM respectively. TF1 is slightly more resistant probably due to the fact that this is a primary cell line from a patient previously treated with sunitinib and that had developed resistance to this drug.

5. In Fig 4D, the authors are comparing Ox Phos score of tRCC to ccRCC but the more relevant comparison should be with NAT, is the signature higher in tRCC in this scenario?

We agree with the referee. We have now added a new Supplemental Dataset (Dataset S6) where we make all of the pairwise comparisons between the ccRCC, tRCC and NAT. We performed Wilcoxon test and found that when comparing tRCC clusters with NAT, there is no significant difference in OxPhos score between tRCC.C1 or tRCC.C2 and NAT. OxPhos score is however significantly higher in NAT than in tRCC.C3 (pval=0.0223, *). This consolidates the idea that tRCC samples present comparable OxPhos score to normal adjacent tissue and only the tumours with highest EMT (tRCC.C3) showed a reduced OxPhos compared to NAT. The OxPhos score of the tRCC.C3 group is however much higher than the equivalent ccRCC group.

6. The study has conducted a well-executed, in -depth analysis of transcriptional signatures of TFE3 fusion protein containing cell lines and samples from patients. However, the Ox-Phos related gene cluster upregulated in response to the TFE3-NONO and PRCC fusions do not appear to be consistent with the patient data, where the Ox-Phos data is prominent for the non-NONO and non-PRCC cluster 3. So, although the story came together very well with Ox-Phos and ferroptosis related gene signatures being upregulated in cell lines, the patient data does not directly support the cell line data.

We do not fully understand the comment of the referee here. At no point in the manuscript did we state, or wish to imply, that activation of the OxPhos program was a specific function of only the NONO- or PRCC fusions. In fact, in the discussion, we rather put forward the idea we describe general properties common to all TFE3 fusions. To underline this, we include a novel Supplemental Dataset (Dataset S6) where we compare the OxPhos score of the PRCC-TFE3 or the NONO-TFE3 fusions to the other fusions in each cluster. Comparing the OxPhos scores showed no significant difference between those of the NONO-TFE3 and the PRCC-TFE3 fusions with those of the other fusions in each cluster. Thus, in cluster 3 for example, ASPSCR1-TFE3 fusions have similar OxPhos scores to the other fusions of this cluster. These data support the idea that all the TFE3 fusions have similar abilities to activate the OxPhos program and that they are similarly affected by EMT. These data are further consolidated by the pre-print manuscript of Li et al., <https://doi.org/10.1101/2024.08.09.607311>, cited and discussed in the revised version, describing genomic profiling of the ASPSCR1-TFE3 fusion and its ability to directly drive the OxPhos program. We do not see how this is contrary to the cell line data, it is rather an extension of the cell line data consolidating the idea that activation of OxPhos program is a common property of all TFE3 fusions. However, as this issue was raised several times by the referee, we have modified the discussion to make this point more clearly.

7. The comparisons to ccRCC need to be rationalized and explicitly explained.

In the original version of the text, we explained that

“To compare the tRCC OxPhos scores to those of ccRCC, we stratified ccRCC tumours based on our previously reported single cell RNA-seq signatures that defined an EMT gradient associated with poor survival and an OxPhos to glycolysis switch”.

We made this stratification as we reasoned that we could not compare stratified tRCC patients with non-stratified ccRCC patients that represent a heterogeneous collection of tumours with different EMT and metabolic states. To make this clearer we have modified the text to read

“We compared the tRCC OxPhos scores to those of ccRCC. So as not to compare stratified tRCC patients with non-stratified ccRCC patients that represent a heterogeneous collection of tumours with different EMT and metabolic states, we stratified ccRCC tumours based on our previously reported single cell RNA-seq signatures that defined an EMT gradient associated with poor survival and an OxPhos to glycolysis switch. Based on deconvolution of the TCGA-KIRC collection with the ccRCC tumour cell signatures, we selected 20 tumours of the ccRCC.epi (epithelial), ccRCC.int (intermediate) and ccRCC.mes (mesenchymal) states and scored their OxPhos signatures (**Fig. 4D**)”

By doing this we have been able to make a much more meaningful comparison between the ccRCC and tRCC tumour sets and the NAT. As mentioned above we have now added a new Supplemental data set that presents the full statistics of all of the pairwise comparisons shown in Fig 4D.

8. There is no direct evidence suggesting that elevated Ox-Phos in tRCC is driving oncogenesis or promoting progression.

We agree with the referee that we provided no direct evidence that elevated OxPhos drives tumorigenesis even if, as we cited in the Discussion, there are many previously published studies showing how OxPhos promotes tumorigenesis and tumour progression. Nevertheless, a pre-print manuscript by Li et al., <https://doi.org/10.1101/2024.10.24.620074> reported that numerous genes involved in mitochondrial function and OxPhos, in particular components of the electron transport chain that we define here as direct TFE3 targets were selective vulnerabilities of tRCC. These data are complementary to those shown here and strongly consolidate the idea that TFE3-fusions directly activate OxPhos genes that are essential for tRCC tumorigenesis. This has now been clarified in the discussion.

The same is true for ferroptosis, in fact there is no ferroptosis signature in patient samples. Additional experimentation will be required to make these claims, and the discussion should accordingly be modified to reflect this.

The referee is correct to point out that the ferroptosis pathway did not stand out in the ontology analyses of the patient transcriptomic data presented in Fig. S9D of the original manuscript. To address this issue, we defined two ferroptosis signatures based on literature screening, one encompassing anti-ferroptosis genes (signature of 57 genes) and comprising for example those involved in glutathione metabolism, GCLC; GCLM, the X_c system, and the other comprising genes (signature of 37 genes) that promote ferroptosis, such as the ASCL1-6 that catalyze beta-oxidation of lipids or TFRC the transporter that imports iron into the cell to promote ferroptosis. We then assessed the expression of these genes in the NAT compared to tRCC clusters. As can be seen from the heatmap now included in the revised manuscript as Fig. 6A, it is striking that many key genes involved in ferroptosis resistance are upregulated in the tRCC compared to NAT (GPX4, GCLM, GCLC, SLC7A2, SLC7A11, TXNRD1), whereas those involved in promoting ferroptosis are repressed (TFRC, ASCL3, 4 and 6). These data therefore showed that tRCC tumorigenesis was accompanied by increased expression of ferroptosis resistance genes and repression of critical mediators of ferroptosis. This data has been included in the revised version along with novel data showing enhanced generation of oxidized glutathione in a RSL3-repressible manner in HEKT cells expressing ectopic fusion proteins and diminished generation of oxidized glutathione in tRCC cells silenced for TFE3, consistent with the preprint of Li et al cited above (Fig. 6D-E). Altogether, this new data strongly supports a role for direct TFE3-fusion regulation of glutathione metabolism and ferroptosis in tRCC.

9. This manuscript is comprehensive and feels like two manuscripts. The data at this juncture will benefit from simply addressing what happens when you have the NONO and PRCC fusion proteins and when you have non-NONO and PRCC fusion proteins.

As explained above, at no point in the manuscript did we state, or wish to imply, that activation of the OxPhos program was a specific function of only the NONO- or PRCC fusions. In fact, in the discussion, we rather put forward the idea that we describe general properties common to all TFE3 fusions. As mentioned above, the new analyses and literature show that activation of the OxPhos program is not specific to NONO-TFE3 or PRCC-TFE3, but a general property of TFE3 fusions. This is further supported by the observation of Li et al mentioned above that the mitochondrial OxPhos genes are vulnerabilities of both UOK109 expressing NONO-TFE3 and the 2 cell lines expressing ASPSCR1-TFE3. Thus, we do not see where the misunderstanding between the “the NONO and PRCC fusion proteins and when you have

non-NONO and PRCC fusion proteins” comes from. However, as explained above we have modified the discussion to make this point more clearly.

10. The motif/element that the fusion proteins bind to does not appear to make a difference since non-NONO and PRCC patient samples showed maximal OX-PHOS gene signatures.

Again, the referees comment implies the same misunderstanding as above, we propose that activation of the OxPhos program is a general property of all TFE3 fusions. It is likely that as all TFE3 fusions comprise the TFE3 DNA binding domain, they share a large set of common sites and it is likely that TFE3 fusions activate their target genes irrespective of whether they are bound via the E/M-box motif or are indirectly tethered via other transcription factors. This is further supported by the pre-print of Li et al as mentioned above, but also by the results of Sicinska et al who characterized genomic binding of ASPSCR1-TFE3 in soft part alveolar sarcoma (PMID 38657118). In this study, they performed HA-ChIP-seq in sarcoma cells ectopically expressing a HA-tagged ASPSCR1-TFE3 fusion protein finding that it binds broadly to active chromatin sites and is enriched at proximal promoters. Hence, these data together show that promiscuous promoter binding is a common property of TFE3 fusion proteins shared in both tRCC and sarcomas. As mentioned above, our results go one step further reporting that the consequence is a general stimulation of mRNA synthesis, a phenomenon previously shown to contribute to oncogenic transformation. We have discussed these issues in the revised version of the manuscript.

11. The discussion is too lengthy and needs to be significantly shortened.

We agree with this remark and have shortened the discussion accordingly. In particular in response to the comments of referee 2, we have completely deleted the section on the gene signatures and their potential as therapeutic targets. This was the most speculative aspect of the paper, although we note that some of these genes represented cell-specific vulnerabilities (see below).

Moreover, to address the comment of the referee on length and the impression that it is 2 manuscripts, we have reorganized the order or presentation to separate the data on OxPhos from that concerning ferroptosis (new Fig. 6), with the new data that has been added (see above and below). We believe this provides a more coherent and logical format.

Referee #2 (Remarks for Author):

The manuscript by Helleux et al. reports that 'TFE3 fusions drive oxidative metabolism and ferroptosis resistance in translocation renal carcinomas'. The authors conducted well-designed in vitro experiments to prove the necessity and sufficiency of translocation TFE3 to better understand the tumorigenesis of translocation renal cell carcinomas (tRCCs). Firstly, the oxidative phosphorylation pathway was found to be enriched both by silencing TFE3 in TFE3-translocated cell lines and by expressing translocated TFE3 in normal kidney HEK293T cells. These results were validated by assessing the Oxygen Consumption Rate (OCR). The authors then performed state-of-the-art

Cleavage Under Targets and Tagmentation (CUT&Tag) in the tRCC cell lines along with H3K27ac ChIP-Seq to identify active cis-regulatory elements of translocation TFE3. As expected, the M/E-box motif was highly overrepresented, followed by SP1/KLF and NFY motifs, in particular in NONO-TFE3 fusion cells. Since the ferroptosis pathway was overrepresented in the promoter occupancy at M/E-box sites of TFE3 fusion proteins, the tRCC cell lines were treated with the GPX4 inhibitor RSL3 and showed a sensitivity that was blocked by the ferroptosis inhibitor ferristatin-1. Furthermore, analyses of tRCC patient samples confirmed the overrepresentation of the oxidative phosphorylation pathway, besides the identification of epithelial to mesenchymal transition (EMT) and myofibroblastic cancer-associated fibroblasts (myCAFs), which were associated to poor patient survival. Finally, a 10-gene signature of TFE3-fusion regulated genes was proposed, which was correlated with poor patient overall survival. Overall, the manuscript is well written (despite some typos) and most experiments were carefully controlled. Conclusions are mostly supported by the data presented. However, there are several concerns that need to be addressed before the manuscript could be accepted for publication, as follows:

- The novelty of several findings reported in this study seem to be compromised by some recent publications. For instance, the involvement of oxidative phosphorylation metabolism and ferroptosis (glutathione metabolism) in the tumorigenesis of tRCC were previously described in Reference 45 (Qu et al. 2022).

The involvement of EMT in tRCC tumorigenesis was already discussed in Reference 25 (Sun et al. 2021) and Prakasam et al. 2024 (doi.org/10.1172/JCI170559).

Therefore, the findings described in this manuscript should be placed in context of previous discoveries already reported in the literature in the Introduction and/or Discussion sections. In addition, a preprint posted shortly after the submission of this manuscript (doi.org/10.1101/2024.08.09.607311) might also be discussed.

We would like to thank reviewer 2 for his comments and suggestions to improve our manuscript. We agree with the referee that the observation that tRCC tumours display a high OxPhos signature is not novel and indeed, Qu et.al. paper (as well as others we cited in our manuscript) had already underlined these conclusions. We did not claim that our study was novel in this respect. The novelty of our study is the insight it brings on the molecular mechanisms underlying this OxPhos signature. The manuscript by Qu et al. reported the phosphoproteomic profiling of 74 tRCC but did not provide experimental evidence that TFE3 directly activates the OxPhos program. Here, we report for the first time a fully integrated genomic profiling which allowed us to define a core set of TFE3-fusion target genes with E/M-box motifs in their proximal promoters, common to 4 different tRCC cell lines and 2 distinct fusions, including numerous genes involved in OxPhos. In addition, we confirmed these observations in parallel gain-of-function experiments showing activation of OxPhos genes and OxPhos activity levels through direct binding of ectopically expressed TFE3 fusion proteins. All of these data are novel, have never before been reported and provide the most comprehensive framework to date for understanding how TFE3 fusions activate the OxPhos program and more generally gene expression in tRCC.

Concerning the Sun et.al. study, we fully agree with the referee's comment, however this analysis was limited to bulk RNA-seq and concluded that ASPSCR1-TFE3 fusions had propensity to undergo EMT. In our study, we highlight that EMT is also commonly observed with other TFE3 fusions, not just ASPSCR1 fusions. More importantly, we additionally show that EMT is associated with presence of myCAFs whose enrichment strongly correlates with poor patient survival. Again, these are novel observations and are of important clinical relevance.

We agree with the referee on the missing citation to Prakasam et.al. and this has been added to the revised version. Similarly, we now cite and discuss the pre-print manuscript of Li et al <https://doi.org/10.1101/2024.08.09.607311> that describe genomic profiling of TFE3 fusion proteins and activation of OxPhos analogous to those described here as well as a second recent pre-print manuscript by Li et al., <https://doi.org/10.1101/2024.10.24.620074> reporting that numerous genes involved in mitochondrial function and OxPhos, in particular components of the electron transport chain that we define here as direct TFE3 targets, were selective vulnerabilities of tRCC. These data are complementary to those shown here and strongly consolidate the idea the TFE3-fusions directly activate OxPhos genes that are essential for tRCC tumorigenesis.

- This study performed sophisticated CUT&Tag experiments not present in similar publications in the literature to identify active cis-regulatory elements of TFE3 translocation. It is a pity that, besides the well-known M/E-box motif, the molecular characterization, validation and functional significance of novel motifs is very limited.

We are not sure what the referee is asking here. We report for the first time a fully integrated genomic profiling for 2 TFE3 fusion proteins in 4 tRCC cell lines. Through these binding experiments, we define a core set of TFE3-fusion target genes with E/M-box motifs in their proximal promoters, common to 4 different tRCC cell lines and 2 distinct fusions. We show that TFE3 fusions bind promiscuously to active promoters and generally stimulate mRNA synthesis, a phenomenon previously related to oncogenic transformation in other contexts. In that sense the functional significance of novel motifs (GC-rich and NFY) lies in the fact that they are prevalent in core promoters and provide a link with the increased levels of mRNA synthesis that we observe. Our data are analogous to the results of Sicinska et al who characterized genomic binding of ASPSCR1-TFE3 in soft part alveolar sarcomas (PMID 38657118). In this study, they performed HA-ChIP-seq in sarcoma cells ectopically expressing a HA-tagged ASPSCR1-TFE3 fusion protein finding that it binds broadly to active chromatin sites and is enriched at proximal promoters. While these findings are analogous to ours, *stricto-senso*, it could have been argued that this was a consequence of ectopic overexpression or a specific property of the ASPSCR1-TFE3 fusion associated with more aggressive tumours. Nevertheless, we make the same observations on the endogenous NONO-TFE3 and PRCC-TFE3 proteins in tRCC cells using a different technique (Cut&Tag vs ChIP-seq). Hence, these data together show that promiscuous promoter binding is a common property of TFE3 fusion proteins shared in both tRCC and soft part alveolar sarcomas. As mentioned above, our results go one step further reporting that the consequence of this is a general stimulation of mRNA synthesis, a phenomenon previously shown to contribute to oncogenic transformation. We have discussed these issues in the revised version of the manuscript.

- Ferroptosis is highlighted in the title, abstract and throughout the manuscript, but the evidence presented to support the involvement of this pathway in tRCC tumorigenesis is limited to the promoter occupancy at M/E-box sites of TFE3 fusion proteins and the *in vitro* response to ferroptosis drugs.

The RNA-Seq of gain and loss-of-function experiments for the ferroptosis genes should be compared.

We agree with the referee and this issue was addressed in response to the comments of referee 1. As mentioned above, we defined two ferroptosis signatures based on literature screening, one encompassing ferroptosis resistance genes (signature of 57 genes) and comprising for example those involved in glutathione metabolism, GCLC; GCLM, the Xc system, and the other comprising genes (signature of 38 genes) that promote ferroptosis, such as the ASCL1-6 that catalyze beta-oxidation of lipids or TFRC the transporter that imports iron into the cell to promote ferroptosis. We then assessed the expression of these genes in the NAT compared to tRCC clusters. As can be seen from the heatmap now included in the revised manuscript as Fig. 6A, it is striking that many key genes involved in ferroptosis resistance are upregulated in tRCC compared to NAT (GPX4, GCLM, GCLC, SLC73A2, SLC7A11, TXNRD1), whereas those involved in promoting ferroptosis are repressed (TFRC, ASCL3, 4 and 6). These data therefore showed that tRCC tumorigenesis was accompanied by increased expression of ferroptosis resistance genes and repression of the critical mediators of ferroptosis. We also compared the expression of these genes in the HEKT cells after Dox stimulation. While the results are less clear cut than with the patients, the analyses show that the ectopic fusion proteins (in particular PRCC-TFE3 that is best expressed in this model system) activate the ferroptosis resistance genes GPX4, GCLC, SLC73A2, SLC7A11, TXNRD1, whereas ASCL4 and TFRC are repressed (Fig. 6C). These data therefore showed that tRCC tumorigenesis was accompanied by increased expression of ferroptosis resistance genes and

repression of critical mediators of ferroptosis. This data has been included in the revised version along with novel data showing enhanced generation of oxidized glutathione in a RSL3-repressible manner in HEKT cells expressing ectopic fusion proteins and diminished generation of oxidized glutathione in tRCC cells silenced for TFE3, consistent with the preprint of Li

et al., cited above. (Fig. 6D-E). Altogether, this new data strongly supports a role for direct TFE3-fusion regulation of glutathione metabolism and ferroptosis in tRCC.

The treatments with RSL3 and ferristatin-1 should also be performed on the tRCC cell lines with TFE3 silencing and HEK cells with TFE3 overexpression to rule out the involvement of TFE3 translocation. In addition, similarly to OCR assays to validate oxidative phosphorylation, it would be necessary to show some ferroptosis markers, such as deregulated glutathione metabolism and/or lipid peroxidation, in the gain and loss-of-function in vitro experiments to claim involvement of ferroptosis in tRCC tumorigenesis.

As mentioned above we have performed GSSG-GSH assays showing that TFE3 and its fusion proteins can modulate glutathione metabolism in cell culture, but also analyses of the patient transcriptomic data that shows acquisition of a ferroptosis resistance expression program upon tumorigenesis. As with the non-transfected cells, tRCC cells that were treated with siTFE3 and RSL3 presented higher cell death, while the viability of both siCTR and siTFE3 cells was mildly increased by 1mM ferrostatin-1 (see adjacent Figure showing the results of clonogenic assays after 8 days of culture). There hence seems to be a low basal level of ferroptosis in the tRCC cells, however the effects are mild and while we include the data

here for the referee, given the novel data now presented in Fig. 6, we did not include them in the revised manuscript.

We also performed treatment with RSL3 and ferrostatin-1 in the HEK cells expressing ectopic TFE3-fusions. However, although little effect was seen at 24 hours, after 48 and 72 hours of Dox treatment, the ectopic TFE3-expressing cells showed high levels of

senescence (see adjacent Figure with Senescence-associated beta galactosidase staining) associated with induced expression of P21 (CDKN1A, see adjacent Figure of immunoblot after 48 hours) and growth arrest. As the cells arrest their proliferation, we could not perform meaningful proliferation or survival experiments with them.

- Several potential therapeutic targets for tRCC are speculated in the Discussion of this study. At least, some in vitro and, preferentially, pre-clinical evidence with models of tRCC would be desirable to be conducted to show the medical feasibility

of this study.

We agree with the referee that this was the most speculative aspect of the study. As mentioned above in response to referee 1, we decided to remove these data from the manuscript and we reorganized the text to make a more coherent story. Nevertheless, as requested by the referee, we tested the function of 3 genes from the 10 gene signature in the tRCC cell lines by performing their siRNA silencing.

As can be seen in the adjacent figure, we found that siRNA silencing of UPP1 had a strong detrimental effect on viability of the UOK109 cells in a colony forming assay and more

limited effect on the UOK146 cells. SiCORO1C impacted viability of the UOK146 and UOK120 cells and siBLOC1S3 impacted viability of UOK146 cells.

As can be seen from these results, none of these genes are common vulnerabilities in all of the tested tRCC cell lines, however they are cell-specific vulnerabilities. It should also be noted that in the recent pre-print of Li et al cited above, some of the vulnerabilities they identified also displayed cell type specificity such as MDM2 that strongly impacted viability of the UOK109 and UOK146 cells, but not of the cell lines with ASPSCR1-TFE3 fusions.

Thus, while our data validate the idea that some of these genes represent vulnerabilities in tRCC, we agree with the referee that this aspect of the study will require a more in-depth investigation and we have thus removed this data from the current manuscript.

- The OCR assays should also be performed on the tRCC cell lines with TFE3 silencing and ccRCC cell lines with TFE3 silencing to further evaluate the involvement of TFE3 in the basal and maximal respiration levels.

In the original version of the manuscript (Fig. 1I), we showed that siTFE3 reduced the maximal but not basal OCR levels in tRCC cells. These observations are highly similar to those seen also with UOK109 cells and sgRNAs in Li et al (<https://doi.org/10.1101/2024.08.09.607311>) and to those of the ASPSCR1-TFE3 lines used in the above study. Moreover, in their preprints cited above, Li et al., reported that TFE3 knockdown had no effect on OxPhos or other aspects of metabolism in ccRCC cells and reported that while TFE3 is a vulnerability in tRCC cells it is not essential in ccRCC cells.

- Statistics need to be revised. The Dunnett's multiple comparison test compares the means from several experimental groups against only one control group, whereas the Wilcoxon test compares the means of only two groups. Unfortunately, the text to the Fig. 4D in the Results section describes comparisons not tested for some groups. Therefore, the Tukey post hoc test should be more appropriate for comparing means among all the experimental groups after ANOVA. In addition, the authors claim that TF1 showed "higher basal, but not maximal OCR levels (Fig. 1H)", but differences in basal respiration of TF1 show no significance.

We apologize for the discrepancies between the text in results section and the actual comparison represented on Fig. 1H and Fig. 4D. This has been modified in the revised text.

We used Dunnett comparison to compare the means of each tRCC to the group of ccRCC in the OCR experiments of Fig. 1H, whereas we used Wilcoxon to perform the pairwise comparisons of the OxPhos scores in Fig 4D.

In addition, we have now added a new Supplemental Dataset (Dataset S6) where we make all of the pairwise comparisons between the ccRCC, tRCC and NAT. We performed Wilcoxon test and found that when comparing tRCC clusters with NAT, there is no significant difference in OxPhos score between tRCC.C1 or tRCC.C2 and NAT. OxPhos score is however significantly higher in NAT than in tRCC.C3 (pval=0.0223). This consolidates the idea that tRCC samples present comparable OxPhos score to normal adjacent tissue and only the tumours with highest EMT (tRCC.C3) showed a reduced OxPhos compared to NAT. The OxPhos score of the tRCC.C3 group is however much higher than the equivalent ccRCC group.

- Information regarding the ethics approvals and the informed patient consents for the in-house tRCC patient samples must be provided. In addition, a file with the individual clinicopathological characteristics for each patient from the in-house cohort and TCGA cohort (summarized in Fig. S9A) should be provide as supplementary dataset.

We agree with the referee, we have added a paragraph to clarify the informed patient status and a Supplemental dataset (Dataset S5) describing the clinical and pathological characteristics of the in-house cohort (page 1 of Dataset S5), and TCGA cohort (page 2 of Dataset S5).

- All the multi-omics data reported in this study, such as RNA-Seq, ChiP-Seq and CUT&Tag, should be deposited in a public repository, such as Gene Expression Omnibus (GEO) or ArrayExpress.

We totally agree with the referee, the data was submitted to the GEO data base with accession number GSE268093.

No pre-clinical model system is described in this study. Only cell lines in vitro and several patient samples.

We agree that we do not present results from a pre-clinical model system, nevertheless we report a novel cohort of 38 patient samples, one of which presents a novel fusion partner (GTF2H3), and confirmed the observations in a second public cohort of 55 patients. Regarding the rarity of these tumors, this is represented significant number of patients.

- Some typos should be double checked. For instance, UOK129 in Fig. 1B should be UOK120. This has been corrected.

Referee #3 (Remarks for Author):

In this manuscript Helleux A et al investigated the oncogenic mechanisms by which TFE3 fusion proteins drive tRCC. The authors report that high nuclear accumulation of NONO-TFE3 or PRCC-TFE3 fusion proteins promotes binding across the genome at H3K27ac-marked active chromatin and engage M/E-box containing regulatory elements to activate gene expression and stimulate mRNA synthesis. Furthermore, TFE3 fusions directly regulate genes involved in ferroptosis resistance and oxidative phosphorylation metabolism. The authors also report that tRCC tumor aggressiveness is associated with EMT and the presence of mesenchymal tRCC cancer cells associates with myofibroblast cancer-associated fibroblasts. Overall, the authors define tRCC as a novel metabolic subtype of renal cancer suggesting that genomic binding of TFE3 fusion proteins promotes tRCC tumorigenesis by regulating oxidative phosphorylation and ferroptosis resistance.

We would like to thank reviewer 3 for his thorough review of our manuscript and kind evaluation of our work.

1. The authors should discuss recent papers on the oncogenic properties of different TFE3 fusion proteins and the OxPhos transcriptional program in tRCC (Li J et al TFE3 fusions direct an oncogenic transcriptional program that drives OXPHOS and unveils vulnerabilities in translocation renal cell

carcinoma BioRxiv August 10, 2024; Damayanti N et al TFE3-Splicing Factor Fusions Represent Functional Drivers and Druggable Targets in Translocation Renal Cell Carcinoma. Cancer Res April 15, 2024). Thus, the authors should highlight the novelty of their findings.

We agree with the referee and as mentioned above we have included discussion of:

-The pre-print of Li et al., <https://doi.org/10.1101/2024.08.09.607311> who provide data supporting the idea that the ASPSCR1-TFE3 fusion has similar abilities to activate the OxPhos program.

-The pre-print by Li et al., <https://doi.org/10.1101/2024.10.24.620074> who reported that numerous genes involved in mitochondrial function and OxPhos, in particular components of the electron transport chain that we define here as direct TFE3 targets were selective vulnerabilities of tRCC. These data are complementary to those shown here and strongly consolidate the idea the TFE3-fusions directly activate OxPhos genes that are essential for tRCC tumorigenesis.

-The study of Sicinska et al who characterized genomic binding of ASPSCR1-TFE3 in soft part alveolar sarcomas (PMID 38657118). In this study, they performed HA-ChIP-seq in sarcoma cells ectopically expressing a HA-tagged ASPSCR1-TFE3 fusion protein finding that it binds broadly to active chromatin sites and is enriched at proximal promoters. While these findings are analogous to ours, *stricto-senso*, it could have been argued that this was a consequence of ectopic overexpression or a specific property of the ASPSCR1-TFE3 fusion associated with more aggressive tumours. Nevertheless, we make the same observations on the endogenous NONO-TFE3 and PRCC-TFE3 proteins in tRCC cells using a different technique (Cut&Tag vs ChIP-seq). Hence, these data together show that promiscuous promoter binding is a common property of TFE3 fusion proteins shared in both tRCC and soft part alveolar sarcomas. As mentioned above, our results go one step further reporting that the consequence of this is a general stimulation of mRNA synthesis, a phenomenon previously shown to contribute to oncogenic transformation.

-The results of Damayanti N et al who used deep learning-based in silico modeling to reveal new protein structures and changes in conformation of TFE3 fusion proteins as compared with their native unfused counterparts. Specifically, they suggested that TFE3-fusions have larger C-terminal domains and increased binding sites for DNA, RNA and protein. They further suggest that the conformation of the bHLH-LZ domain is altered with PRCC-TFE3 showing the closest conformation to the native TFE3 bHLH-LZ. These observations provide a potential framework to understand the observation that the NONO-TFE3 and PRCC-TFE3 fusions bind promiscuously over the genome through both binding to E/M-box motifs and via indirect tethering via other transcription factors at the proximal promoter. It is also intriguing that the PRCC-TFE3 is the fusion with the closest predicted conformation to the native TFE3 bHLH-LZ, and that in the tRCC lines, we saw predominant binding via E/M-box motifs, similar to overexpressed native TFE3, whereas NONO-TFE3 showed enhanced indirect binding via tethering. We also discuss the idea that these TFE3 fusions alter splicing and how this may be related to their potential interactions with the RNA Pol II CTD. The fact that splicing and transcription are functionally linked suggests that both are different faces of the same properties of the fusions explained by their promiscuous promoter recruitment.

2. The authors should comment on why in In Fig. 1, despite a very modest KD of TF3 protein expression in UK109 cells as compared with the other cell lines, there is still a significant inhibition of cell proliferation.

We agree with the referee. For the UOK109 and TF1 cells, the expression levels are very high and the blots are rapidly saturated. In fact, the levels of silencing are much more evident in both the RT-qPCR experiments, Fig. 1A or the RNA-seq (Fig. S2A and B). We have repeated these experiments and we provide new blots for UOK109 and TF1 where we loaded 10 less protein extract, leading to much less saturated signals and a better correlation between the protein and RNA-levels following siTFE3.

In addition, the Western blots show differential TFE3 protein expression across the different models but the Seahorse analysis for baseline OCR does not seem to correlate with that (TF1 and UK146 cells appear to have similar baseline OCR despite significant difference in TFE3 protein expression). The authors should elaborate more on whether additional factors might be involved in regulating OxPhos in these models.

We agree with the referee. One possible explanation is that the lower levels of the PRCC-TFE3 fusion protein are already sufficient to fully saturate the binding sites at the OxPhos genes. This is borne out by the Cut&Tag data where strong occupancy similar to that of NONO-TFE3 was observed. However, we cannot exclude that other parameters/factors may be involved and we have discussed this in the revised version.

3. What was the rationale for introducing four ccRCC lines (RCC4, UOK121, A-498, and 786-0) but only progressing with two (UOK121 and 786-0-Figure 1G)? Why were these two lines chosen over RCC4 and A-498?

We chose these 2 lines since they had the highest levels of endogenous TFE3. Nevertheless, while in revision we note that the pre-print manuscript of Li et al <https://doi.org/10.1101/2024.08.09.607311> reported that TFE3 does not activate OxPhos in ccRCC lines.

4. The overlap in upregulated oxidative phosphorylation consistent across HEK+TFE3, HEK+NT, and HEK+PT challenges the claim that "TFE3 fusion proteins activated genes of the OxPhos pathway leading to enhanced OxPhos metabolism." The results instead suggest the increase in oxidative phosphorylation is driven by TFE3, independent of the fusion. A greater discussion on the potential specific role of the fusion protein (i.e fusion partner) should be provided.

We fully agree with the referee. We have modified the text to highlight that when TFE3 is overexpressed and accumulates in the nucleus, it can activate the OxPhos program although less potently than the fusion proteins. This can be observed looking at the heatmap in Fig 2G, where the strongest activation is seen with PRCC-TFE3, that accumulates to levels equivalent or lower than native TFE3 and more than NONO-TFE3. Similarly, as mentioned above in response to referee 1, the transcriptomic results showed that each fusion protein activated a larger contingent of genes than native TFE3. The fusion proteins therefore have enhanced activation potential compared to native TFE3, possibly through specific interactions with additional cofactors as mentioned above. We have modified the text to acknowledge these issues.

5. The authors should comment on potential mechanisms responsible for a differential sensitivity to ferroptosis induction between TF1 and UK109 cells (Fig. 3I) which carry the same fusion protein.

As mentioned above in response to referee 1, it is possible that TF1 shows a higher IC50 due to the fact that this is a primary cell line from a patient previously treated with sunitinib and that had developed resistance to this drug.

6. The experimental platform would benefit from including an ASPSCR1-TFE3 fusion model since the authors show that PRCC and NONO somewhat cluster together in patients while ASPSCR1 seems to have a different phenotype.

We agree with the referee that ideally inclusion of an ASPSCR1-TFE3 model would have been useful. Nevertheless, we note that Li et al. report in their preprint use of ASPSCR1-TFE3 models with comparable conclusions to those drawn here. Similarly, as mentioned above, Sicinska et al (PMID 38657118) characterized genomic binding of ASPSCR1-TFE3 in sarcomas by CHIP-seq finding that it binds broadly to active chromatin sites and is enriched at proximal promoters as shown here using Cut&Tag in tRCC lines. All of these data support the idea of shared mechanisms of action of many if not all TFE3 fusions.

7. Regarding the patient data cohorts, the authors identified Cluster 1 and 2 as being more oxidative while Cluster 3 was primarily associated with EMT and poorer prognosis. ASPSCR1 fusion is already known to be more aggressive and has been reported to induce EMT (i.e Prakasam G et al Comparative genomics incorporating translocation renal cell carcinoma mouse model reveals molecular mechanisms of tumorigenesis JCI February 22, 2024).

The authors should also elaborate more on the potential mechanisms underlying differential expression of EMT across the different fusion proteins despite a similar OxPhos induction.

We agree with the referee, existing data shows that tumours with ASPSCR1-TFE3 fusion have a strong propensity to undergo EMT. The most likely explanation is that while activation of OxPhos is common to most if not all TFE3 fusions, each fusion can likely also activate additional specific programs (as confirmed by the HEKT results shown in Fig. S3A-B showing activation of specific gene sets by each fusion). The EMT program seems to be one of the main additional programs targeted by ASPSCR1-TFE3. We have discussed this in the revised version.

Minor:

1. In the abstract the following sentence seems to be incomplete: We further show that tRCC tumour aggressiveness is related to EMT which although enriched in tumours harbouring ASPSCR1-TFE3 fusions.

This has been corrected.

2. The statement that tRCC is "highly resistant to immune checkpoint inhibitors" (page 24) is not correct as different studies have shown activity in this patient population (Lee CH et al ,. Phase II Trial of Cabozantinib Plus Nivolumab in Patients With Non-Clear-Cell Renal Cell Carcinoma and Genomic Correlates. J Clin Oncol. 2022; Albiges L et al , Pembrolizumab plus lenvatinib as first-line therapy for advanced non-clear-cell renal cell carcinoma (KEYNOTE-B61): a single-arm, multicentre, phase 2 trial. Lancet Oncol. 2023).

We agree with the referee, this has been corrected.

17th Feb 2025

Dear Prof. Malouf,

Thank you for submitting your revised study, and please accept my apologies for the delay in getting back to you as one referee needed more time to complete his/her review. We have now received the reports from the referees who evaluated your revised manuscript. As you will see from the reports below, they are overall satisfied with the revisions, and I will therefore be able to accept your manuscript once the following editorial issues are addressed:

1/ Referees' comments:

Please consider the comment of referees #1 & #2 and address them via adequate discussion/editing. No additional experiment is requested at this stage.

2/ Manuscript text:

- Please remove the blue font and only keep in track changes mode any new modification.
- Authors: there is a discrepancy between Thibaut Tricard in the manuscript file vs. Thibault Tricard in the submission system, please correct. The corresponding author's email address should be listed on the title page.
- We can accommodate a maximum of 5 keywords, please adjust accordingly.
- Methods:
 - o Cells: please indicate whether the cells were authenticated and tested for mycoplasma contamination.
 - o Please provide antibodies dilutions/concentrations.
 - o Statistical analysis: please provide a statement on randomization.
- Data Availability: please provide a URL to access the dataset.
- Acknowledgements. The information provided in the manuscript and the submission system should match, please adjust accordingly (currently, Ministère de la Recherche and the Ligue Nationale contre le Cancer/'France Génomique' consortium (ANR-10-INBS-0009) are missing in the submission system). "MSD-AVENIR" is provided in the Comments box and needs to be removed
- Please rename "Conflict of interest statement" to "Disclosure statement and competing interests". Please review our updated policy <https://www.embopress.org/competing-interests> and update your competing interests if necessary. This section should be placed below Acknowledgments.
- Author contributions: CRediT has replaced the traditional author contributions section because it offers a systematic machine readable author contributions format that allows for more effective research assessment. Please remove the Authors Contributions from the manuscript and use the free text boxes beneath each contributing author's name in our system to add specific details on the author's contribution. More information is available in our guide to authors.
- Please rename "Bibliography" to "References".

3/ Figures and Appendix:

- Appendix Figure S1B TEF3: please clarify in the legend whether same blots with different exposures are displayed, or different blots.
- Dataset EV legends: there are 7 datasets uploaded, but the nomenclature (source file names, titles, manuscript callouts) needs to be updated to Dataset EV1-Dataset EV7; the legends need to be removed from the manuscript file and each should be provided in the corresponding Excel file (as separate tab/sheet).
- Appendix: the 8 appendix figures need to be merged in a single Appendix pdf file, with page number and table of content. Each legend should follow its figure and be removed from the manuscript.
- Please address the queries from our copy editors in the figure legends:
 1. Please define the annotated p values ****/****/**/* as well as provide the exact p-values for the same in the legend of figure 2A as appropriate.
 2. Please note that the exact p values are not provided in the legends of figures 1A, C, D, H, I; 2I, 3D, 4D, 6D, E; EV5 G
 3. Please indicate the statistical test used for data analysis in the legends of figures 1D, 2E, 3E, EV1 C, EV3 B, EV4 D, E; EV5 F.
 4. Please note that in figures 4D there is a mismatch between the annotated p values in the figure legend and the annotated p values in the figure file that should be corrected.
 5. Please note that information related to n is missing in the legends of figures 1H, 4D, EV5 G.
 6. Please note that the error bars are not defined in the legends of figures 1A, C, G, H, I; 2A, H, I; 3D, 6B, D, E.

4/ Checklist:

- Please fill in the manuscript and author information (top left corner).
- Please fill in all subsections in the "experimental study and statistics" section.

5/ Reagents and Tools table:

Please download and fill our Reagents and Tools Table template (.docx), which you can find in our author guidelines:

6/ Synopsis:

- please remove the synopsis text from the manuscript file and upload it as an individual file. Note that the stand-first should be a full sentence (i.e. the gene expression programs regulated by TFE3 fusion proteins in translocation renal cell carcinoma had not been elucidated to date, or something similar to the beginning of your Paper Explained).
- please also remove the synopsis image from the manuscript file, and upload it as a TIFF, jpeg, or png file 550 px wide x 300-600 px high. Please ensure that the resolution is sufficient, and that the text remains legible.

7/ As part of the EMBO Publications transparent editorial process initiative (see our Editorial at <http://embomolmed.embopress.org/content/2/9/329>), EMBO Molecular Medicine will publish online a Review Process File (RPF) to accompany accepted manuscripts.

This file will be published in conjunction with your paper and will include the anonymous referee reports, your point-by-point response and all pertinent correspondence relating to the manuscript. Let us know whether you agree with the publication of the RPF and as here, if you want to remove or not any figures from it prior to publication.

We note that you deposited your revised manuscript on bioRxiv (additionally to the initial version). Please know that this is usually not allowed, as it creates multiple versions of record for a same manuscript. However, there is the possibility to link the initial deposited manuscript to the final published article.

I look forward to receiving your revised manuscript.

Yours sincerely,

Lise Roth

***** Reviewer's comments *****

Referee #1 (Remarks for Author):

I do agree that this manuscript is significantly improved in terms of clarity and content. I still believe that the Discussion is too lengthy and needs to be made concise. But with the added improvements and the rearrangement for better flow it is a great body of work.

Referee #2 (Comments on Novelty/Model System for Author):

2. Several recent publications, including preprints, have already described the involvement of oxidative phosphorylation, ferroptosis (glutathione metabolism) and EMT in tRCC tumourigenesis. This study expands the molecular mechanism behind some of these processes at the transcriptional level of genomic binding.

3. Despite the advances in the understanding of the molecular mechanisms of tRCC tumourigenesis in this study, its medical feasibility is limited because potential therapeutic targets for tRCC have not been adequately tested in vitro or preferentially in preclinical models of tRCC.

4. No preclinical model system is described in this study. Only in vitro cell lines and several patient samples.

Referee #2 (Remarks for Author):

The authors have addressed most of the concerns raised by this reviewer. However, the first two points of the Synopsis included in the revision (as well as the entire manuscript) should be reworded to emphasise the novelty of this study, as these general statements have been described in previously published studies. Despite the advances in the understanding of the molecular mechanisms of tRCC tumorigenesis in this study, its medical feasibility is limited because potential therapeutic targets for tRCC have not been adequately tested in vitro or preferentially in preclinical models of tRCC. Therefore, acceptance of this manuscript is at the editor's discretion.

Referee #3 (Remarks for Author):

The authors have addressed the points raised by the reviewers.

***** Reviewer's comments *****

Referee #1 (Remarks for Author):

I do agree that this manuscript is significantly improved in terms of clarity and content. I still believe that the Discussion is too lengthy and needs to be made concise. But with the added improvements and the rearrangement for better flow it is a great body of work.

We thank the referee for the positive comments. We are glad that our reorganization of the manuscript and the addition of novel data addressed the original concerns. We have gone again through the Discussion to shorten it, removing redundant sentences and reorganizing some paragraphs.

Referee #2 (Comments on Novelty/Model System for Author):

2. Several recent publications, including preprints, have already described the involvement of oxidative phosphorylation, ferroptosis (glutathione metabolism) and EMT in tRCC tumourigenesis. This study expands the molecular mechanism behind some of these processes at the transcriptional level of genomic binding.

3. Despite the advances in the understanding of the molecular mechanisms of tRCC tumourigenesis in this study, its medical feasibility is limited because potential therapeutic targets for tRCC have not been adequately tested in vitro or preferentially in preclinical models of tRCC.

4. No preclinical model system is described in this study. Only in vitro cell lines and several patient samples.

As previously discussed in the original rebuttal, the primary aim of this study was not to identify novel therapeutic targets, but to understand in mechanistic terms the basis for the high OxPhos signature of tRCC that had previously been noted in other publications. As a result, our study provides the most comprehensive understanding of TFE3-fusion-driven gene expression to date.

We agree with the referee that we did not use pre-clinical models, but we did perform the most in depth analyses of human patient data to date. Compared to previous studies, we used stratified ccRCC tumours as comparison to address the issue of OxPhos signatures and we analyzed the expression of the ferroptosis related genes linking their expression to the binding of TFE3-fusion proteins in cell models. Moreover, we analyzed the patient gene expression data by deconvolution revealing a novel association between tRCC EMT and the presence of myCAFs in the microenvironment highlighting the association of this population with poor disease outcome. These are important and novel features of clinical relevance that were not previously in the literature.

Referee #2 (Remarks for Author):

The authors have addressed most of the concerns raised by this reviewer. However, the first two points of the Synopsis included in the revision (as well as the entire manuscript) should be reworded to emphasise the novelty of this study, as these general statements have been described in previously published studies.

As suggested by the referee we have reworded the Synopsis and parts of the manuscript to emphasize the novelty of this study.

Despite the advances in the understanding of the molecular mechanisms of tRCC tumorigenesis in this study, its medical feasibility is limited because potential therapeutic targets for tRCC have not been adequately tested in vitro or preferentially in preclinical models of tRCC. Therefore, acceptance of this manuscript is at the editor's discretion.

See response above.

Referee #3 (Remarks for Author):

The authors have addressed the points raised by the reviewers.

We thank the referee for the positive comments. We are glad that our reorganization of the manuscript and the addition of novel data addressed the original issues.

4th Mar 2025

Dear Prof. Malouf,

Thank you for submitting your revised files. I am pleased to inform you that your manuscript is accepted for publication and is now being sent to our publisher to be included in the next available issue of EMBO Molecular Medicine!

Yours sincerely,

Lise Roth
